# Zika viruses encode 5′ upstream open reading frames affecting infection of human brain cells

Charlotte Lefèvre [1], Georgia M. Cook [1], Adam M. Dinan [1,7], Shiho Torii[2], Hazel Stewart [1], George Gibbons[3], Alex S. Nicholson [4], Liliana Echavarría-Consuegra[1], Luke W. Meredith[1], Valeria Lulla [1], Naomi McGovern [5], Julia C. Kenyon [1], Ian Goodfellow [1], Janet E. Deane [4], Stephen C. Graham [1], András Lakatos [3,6], Louis Lambrechts [2], Ian Brierley[1] & Nerea Irigoyen [1] ✉

Zika virus (ZIKV), an emerging mosquito-borne flavivirus, is associated with congenital neurological complications. Here, we investigate potential pathological correlates of virus gene expression in representative ZIKV strains through RNA sequencing and ribosome profiling. In addition to the single long polyprotein found in all flaviviruses, we identify the translation of unrecognised upstream open reading frames (uORFs) in the genomic 5′ region. In Asian/American strains, ribosomes translate uORF1 and uORF2, whereas in African strains, the two uORFs are fused into one (African uORF). We use reverse genetics to examine the impact on ZIKV fitness of different uORFs mutant viruses. We find that expression of the African uORF and the Asian/American uORF1 modulates virus growth and tropism in human cortical neurons and cerebral organoids, suggesting a potential role in neurotropism. Although the uORFs are expressed in mosquito cells, we do not see a measurable effect on transmission by the mosquito vector in vivo. The discovery of ZIKV uORFs sheds new light on the infection of the human brain cells by this virus and raises the question of their existence in other neurotropic flaviviruses.

Zika virus (ZIKV) is an emerging *Aedes* mosquito-borne *Flavivirus*, a genus which also includes dengue, yellow fever, Japanese encephalitis, tick-borne encephalitis and West Nile viruses (WNV). ZIKV was isolated initially from a febrile monkey in Uganda in 1947[1] but was considered of low importance as most human infections appeared to be asymptomatic. However, it gained prominence from 2007, when the first large epidemic was reported from Yap Island in Micronesia and spread

to French Polynesia in 2013[2]. In 2015, the virus reached Brazil and spread across the Americas. Typically, symptoms associated with ZIKV are mild and might include fever, maculopapular rash, conjunctivitis and myalgia. In adults, the virus can cause severe neurological symptoms, including Guillain-Barré syndrome[2], and infection of pregnant women can lead to vertical transmission of the virus to cells of the developing brain of the foetus[3]. A key pathological aspect of such

[1]Division of Virology, Department of Pathology, University of Cambridge, Cambridge, UK. [2]Institut Pasteur, Université Paris Cité, CNRS UMR2000, Insect-Virus Interactions Unit, Paris, France. [3]John van Geest Centre for Brain Repair, University of Cambridge, Cambridge, UK. [4]Cambridge Institute for Medical Research, University of Cambridge, Cambridge, UK. [5]Division of Immunology, Department of Pathology, University of Cambridge, Cambridge, UK. [6]Cambridge Stem Cell Institute, Cambridge, UK. [7]Present address: Department of Medicine, MRC Laboratory of Molecular Biology, University of Cambridge, Cambridge, UK. ✉e-mail: ni236@cam.ac.uk

infection is impaired development of the unborn brain, leading to gestational abnormalities, including congenital Zika syndrome (CZS). The most well-known feature of CZS is microcephaly[4], but the syndrome also includes general neurological impairment, neurosensory alterations, delays in motor acquisition and an 11-fold greater risk of death during the first 3 years of life[5,6]. The devastating impact of ZIKV on newborns has far-reaching social consequences, including stigmatisation of affected mothers and babies plus an increase in illegal abortions across Latin America[7]. Recent epidemiolocal studies also indicate that immunity to ZIKV may not persist for as long as previously thought, with neutralising antibody levels declining over time in adults[8].

The relative pathogenesis of African and Asian/American ZIKV isolates has been the subject of debate. Whereas the Asian/American lineage is responsible for the 2015/2016 American epidemic and has been associated with significant neurological problems (e.g., microcephaly), more serious disease manifestations and pathogenesis of the African strain may have been missed in the past because, paradoxically, they were more severe (i.e., early abortion instead of birth defects) and thus, less visible[9–11]. On the other hand, it is possible that the scarcity of clinical reports of microcephaly in Africa might be due to inadequate public health infrastructure and surveillance systems. Here, we investigate potential pathological correlates of virus gene expression in representative strains of the Asian/American and African ZIKV lineages in mammalian and mosquito cells through RNA sequencing (RNA-Seq) and ribosome profiling (Ribo-Seq). Like other flaviviruses, ZIKV has a positive-sense, single-stranded RNA genome (gRNA) of ~11 kb, which contains a single long open reading frame (ORF) flanked by 5′ and 3′ untranslated regions (UTRs) of approximately 100 and 400 nucleotides, respectively. The ORF encodes a large polyprotein that is cleaved by host and viral proteases to yield three structural proteins derived from the N-terminal region (capsid−C, precursor/membrane−pr/M and envelope−E) and seven non-structural proteins (NS1, NS2A, NS2B, NS3, NS4A, NS4B, NS5)[12]. We show here that in addition to the polyprotein, ZIKV encodes previously unrecognised upstream ORFs (uORFs) in the 5′ UTR, initiating from non-AUG start codons. In African ZIKV isolates, a single uORF (African uORF) is present, whereas, in Asian/American strains, this uORF is split into two uORFs (uORF1 and uORF2) by insertion of a single nucleotide.

The impact on ZIKV fitness of the expression of single or dual uORFs was explored by analysing a panel of mutant viruses. We find that expression of the African uORF and the American uORF1 can modulate virus growth, virulence and tropism in human brain organoids but has a limited role in the mosquito vector. These experiments reveal novel players with a potential role in ZIKV neurotropism.

## Results

### Ribosome profiling reveals the presence of novel uORFs in the 5′ UTR of different ZIKV strains

We utilised Ribo-Seq, in combination with whole transcriptome sequencing (RNA-Seq), to investigate the translation of the ZIKV genome at sub-codon resolution. Ribo-Seq exploits the capacity of elongating ribosomes to protect ~30 nucleotides of messenger RNA (mRNA) from digestion during nuclease incubation of cell extracts[13]. Such ribosome-protected fragments (RPFs) are purified and deep sequenced, revealing the position of translating ribosomes on the mRNA at the time of harvesting with single-nucleotide precision[14]. African green monkey (Vero) cells and human glioblastoma–astrocytoma (U251) cells were infected at a multiplicity of infection (MOI) of three with the ZIKV American isolate PE243, representative of the Asian/American lineage[15], and the ZIKV African isolate Dak84, a prototypic African isolate[16]. To preserve the positions of translating ribosomes upon cell lysis, infected cells at 24 h post-infection (h p.i.) were flash-frozen or pre-treated for 3 min with the translation inhibitor cycloheximide (CHX) before harvesting. CHX treatment is widely used in Ribo-Seq studies but can lead to the accumulation of 80S ribosomes on start codons and, in stressed cells, can induce the accumulation of RPFs in the 5′ region of coding sequences[17]. For this reason, cells were harvested by flash-freezing to avoid these potential biases (unless stated). Ribo-Seq and RNA-Seq libraries were prepared and deep sequenced as previously described[18]. In addition, quality control of the different libraries was also conducted as previously described[18], and the data were deemed to be of high quality (Supplementary Figs. 1–6).

At 24 h p.i., the viral envelope (E) protein of each strain is robustly expressed in infected Vero and U251 cells confirming that the infection was well-established (Fig. 1A). The Ribo-Seq (red) and RNA-Seq (green) read densities on the virus genome for infected Vero and U251 cells are illustrated in Fig. 1B and Supplementary Fig. 7A–C. Consistent with the translation of a single polyprotein from the genomic mRNA, read coverage across the main ORF was even, although localised variations in RPF density appear, that may arise from technical biases (ligation,

**A.**

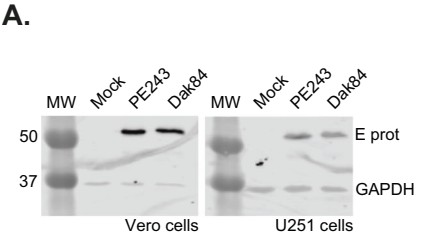

**B.**

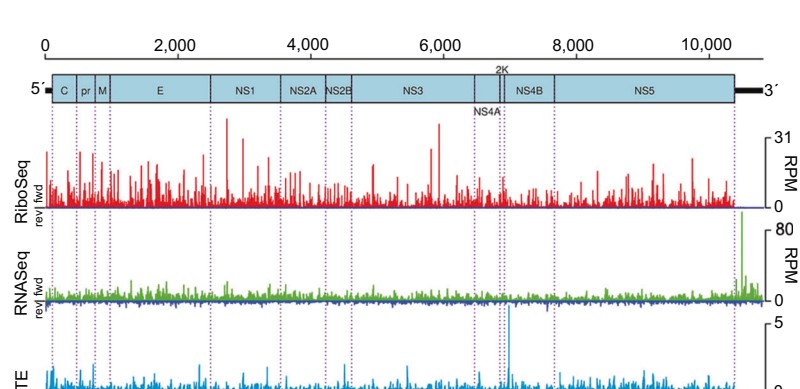

**Fig. 1 | ZIKV RNA synthesis and translation. A** Western blot analysis of ZIKV E protein and GAPDH in Vero and U251 cells infected with American isolate PE243 and African isolate Dak84 (MOI:3) for 24 h. GAPDH was used as a loading control. Molecular masses (kDa) are indicated on the left. Infections were performed in triplicate with similar results. **B** Map of the 10,807-nt ZIKV/Brazil/PE243/2015 genome. The 5′ and 3′ UTRs are in black, and the polyprotein ORF is in pale blue with subdivisions showing mature cleavage products. Histograms show the read densities, in reads per million mapped reads (RPM), of Ribo-Seq (red) and RNA-Seq (green truncated at 100 RPM for better visualisation) reads at 24 h p.i. (repeat 1) in Vero cells pre-treated with CHX. The positions of the 5′ ends of reads are plotted with a +12 nt offset to map (for RPFs) approximate P-site positions. Negative-sense reads are shown in dark blue below the horizontal axis. In light blue, the translational efficiency (TE) is calculated as the positive-sense Ribo-Seq/RNA-Seq ratio. Source data are provided as a Source Data file.

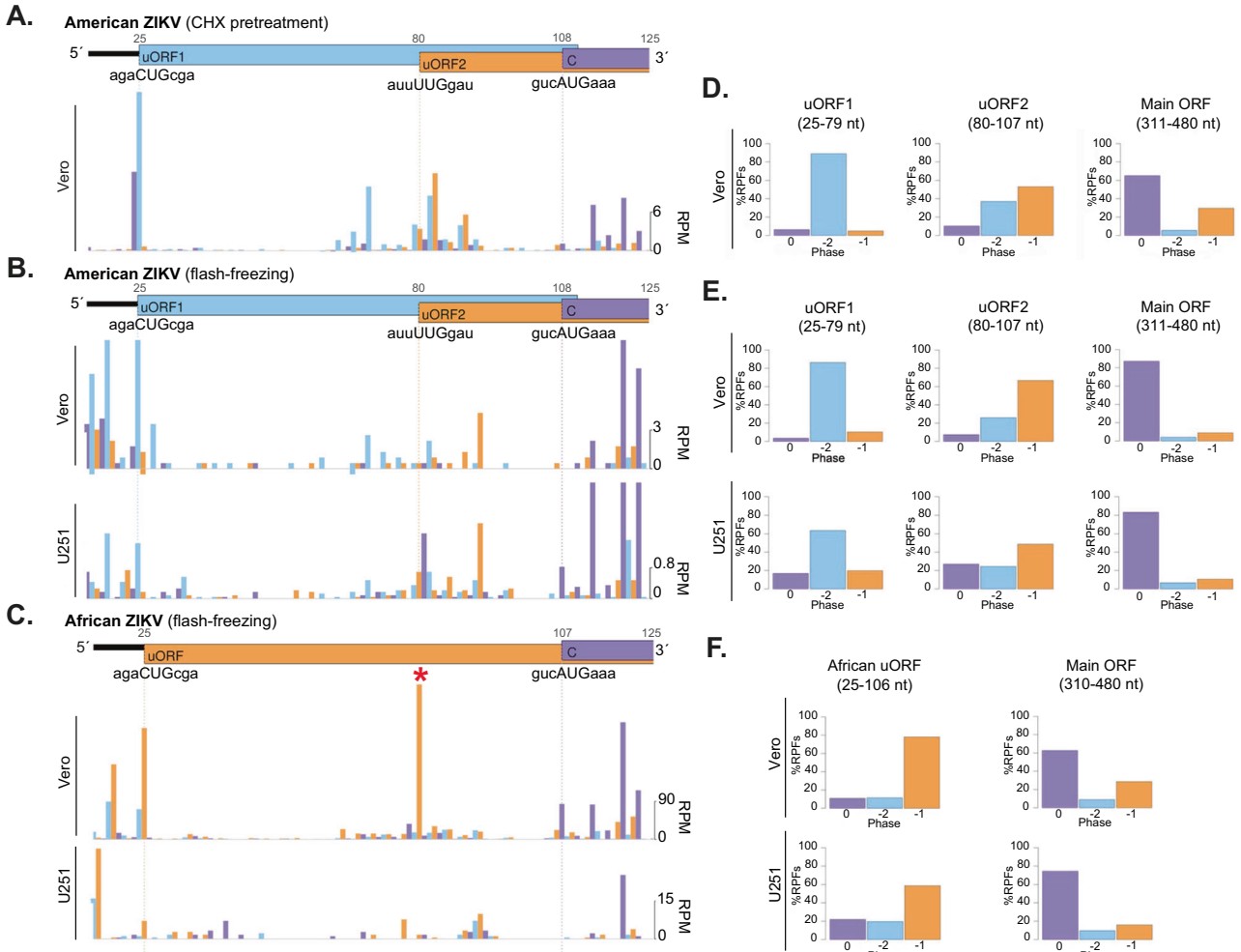

**Fig. 2 | ZIKV 5′ region.** The 5′ region of the ZIKV American isolate PE243 genome shows two non-AUG uORFs in Vero cells pre-treated with CHX (**A**), flash-frozen Vero cells (**B**, upper panel) and flash-frozen U251 cells (**B**, lower panel). Note that in order to visualise RPFs across the uORFs properly, the *y*-axis has been truncated at 10 RPM for the Ribo-Seq samples for Vero cells and 3 RPM for U251 cells, leaving some RPF counts, mainly for the main ORF, off-scale. **C** The 5′ region of the ZIKV African isolate Dak84 genome shows the African uORF in flash-frozen Vero (upper panel) or flash-frozen U251 cells (lower panel). Note that in Vero-infected cells, the peak marked with a red asterisk, unlike all other peaks, has an unusual read-length distribution centred on 26 nt, and this is not seen in U251-infected cells. For U251 cells infected with Dak84, only RPFs with a read length of 28 and 29 were plotted. Histograms show the positions of the 5′ ends of reads with a +12 nt offset to

map the approximate P-site. Reads whose 5′ ends map to the first, second or third phases relative to codons in the polyprotein reading frame are indicated in blue, orange or purple, respectively. Capsid (C) denotes the initiation of the polyprotein (main ORF). **D**–**F** Bar charts of the percentage of ribosome-protected fragments (RPFs) in each phase relative to the polyprotein ORF. Regions with the least amount of overlapping, (coordinates given in plot titles), were selected. Reads whose 5′ ends map to the −2, −1 or 0 phases are indicated in blue, orange or purple, respectively. CHX-treated Vero cells infected with American isolate PE243 (**D**); flash-frozen Vero (**E**, upper panel) and U251 (**E**, lower panel) cells infected with PE243; and flash-frozen Vero (**F**, upper panel) and U251 (**F**, lower panel) cells infected with African isolate Dak84.

PCR and nuclease biases[19]) or, potentially, from ribosome pausing during translation[20]. Few RPFs were present in the 3′ UTR, consistent with the absence of translation in this region. The 3′ UTR RNA-Seq density was noticeably higher (65–70%) than across the upstream part of the genome in each cell line (Fig. 1B). Structured flavivirus 3′ UTRs resist degradation by the 5′–3′ Xrn1 host exonuclease, giving rise to non-coding subgenomic flavivirus RNAs (sfRNAs) that accumulate during infection[21] and are linked to cytopathic and pathologic effects[15]. A sharp spike of RNA-Seq density is seen in the ZIKV American isolate PE243 (nt 10,478) and ZIKV African isolate Dak84 (nt 10,477) libraries (Supplementary Fig. 7D, E), consistent with the presence of a nuclease-resistant RNA structure at this position. Indeed, this location is two nucleotides upstream of the predicted 5′ end of RNA 'stem-loop 2' (SL2)[21].

In Ribo-Seq samples, the number of negative-sense reads (dark blue) was negligible (<0.005% of mapped reads), indicating that they

are unlikely to be genuine RPFs[14,20]. In comparison, in RNA-Seq samples (Fig. 1B and Supplementary Fig. 7A–C), a low, uniform coverage of negative-sense reads was observed (at ~1% c.f. positive sense), corresponding to negative-sense intermediates that act as templates for genome replication. The translational efficiency (TE) of each virus genome was calculated by applying a 15-nt running mean filter and dividing the number of Ribo-Seq reads by the number of RNA-Seq reads (light blue), revealing relatively even coverage across the genome. Strikingly, we found substantial TE within the 5′ UTR in both CHX-treated and flash-frozen cells (Fig. 2A, B, C). In the ZIKV American isolate PE243-infected cells, a prominent peak of RPF density was seen at nucleotide 25 of the 5′ UTR in Vero and U251 cells, coinciding with a non-canonical (CUG) initiation codon (Fig. 2A, B). RPFs mapped in the −2 phase along the length of the associated 29-codon upstream ORF (uORF1, in blue), which ends at nucleotide 111 (the 4th nucleotide of the polyprotein ORF). Additionally, RPFs mapped in the −1 phase to a

second uORF (uORF2, in orange), which appears to be translated via a non-canonical (UUG) initiation codon at nucleotide 80 (Fig. 2A, B). This uORF2 is 77 codons in length and extends 202 nucleotides into the polyprotein ORF. Analysis of the mapping of Ribo-Seq reads to the ZIKV African isolate Dak84 5′ UTR (Fig. 2C) revealed a single uORF (African uORF), translated in the −1 frame. The African uORF initiates at nucleotide 25 (the same initiation codon used by uORF1) and terminates at nucleotide 309, at the equivalent stop codon of uORF2 encoding a putative protein of 94 amino acids in length. In all African ZIKV isolates with available sequence data, uORF1 and uORF2 are present as a single ORF, which appears to have been split in two by the insertion of a uracil residue at position 81 in the 1966 Malaysian lineage that gave rise to the Asian/American strain[22] (Supplementary Fig. 8).

To provide further support for the presence of ZIKV uORFs, we examined the phasing of RPFs in viral 5′ UTRs. For RPFs, mapping of the 5′ end positions to coding sequences (CDSs) characteristically reflects the triplet periodicity (herein referred to as 'phasing') of translational decoding[14,18]. We summarised phase relative to the polyprotein (main) ORF by plotting bar charts of the percentage of reads in each phase (Fig. 2D–F). To avoid the potential confounding effects of overlapping ORFs on these calculations, regions with the least overlap were selected. For uORF1, ribosomal phasing was measured over a 55 nucleotide region (position 25–79), which does not overlap another ORF. Here, a clear dominance of the −2 phase was seen (light blue), consistent with translation in the uORF1 frame (Fig. 2D, E). For uORF2, a short region of 27 nucleotides (position 80–107) was selected, to avoid the high ribosomal occupancy in the 0 phase corresponding to the main ORF that starts at position 108. Here, the majority of reads map to the −1 phase, consistent with uORF2 translation; however, the overlap with uORF1 is still evident from the increased read density in the −2 phase (Fig. 2D, E). For the African uORF (Fig. 2F), within the chosen region (position 25–106), the majority of reads are attributed to the −1 phase, supporting the expression of this fused uORF. As a positive control, phasing within the polyprotein ORF was assessed, in a region with no known overlapping ORFs (position 310/311–480), revealing a clear dominance of reads attributed to the 0 phase (Fig. 2D–F). Additionally, as shown in Supplementary Fig. 9, the length distribution of Ribo-Seq reads mapping to the ZIKV uORFs mirrored that of polyprotein-mapping RPFs, indicating that they are bona fide ribosome footprints.

### The translation of ZIKV uORFs can modulate main ORF expression

The presence of uORFs in mRNAs is often associated with the regulation of downstream gene expression[23,24]. To investigate the potential modulation of ZIKV main ORF expression by the 5′ uORFs, capped T7 RNA polymerase-derived synthetic reporter mRNAs were prepared in which American PE243 uORF1, uORF2 or a 5′ portion of the main ORF was placed upstream of, and in frame with, the Firefly luciferase (FF-Luc) reporter gene. The ZIKV and FF-Luc sequences are separated by the short, foot and mouth disease virus 2A autoprotease-encoding sequence that liberates the FF-Luc enzyme following expression in cells (Fig. 3A). RNAs were reverse transfected into Vero cells alongside a T7-derived RNA expressing Renilla luciferase (Ren-Luc) as transfection control. FF-Luc and Ren-Luc activities were measured at 30 h post-transfection (h p.t.), and translation efficiencies were determined after normalisation with main ORF (American wild-type, American WT) translation levels as 100% (Fig. 3B). Under these conditions, uORF1 and uORF2 expression levels were 0.80% and 4.13%, respectively (Fig. 3B). To assess whether translation of the uORFs could affect main ORF translation, mutations were introduced into uORF1 and uORF2 that were predicted to reduce or increase their expression (Fig. 3C–E). As shown in Fig. 3E, mutation of the uORF1 start codon from CUG to CUA (uORF1-KO, blue) led to a modest reduction in uORF1 expression (left panel), no change in uORF2 expression (middle panel) and a small

increase in main ORF translation (right panel). Reducing uORF2 expression by changing the initiation codon from UUG to UUA (uORF2-KO, pink), or introducing a premature stop codon within uORF2 at residue number 6 (uORF2-PTC1, orange), reduced uORF2 translation by 50% (Fig. 3E, middle panel) and led to a slight decrease in main ORF expression (Fig. 3E, right panel). Replacing the uORF2 initiation codon with a canonical AUG codon (in purple) led to a substantial increase in uORF2 translation and prevented main ORF expression (Fig. 3E, middle and right panels). A fusion of uORF1 and uORF2 that recapitulated the African uORF (African-like; in green) did not significantly change uORF expression (Fig. 3E, middle panel) and led to a modest, albeit significant, increase in main ORF translation (Fig. 3E, right panel). Similar results were obtained in U251 cells transfected with main ORF-2A-FF-Luc mutants (Supplementary Fig. 10A).

We went on to ask whether viral infection could influence the relative utilisation of main and uORFs. In these experiments, reporter mRNA-transfected cells were infected at 6 h p.t. with the American isolate PE243 (MOI:3) and harvested 24 h later. In the context of infection, the expression of the uORFs and main ORF relative to each other remained similar, although the total expression of each ORF increased significantly compared to uninfected cells (Fig. 3F). Notably, the raw luciferase values (Supplementary Table 1) were slightly lower in the presence of the virus, which may reflect some impairment of translation initiation as a result of the phosphorylation of the alpha subunit of the initiation factor 2 (p-eIF2α) during infection (Supplementary Fig. 10B). In relative terms, expression of the transfection control mRNA, Ren-Luc, in comparison to the ZIKV FF-Luc mRNAs, was reduced in the presence of ZIKV (Supplementary Table 1). This might indicate that the viral 5′ UTR arrangement selectively preserves the expression of the main and uORFs during infection, although this requires further investigation. The effect of uORF mutations on main ORF expression in infected cells was also tested (Fig. 3G). In all cases, except uORF2-KO, the relative expression of the main ORF was increased modestly. In conclusion, virus infection modestly and uniformly increased expression from upstream and main ORFs.

### uORF translation modulates virus replication

To investigate the potential role of ZIKV uORFs in virus replication, a panel of viruses containing the mutations tested above was generated using reverse genetics of an American ZIKV infectious clone[25], detailed in Fig. 4A. Given that structured RNA elements and long-range interactions in the 5′ and 3′ terminal regions of the ZIKV genome are essential for virus translation and replication[26,27], we began by confirming that the 5′ end structures were retained in full-length RNA transcripts of the mutant viruses. Using selective 2′-hydroxyl acylation analysed by primer extension (SHAPE), we found that the structure of the 5′ UTR and the start of the main ORF of the mutant viruses generally very closely matched that of the American WT infectious clone (Fig. 4B and Supplementary Fig. 11A). Two differences were observed; modelling of the African-like mutant virus indicated a slightly shorter third stem-loop (cHP), with loss of two base pairs at the bottom of the helix (Fig. 4B), and in the uORF2-AUG mutant, the internal loop in the centre of the second stem-loop (SLB, Supplementary Fig. 11A) is base-paired. Mutant viruses were also analysed for the stability of the introduced mutations. After five passages, RT-PCR analysis of intracellular viral RNA (initially infected at MOI 0.01 PFU/cell) revealed that all mutations were stable, with two exceptions. ZIKV uORF2-KO showed reversion to WT after passage 1, and we could not recover any virus following electroporation of the ZIKV uORF2-AUG mutant (Supplementary Fig. 11B). The latter observation may reflect reduced translation initiation at the main polyprotein AUG as a consequence of increased recognition of the uORF2-AUG start codon. The rapid reversion to the WT sequence seen with the uORF2-KO virus may indicate a role for the uORF2 start codon in the translation of the viral

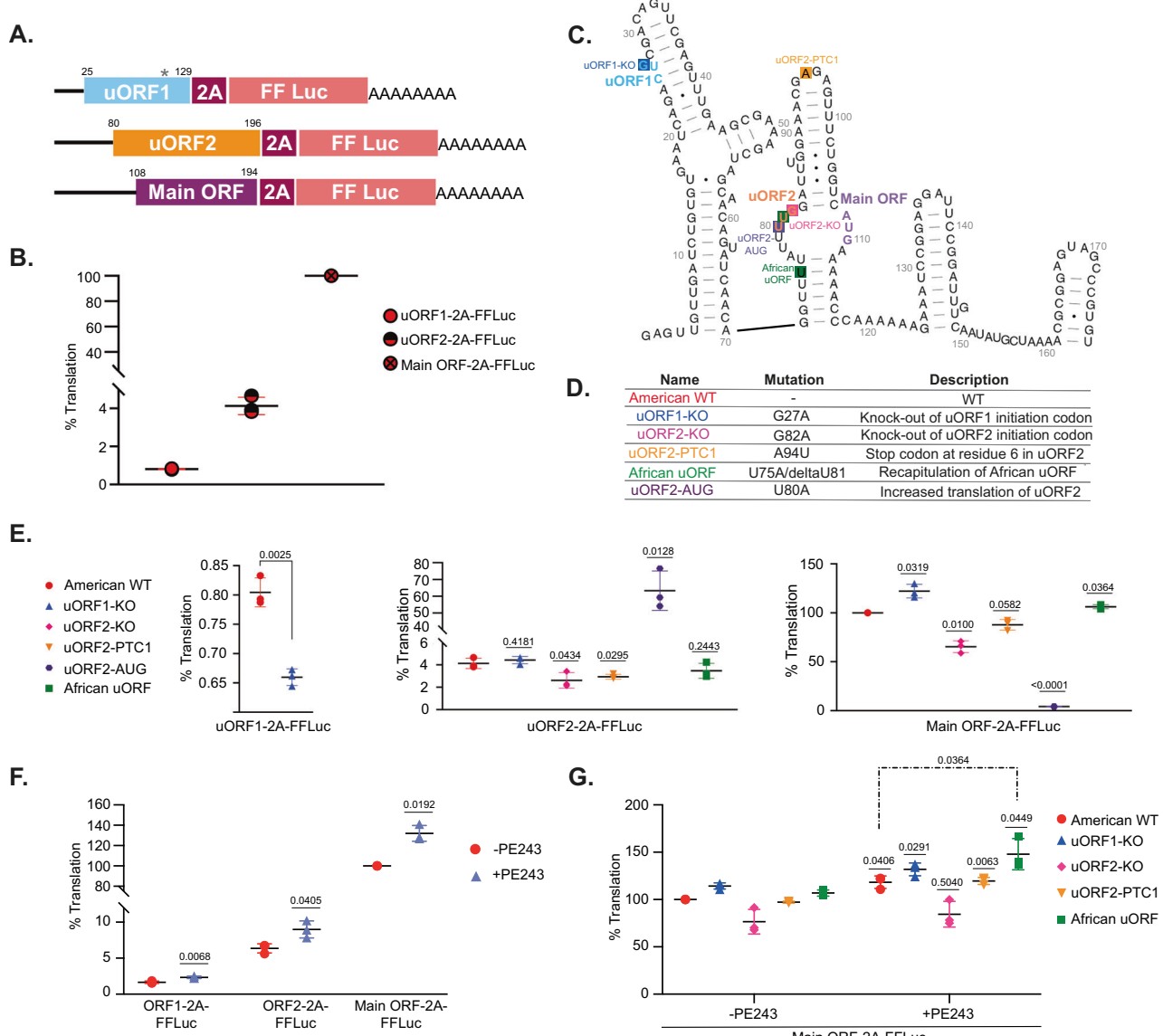

**Fig. 3 | Analysis of ZIKV uORF translation. A** Firefly luciferase (FF-Luc) reporter constructs scheme where the 2A-FF-Luc cassette is positioned downstream of and in frame with uORF1, uORF2 or main ORF. uORF1-2A-FF-Luc includes the complete 5′-UTR (107 nucleotides) plus 22 nucleotides of the polyprotein. In this case, a tryptophan residue substitutes the uORF1 stop codon (grey asterisk) to allow the luciferase reporter expression. uORF2-2A-FF-Luc includes the complete 5′-UTR plus 89 nucleotides of the polyprotein, and the main ORF-2A-FF-Luc includes the complete 5′-UTR plus 87 nucleotides of the polyprotein. Differences in protein length are due to frame correction. **B** Relative FF-Luc activity for uORF1, uORF2 and main ORF normalised to Renilla luciferase (Ren-Luc) used as a transfection control. Vero-transfected cells were harvested at 30 h post-transfection (h p.t.). One-hundred percent translation accounted for the main ORF WT translation. **C** Scheme of the 5′ region of American isolate PE243 with initiation codons for uORF1, uORF2 and main ORF indicated in blue, orange and purple, respectively. Modified nucleotides in the different mutants are indicated by a coloured-filled square

associated with the mutant name, described in (**D**). **E** Relative FF-Luc activity of different mutants for uORF1, uORF2 and main ORF normalised to Ren-Luc as described in (**B**). American WT FF-Luc activities of uORF1, uORF2 and main ORF from (**B**) have been included for clarity. **F** Relative FF-Luc/Ren-Luc ratio of uORF1-, uORF2- and main ORF-2A-FF-Luc reporter mRNAs in Vero cells infected with American isolate PE243 (MOI:3, purple triangle) or mock-infected (red circle) at 6 h p.t. Cells were harvested at 24 h p.i. **G** Relative FF-Luc/Ren-Luc ratio of main ORF-2A-FF-Luc mutant reporters in Vero cells infected with PE243 (MOI:3) or mock-infected at 6 h p.t. Cells were harvested at 24 h p.i. Experiments were performed in triplicate with three biological replicates. In all cases, error bars represent standard errors. All *t*-tests were two-tailed and did not assume equal variance for the two populations being compared. All *p* values are from comparisons of the mutant with the respective non-mutated luciferase reporter (i.e., derived from the American wild-type) in the same ORF. Source data are provided as a Source Data file.

polyprotein, but further experimental analysis will be required to confirm this, including the design and testing of alternative uORF2 knockout strategies.

To assess the growth and infectivity of the stable mutants, U251 cells were infected with sequence-verified American WT, African-like, uORF1-KO or uORF2-PTC1 viruses at low multiplicity (0.01 PFU/cell) in a multi-step growth experiment from 0 to 96 h p.i. As shown in Fig. 4C, from 48 h p.i. onwards, African-like and uORF1-KO mutant

viruses reached significantly higher titres (~6-fold) than the American WT, whereas no difference was observed with the uORF2-PTC1 mutant. This phenotype was confirmed in competition assays in which cells were simultaneously infected with a defined ratio (50:50 and 90:10) of American WT virus:corresponding mutant virus to a final MOI of 0.01 PFU/cell (Fig. 4D; note that a ratio of 10:90 was used for the American WT:uORF2-PTC1 competition experiment as the uORF2-PTC1 virus showed somewhat slower replication in multi-step growth

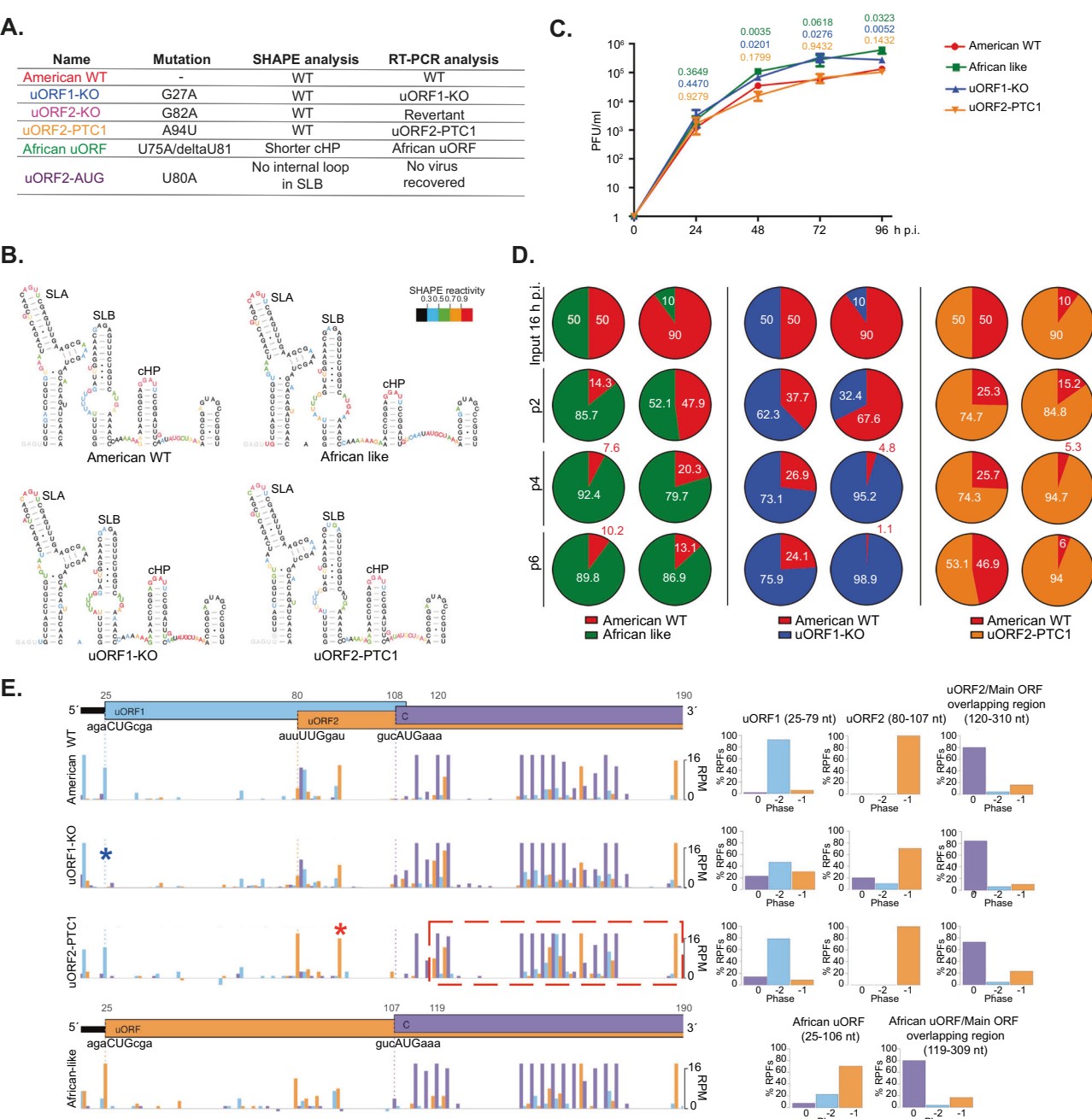

**Fig. 4 | The significance of uORF translation in virus infection. A** Summarising table of SHAPE and RT-PCR data for ZIKV uORFs mutant viruses derived from pCCI-SP6-American ZIKV infectious clone. **B** SHAPE RNA secondary structure of the 5′ region (first 180 nucleotides) of the American WT, African-like, uORF1-KO and uORF2-PTC mutant viruses. Nucleotides are colour-coded based on SHAPE reactivity. SLA stem-loop A, SLB stem-loop B and cHP capsid hairpin. **C** Time-course of infection of U251 cells with ZIKV mutant viruses (MOI 0.01) for 96 h. Plaque assays were performed on serial dilutions of the supernatant containing released virions. Values show the mean averages of the titration of three biological replicates. Error bars represent standard errors. PFU plaque-forming units. All *t*-tests were two-tailed and did not assume equal variance for the two populations being compared. All *p* values are from comparisons of the mutant virus with the American WT at the indicated time points. **D** Pie charts of competition assays of American WT with mutant viruses after 2 (p2), 4 (p4) and 6 (p6) sequential passages in U251 cells.

Different proportions of each virus (50:50 and 90:10) were added as indicated in the first row (input 16 h p.i., passage 0). Chart area indicates the RNA proportion for each virus, as measured from sequencing chromatograms of RT-PCR products. Experiments were repeated independently eight times (Supplementary Table 6). **E** (left panels) Ribo-Seq read density in the 5′ region of the ZIKV genome at 24 h p.i. of flash-frozen Vero cells infected with American WT, uORF1-KO, uORF2-PTC1 and African-like viruses (MOI:3). Histograms show the positions of the 5′ ends of reads with a +12 nt offset to map the P-site as described in Fig. 2A–C. For these plots, the *y*-axis has been truncated at 18 RPM and only RPFs with a read length of 29 was plotted. Blue asterisk indicates uORF1 initiation codon, and red asterisk is the premature termination codon in uORF2. **E** (right panels) Bar charts of the percentage of RPFs in each phase, relative to the main ORF (plot as described in Fig. 2D–F). Source data are provided as a Source Data file.

curves). Supernatants containing virus particles were harvested 72 h later and used to infect fresh cells at a dilution of 1:10,000. This regimen was repeated six times[28], and intracellular RNA was harvested, reverse transcribed, and Sanger sequenced. Input corresponds to 16 h

post-viral infection and represents the starting point. From passage 2, the African-like and uORF1-KO mutant viruses began to dominate the cultures (Fig. 4D), indicating that both mutant viruses can outcompete American WT. In contrast, the proportion of uORF2-PTC1 viral RNA was

relatively unchanged throughout the course of the experiment (Fig. 4D, right panel), indicating no competition with the American WT virus. The phenotypic differences amongst viruses were also evident, albeit subtler, in Vero-infected cells (Supplementary Fig. 11C, D).

In an effort to detect the expression of uORF products in infected cells, antibodies to expressed proteins and peptides were prepared, but they lacked sensitivity and specificity. Therefore, we returned to ribosome profiling to examine translation of the uORFs in ZIKV mutant viruses (Fig. 4E and Supplementary Fig. 11E, F). As seen in Fig. 4E, reads corresponding to the translation of uORF1 (−2 frame) and uORF2 (−1 frame) were visible in the American WT virus, similar to PE243 (Fig. 2A, B). The uORF1-KO mutant, as expected, blocked translation of uORF1, with no reads observed at the initiation codon (blue asterisk); and the African-like virus with fused uORFs recapitulated the pattern of uORF translation of the African virus (as in Fig. 2C). These observations were further confirmed in phasing bar charts, with the highest proportion of reads in the predicted reading frames (Fig. 4E, right panels). Unexpectedly, reads corresponding to the uORF2 phase (Fig. 4E, red rectangle) were present after the premature termination codon in uORF2-PTC1 (Fig. 4E, red asterisk), supported by the high proportion of reads corresponding to the −1 phase in phasing plots (Fig. 4E, right panels). A potential explanation for these reads is −1 ribosomal frameshifting into uORF2 from ribosomes initiating at the main ORF start codon (Supplementary Fig. 12), and supportive evidence for this hypothesis is presented in Supplementary Note 1 section.

### ZIKV uORF1- and African uORF-encoded proteins interact with intermediate filaments
Bioinformatic analyses suggest no homology of ZIKV uORFs to known proteins. Given their predicted sizes (uORF1, 3.1 kDa; uORF2, 8.4 kDa; and African uORF, 10.3 kDa), however, it is likely that some, or all, will have intrinsic biological activity. To examine their cellular localisation, we transiently expressed mCherry-tagged and TAP (Strep-Strep-FLAG)-/FLAG-tagged uORF variants in uninfected and infected (ZIKV American isolate PE243 and ZIKV African isolate Dak84) mammalian cells and performed subcellular fractionation (Fig. 5A and Supplementary Fig. 13). As observed in Supplementary Fig. 13C, D, uORF2- and African uORF-encoded proteins were mainly present in the cytoplasm, and there was no relocalisation upon infection. However, uORF1, tagged with mCherry or TAP, and in two different cell lines (Vero and U251 cells), appeared not only in the cytoplasm but also in the cytoskeletal fraction (Fig. 5A and Supplementary Fig. 13E–H). Note that uORF1-TAP also appeared in the nuclear fraction, but this was probably due to passive diffusion of this small protein through the nuclear membrane. To discern the specific cytoskeletal target of uORF1, we performed confocal analysis with different proteins marking the three types of cytoskeletal filaments: actin for microfilaments, tubulin for microtubules and vimentin for intermediate filaments (Fig. 5B). Whereas the fluorescence intensity profiles of actin and tubulin with uORF1-TAP did not merge (Fig. 5B, right panel), it was found that uORF1-TAP could form denser granular structures that co-aligned with an abnormal 'collapsed' vimentin (Fig. 5B), and this was not observed in mock-transfected cells. Subsequently, we investigated how uORF1 could trigger vimentin rearrangement. AlphaFold2 analysis[29] suggested that the uORF1 peptide adopts an alpha-helical conformation with high confidence (Fig. 5C). To test whether this secondary structure was involved in the collapse of vimentin, we created a mutant version of uORF1-TAP with a helix-destabilising proline residue inserted into the middle of the α-helix (I14P, uORF1-TAP-1X Pro). As observed in Fig. 5D, uORF1-TAP-1X Pro was unable to form granular structures in the cytoplasm in comparison with the unmodified uORF1-TAP (Fig. 5B). In addition, the fluorescence intensity profiles of vimentin and this mutant uORF1 did not co-align (Fig. 5D, bottom panel). This suggests that the helical structure of the uORF1-encoded peptide might be responsible for the collapse of vimentin in transfected cells.

We went on to investigate whether the African uORF, which comprises the American uORF1 and uORF2, was also able to interact with intermediate filaments. Based on AlphaFold2 (Supplementary Fig. 14A), the uORF2-encoded peptide is likely to be intrinsically disordered, but in the African uORF, the N terminus was predicted, albeit with low confidence, to have an α-helical structure similar to uORF1. Plasmids encoding these proteins were transfected, and confocal analysis with different proteins marking cytoskeletal filaments was performed as described before. As observed in Supplementary Fig. 14B, the uORF2-encoded peptide localised in the cytoplasm and did not colocalise with any cytoskeletal marker, whereas the African uORF-encoded protein (Supplementary Fig. 14C) had a more granular perinuclear pattern and partially co-aligned with vimentin, similarly to uORF1 peptide, although in this case, intermediate filaments did not collapse.

### ZIKV uORF1-encoded protein helps in the formation of the cytoskeletal cage during infection
ZIKV infection reorganises microtubules and intermediate filaments to form a cytoskeletal cage surrounding viral factories[30,31]. These factories are the sites of viral RNA replication and virion assembly, and cytoskeletal remodelling might partly contribute to localising these processes in a closer environment for high viral replication efficiency[30]. To test specificity, the cytoskeletal phenotype of the uORF1-KO and African-like mutant viruses were compared to the American WT.

Previously synchronised Vero cells in the G0/G1 phase were infected (at MOI:3) with either virus, the cells fixed at 18 and 24 h p.i. and stained for viral E protein and vimentin (Fig. 6A). At 18 h p.i., vimentin was perinuclear in all cells and radiated toward the cell periphery with a filamentous structure, although a denser arrangement was observed in infected cells (Fig. 6A, upper panels). At 24 h p.i., the cytoskeletal cage, as defined by the collapse of vimentin, was visible in American WT infected cells (Fig. 6A, lower panels), but vimentin mostly remained in a perinuclear distribution in uORF1-KO and African-like infected cells. The compact aggregation of vimentin together with viral protein near the nucleus did not have a notable effect on cell size or overall morphology (Fig. 6A, brightfield panels). The fluorescence intensity profiles (Fig. 6A, right panels) indicate an accumulation of vimentin on one side of the nucleus in infected cells, which is more remarkable at 24 h p.i. An almost perfect co-alignment between the E protein and vimentin was observed in American WT infected cells at this time point. However, this was not the case with the African-like and the uORF1-KO mutant viruses. We quantified the collapse of vimentin in Vero-infected cells by measuring the area occupied by vimentin in relation to the total cell area at an early (18 h p.i.) and a later (24 h p.i.) time point, as previously published[30]. Thirty cells per condition were quantified in three different experiments (Fig. 6B). The area occupied by vimentin was reduced in American WT infected cells from ~68% to ~34%, from ~68% to ~46% in African-infected cells, and by only a small amount with the uORF1-KO mutant virus (from ~71% to ~66%). A similar pattern was observed in U251-infected cells (Fig. 6C and Supplementary Fig. 15). Due to slower viral replication in these cells, vimentin was quantified at later time points (20 and 28 h p.i.). These data indicate that the expression of the uORF1 polypeptide likely helps in the collapse of vimentin and thus in the formation of the cytoskeletal cage in infected cells. We can also conclude that the African-like 5′ UTR arrangement is less efficient than the American WT at promoting the cytoskeletal cage, probably due to its inability to collapse vimentin as described above.

### ZIKV uORFs are involved in the infection of human cortical neurons and cerebral organoids
To assess whether expression of the different uORFs might influence the capacity of ZIKV to infect neural cells, 2D human induced pluripotent stem cell (hiPSC)-derived cortical neuronal mono-cultures and

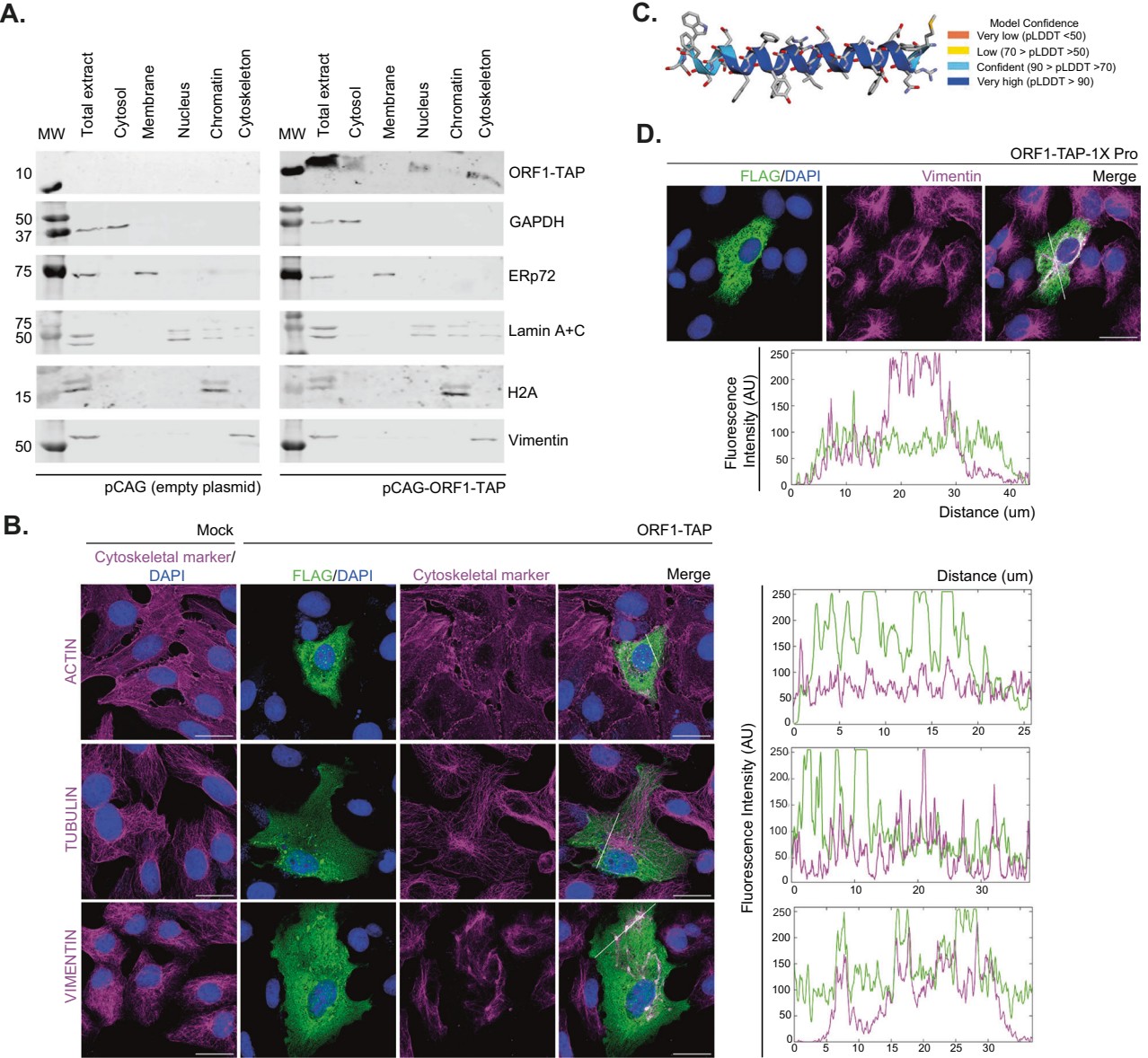

**Fig. 5 | ZIKV uORF1 interacts with the cytoskeleton. A** Subcellular fractionation analysis by western blot of Vero cells transfected for 48 h with plasmid pCAG (left panel) or pCAG expressing the ZIKV uORF1 protein fused with TAP-tag at the C-terminus (pCAG-ORF1-TAP, right panel). The total extract, cytosolic, membrane, nuclear, chromatin and cytoskeletal fractions were probed with antibodies against FLAG (to detect TAP-tag); GAPDH (cytosolic marker); ERp72 (membrane marker); Lamin A + C (nuclear marker); H2A (chromatin marker); and vimentin (cytoskeletal marker). Molecular masses (kDa) are indicated on the left. **B** Representative confocal images of Vero cells transfected for 36 h with pCAG (mock) or the pCAG-ORF1-TAP plasmids. Cells were stained with antibodies against FLAG (green) and different cytoskeletal proteins (i.e., actin, tubulin and vimentin in magenta). Nuclei were counter-stained with DAPI (blue). Images are a maximum projection of a Z-stack. Scale bars, 25 µm. Fluorescence intensity profiles of ORF1-TAP (green) and the different cytoskeletal markers (magenta) were obtained using ImageJ software, along the white straight line shown on the merged image crossing the representative cell. **C** AlphaFold2 prediction for ZIKV uORF1-encoded protein. **D** Representative confocal images of Vero cells transfected for 36 h with pCAG-ORF1-TAP-1X Pro. Cells were stained and analysed as described in (**B**). Experiments in (**A**, **B** and **D**) were repeated three times with similar results. Source data are provided as a Source Data file.

---

human 3D cerebral organoid slice cultures were infected with either American WT, uORF1-KO or the African-like ZIKV. Differentiated hiPSC-derived glutamatergic cortical neurons (CNs) (i³Neurons[32]) were infected at high MOI (10 PFU/cell based on titre in Vero cells) for 4 days. Released virions in the supernatant were quantified and from 48 h p.i. onwards, mutant virus titres were significantly lower, decreasing by 1.5- to 2-log10 at 96 h p.i. (Fig. 7A), in sharp contrast to their increased replication in U251 cells (Fig. 4C). This was further confirmed by immunolabelling and quantification of ZIKV E protein (Fig. 7B and Supplementary Fig. 16A, B), where spread of infection was less obvious with the African-like and the uORF1-KO mutant viruses, only limited to single infected cells, suggesting little to no viral spread.

Next, we used cerebral organoids to investigate infection in a human brain-like 3D tissue environment and to explore whether the African uORF or uORF1 are also required for the infectivity of other cell types. To do so, we infected cerebral organoid slices grown at the air–liquid interface (ALI-COs), which recapitulate cortical cell type-diversity, layering and neurodevelopmental milestones[33,34]. After 82 days in vitro (DIV), reflecting the first trimester of gestation, ALI-COs were infected with the American WT, the uORF1-KO and the African-like viruses at MOI 5. Virus inoculum was removed after 24 h and the ALI-COs were grown for a further 7 days before being fixed. Six micrometres of tissue sections from six different Z-axis sections in ALI-COs were stained for viral E protein and positive cells were quantified in relation to the total

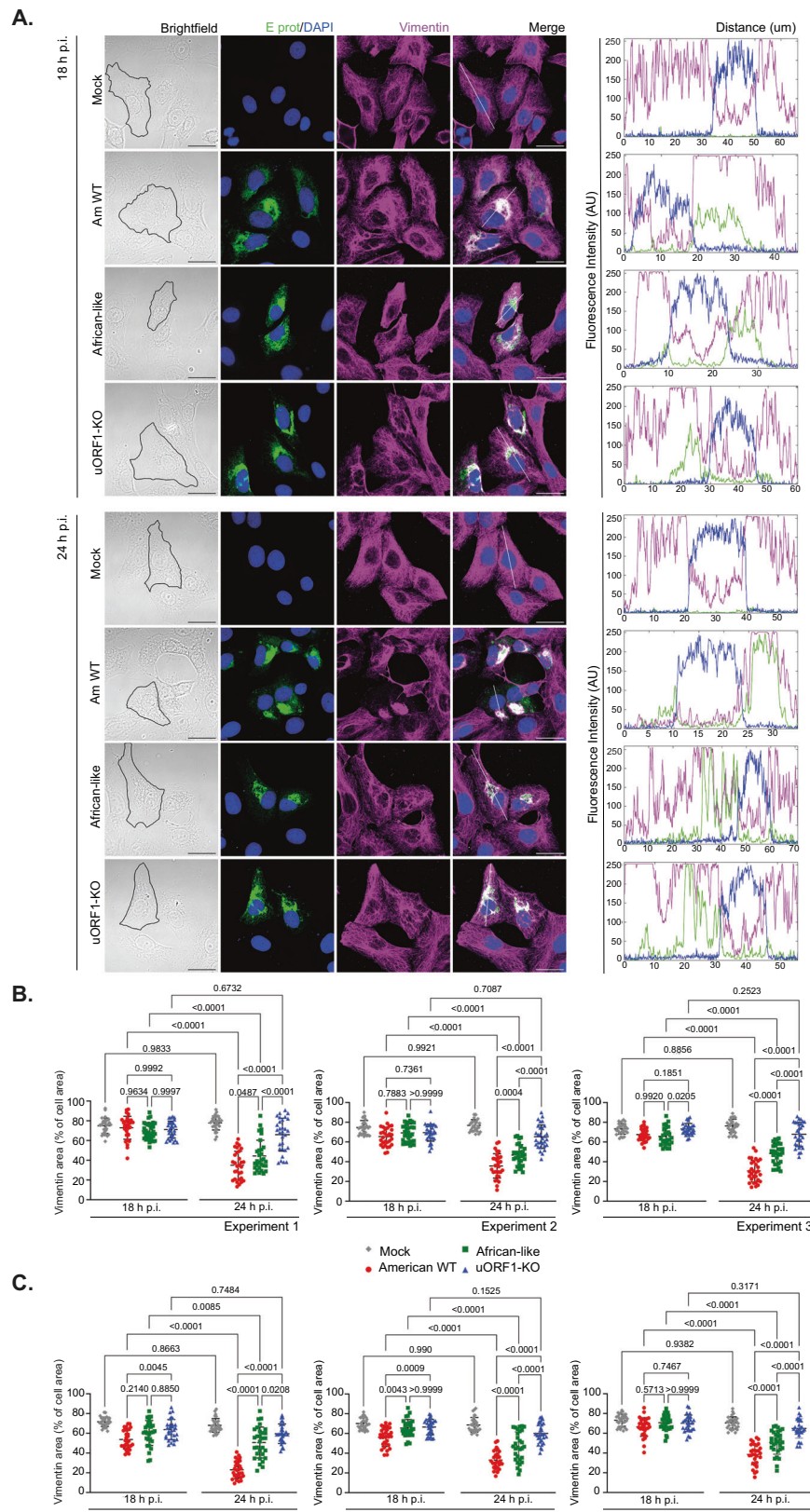

number of nuclei as previously described for cerebral organoids infected with SARS-CoV-2[35]. Approximately, 47% of cells were positive for the E protein in the American WT virus infection experiment. This was significantly reduced in the African-like and uORF1-KO virus infection experiments, to ~33% and ~11%, respectively (Fig. 7C and Supplementary Table 2). This was further confirmed by IF (Fig. 7D–F

and Supplementary Fig. 16), corroborating our findings on the involvement of the ZIKV 5′ UTR in the infection of human brain cells.

To examine the differential neural cell tropism of these mutant viruses, cryostat sections of ALI-COs were immunostained for a panel of cellular markers, including nestin (as a marker for neural progenitor cells, NPCs), GFAP (a marker for the astroglial lineage such as radial

**Fig. 6 | ZIKV uORF1-encoded protein helps in the formation of the cytoskeletal cage during infection. A** Representative brightfield and confocal images of Vero cells infected with the American WT (Am WT), the African-like and the uORF1-KO mutant viruses (MOI:3) for 18 h (upper panel) and 24 h (lower panel). Cells were stained with antibodies against the viral E protein (green) and vimentin (magenta). Nuclei were counter-stained with DAPI (blue). Images are a maximum projection of a Z-stack. Scale bars, 25 μm. Fluorescence intensity profiles (right panel) of E protein (green), vimentin (magenta) and the nuclear staining (blue) were obtained using ImageJ software, along the white straight line shown on the merged image crossing the representative cell. **B** Quantification of the proportion of vimentin area by fluorescence microscopy versus overall cell area, measured by brightfield microscopy, in Vero-infected cells as shown in (**A**), and U251-infected cells (**C**) as shown in Supplementary Fig. 15. Each point represents a single cell ($n = 30$ per condition and biological replicate). Data are represented as mean ± SD from three biologically independent experiments. Statistical analysis was repeated measures two-way ANOVA. All $p$ values are from comparisons of the different viruses at that specific time point and from each virus or mock at the two different time points. Source data are provided as a Source Data file.

glial cells, glial progenitors and astrocytes), and MAP2 (for mature neurons). As shown in Fig. 7D–G, ZIKV viruses preferentially infected nestin- or GFAP-positive cells, whereas very few MAP2-positive cells were also positive for E protein. A similar infectivity pattern was seen with all viruses tested (~36% nestin+, ~30% GFAP+ and ~4% MAP2+, Supplementary Table 3), suggesting that the modifications to the ZIKV 5′ UTR do not affect neural cell tropism.

Together, these data indicate that the expression of ZIKV uORF1 promotes infection of NPCs and precursors of the astroglial lineage in cerebral organoids. These results also demonstrate that the African 5′ UTR arrangement impairs ZIKV infection of the developing brain, although to a lesser extent.

## No detectable effect of ZIKV uORFs in the mosquito vector

ZIKV typically cycles between humans and *Aedes* mosquitoes; hence we also tested whether the uORFs were expressed during the infection of cells derived from the mosquito vector. Ribo-Seq analysis of *Aedes albopictus* (*A. albopictus*) (C6/36) cells infected with the ZIKV American isolate PE243 for 24 h suggested that both uORFs were occupied by ribosomes during infection (Fig. 8A). Then, we compared the transmission dynamics of American WT, African-like and uORF1-KO viruses, derived from infectious clones, in female *Aedes aegypti* (*Ae. aegypti*). Mosquitoes were exposed to an artificial, infectious blood meal containing an expected virus titre of $2 \times 10^6$ PFU/mL of blood. Actual titres ranged from $0.8 \times 10^6$ to $3.6 \times 10^6$ PFU/mL across the different experiments, and this uncontrolled variation was accounted for in the statistical analysis. At several defined time points after the infectious blood meal, we determined the rates of mosquito infection and systemic viral dissemination by RT-PCR and detected the presence of ZIKV in saliva by infectious assay. We calculated infection prevalence as the proportion of blood-fed mosquitoes with a body infection (determined by RT-PCR, Fig. 8B, left panel), dissemination prevalence as the proportion of infected mosquitoes with a virus-positive head (determined by RT-PCR, Fig. 8B, middle panel) and transmission prevalence as the proportion of mosquitoes with a disseminated infection that had infectious saliva (determined by focus-forming assay, Fig. 8B, right panel), as previously described[11,36]. A total of 99, 128 and 234 individual mosquitoes were analysed in the first, second and third experiments, respectively. In each experiment, mosquitoes were collected at the same time points for all groups. In the first experiment, mosquitoes were collected on days 7, 14 and 21 post-exposure. In the two following experiments, the collection time points were changed to days 7, 10, 14 and 17 to gather more meaningful data based on the results of the first experiment. The subsequent analysis combined the three experiments in a well-established statistical framework (logistic regression) that accounts for differences in sample size and does not require all time points to be present in all experiments (i.e., a full-factorial design). The number of mosquitoes for each time point and each experiment is provided in Supplementary Table 4.

Although the vast majority of mosquitoes became infected irrespective of the experiment, virus type and time point (mean: 90.5%; median: 90.9%; range: 66.7%–100%), none of these variables had a statistically significant effect on infection prevalence (Supplementary Table 5). Both dissemination and transmission prevalence significantly increased over time (Fig. 8B), but there was no detectable difference between the different ZIKV mutant viruses (Supplementary Table 5). We conclude that the ZIKV mutant viruses have similar transmission dynamics in mosquitoes regardless of their 5′-UTR arrangement.

## Discussion

In this study, we provide the first high-resolution maps of ZIKV translation in both mammalian and mosquito cells and describe two previously overlooked uORFs (uORF1 and uORF2) in Asian/American ZIKV strains, which exist as a single uORF in African isolates. By generating mutant viruses, we show that the uORFs can modulate virus growth in human brain cells. However, the different ZIKV uORFs mutant viruses had similar infection, dissemination and transmission dynamics in *Aedes* mosquitoes, regardless of their 5′-UTR arrangement, suggesting they may function uniquely in mammalian hosts.

How the uORFs exert their effects is uncertain but is likely through modulation of expression of the main polyprotein ORF and via properties of their encoded proteins. The translational activity of the ZIKV uORFs is noticeably lower than that of the main polyprotein ORF, consistent with their initiation at non-AUG codons. Whilst the effect of flanking bases on initiator AUG codons is well-established (Kozak context), it is less well understood for non-AUG codons, making the strength of non-AUG initiation codons harder to predict[37]. Based on our present understanding, the ZIKV uORFs rank as modest start codons[37,38]. However, their utilisation may also be modulated by RNA structure and, potentially, by virus and host proteins. Recent toe-printing analysis of ZIKV initiation in vitro has also detected the utilisation of non-AUG initiation sites at nucleotide positions $UUG_{41}$, $UUG_{80}$ and $UUG_{86}$ (c.f. $CUG_{25}$ and $UUG_{80}$ here). The magnitude of the toeprints observed was influenced by the availability of particular translation factors in the reconstituted assay system and the RNA circularisation status of the mRNAs used[27]. The absence of a toeprint at $CUG_{25}$ in the latter study might reflect the different experimental milieu (in vitro reconstitution versus infected cells).

In luciferase reporter gene assays, expression of uORF1 had an inhibitory effect on main ORF translation, whereas expression of uORF2 or the African uORF enhanced translation of the reporter gene. This might explain why mutant viruses bearing these changes (uORF1-KO and African-like) in the 5′ UTR grew to higher titres and were fitter, in comparison to American WT, in tissue culture cells. The inhibitory effect of uORF1 expression might be related to the fact that the uORF1 stop codon (UGA) overlaps with the initiation codon of the main ORF (AUGA). We speculate that ribosomes terminating at the stop codon may decrease access of ribosomes to the main ORF AUG, which would be obviated in uORF1-KO, as fewer ribosomes are delivered to the stop codon. In a similar vein, ribosomes translating uORF2 or the African uORF would unwind SLB without terminating within it, increasing access of ribosomes to the main ORF AUG. Viruses with mutations intended to increase (uORF2-AUG) or decrease (uORF2-KO) translation of uORF2 were non-viable or reverted to the WT sequence. The dramatic effect on viability of replacing the uORF2 UUG start codon with AUG in the virus context might reflect upregulation of uORF2 expression reducing the number of scanning ribosomes reaching the main ORF AUG or could be a consequence of over-expression of the uORF2 protein product in infected cells. We also note from SHAPE analysis that this RNA adopts a slightly different fold

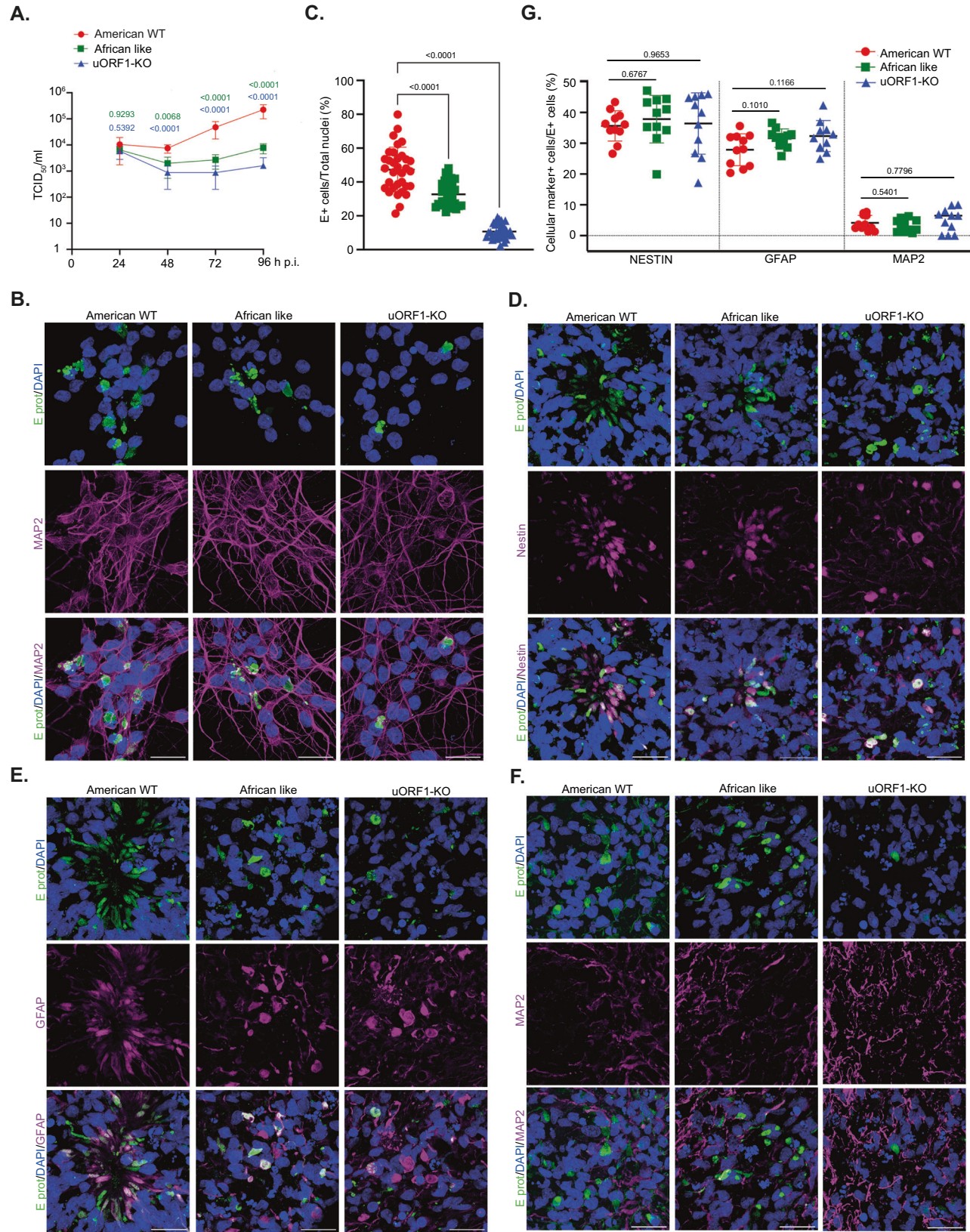

in the SLB region, which could also contribute to improper replication and translation. Further repeats of this experiment, including the use of alternative start codons, would be valuable to clarify these possibilities. The uORF2-KO virus revertant phenotype might be attributable to a reduced amount of uORF2 protein product in infected cells or altered translational dynamics in the complex 5′ UTR of ZIKV.

Our results, and those of others[39,40], reveal that eIF2 becomes phosphorylated on its α-subunit upon ZIKV infection (Supplementary Fig. 10B). eIF2α phosphorylation leads to a general shutdown of protein synthesis by inhibiting translation initiation, but in some cases, such as the translation of the activation transcription factor 4 (ATF4[41]), uORFs act as enhancers of main ORF translation as a response to

**Fig. 7 | ZIKV uORFs are involved in neural cell infection. A** Time-course of i³Neurons infected with ZIKV (MOI 10) for 96 h. TCID$_{50}$ were performed with serial dilutions of the supernatant to measure virion release. Values show the mean averages of the titration from five biological replicates. Error bars represent standard errors. Statistical analysis was repeated measures two-way ANOVA on the log-transformed data. All *p* values are from comparisons of the mutant virus with the American WT at that specific time point. **B** Representative confocal images of i³Neurons infected with the American WT, the African-like and the uORF1-KO viruses (MOI:10) for 96 h. Cells were stained with antibodies against the viral E protein (green) and the mature neuron marker MAP2 (magenta). Nuclei were counter-stained with DAPI (blue). Images represent the maximum projection of a Z-stack. Scale bars, 25 µm. **C** Percentage of E⁺ cells in relation to total number of nuclei in ALI-COs infected with the American WT, the African-like and the uORF1-KO viruses (MOI:5) for 7 days. Thirty-three images per virus type at 20× resolution

(~400–500 nuclei/image) were quantified for E-positive staining. Error bars represent standard errors. Statistical analysis was one-way ANOVA with Gaussian distribution and did not assume equal variance for the two populations being compared. All *p* values are from comparisons of the mutant virus with the American WT. Representative confocal images of infected ALI-COs, showing immunoreactivity for the viral E protein (green) and for different cellular markers (in magenta): nestin (**D**), GFAP (**E**) and MAP2 (**F**). Nuclei were counter-stained with DAPI (blue). Images represent the maximum projection of a Z-stack. Scale bars, 25 µm. **G** Percentage of E⁺ cells that are positive for nestin, GFAP and MAP2. Eleven images per virus type at 20× resolution (~400–500 nuclei/image) were quantified. Statistical analysis was performed as in (**C**). Four ALI-CO slices derived from two independent cerebral organoids were analysed for (**C**) and (**G**). The number of fluorescent cells corresponding to each staining was measured with ImageJ software by splitting the different channels.

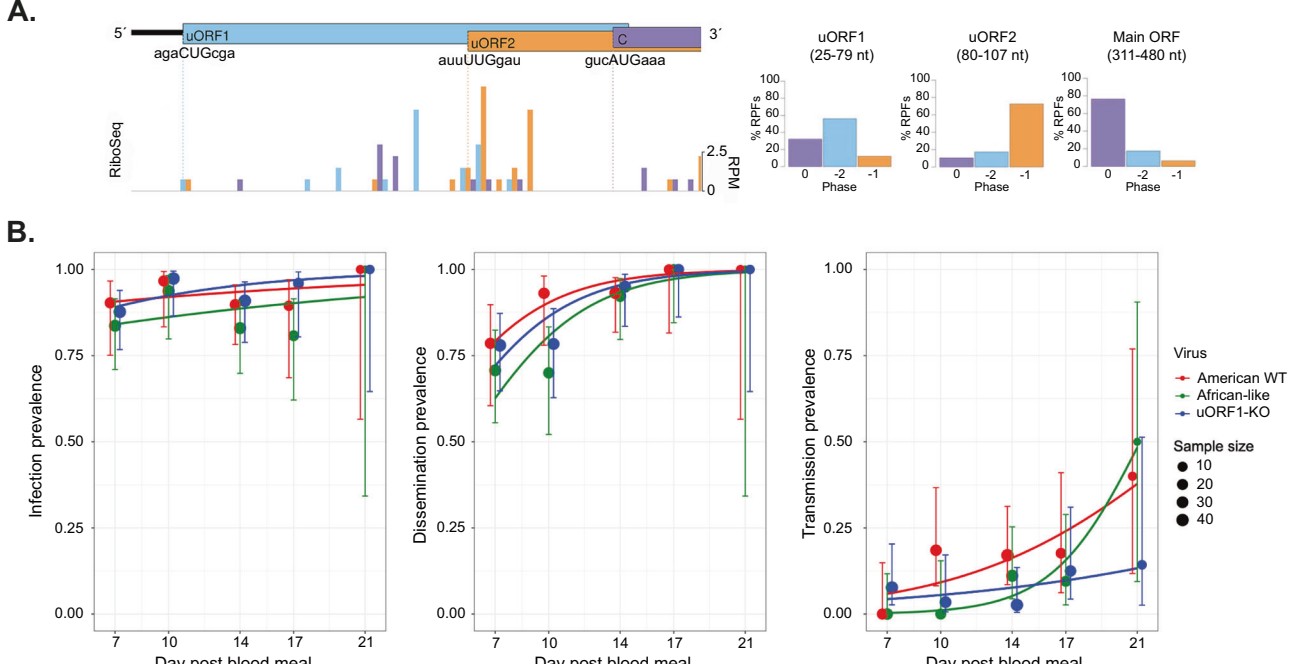

**Fig. 8 | ZIKV mutant viruses display similar transmission dynamics in mosquitoes in vivo. A** (left panel) Ribo-Seq reads in the 5′ region of the ZIKV genome at 24 h p.i. of CHX-treated C6/36 cells infected with PE243 (MOI:3). Histograms show the positions of the 5′ ends of reads with a +12 nt offset to map the approximate P-site as described in Fig. 2A–C. **A** (right panel) Bar charts of the percentage of RPFs translated in each frame in relation to the main ORF as described in Fig. 2D–F. **B** Prevalence of ZIKV infection (left), dissemination (middle) and transmission (right) over time in mosquitoes exposed to an infectious blood meal containing 2 × 10⁶ PFU/mL of virus. Infection prevalence is the proportion of blood-fed mosquitoes with a body infection (determined by RT-PCR), dissemination prevalence is

the proportion of infected mosquitoes with a virus-positive head (determined by RT-PCR), and transmission prevalence is the proportion of mosquitoes with a disseminated infection and infectious saliva (determined by focus-forming assay). The data represent three separate experiments combined, colour-coded by different ZIKV mutants (total number of *Ae. aegypti* mosquitoes per experiment and time points are indicated in Supplementary Table 4). The size of the data points is proportionate to the number of mosquitoes tested, and the lines are the logistic fits of the time effect for each type of virus (ignoring the experiment effect on transmission prevalence in the visual representation). The vertical error bars are the 95% confidence intervals of the proportions.

cellular stress. The finding that the ZIKV ORFs are relatively translationally enhanced during infection suggests that the ZIKV 5′ arrangement, and the likely interplay of the two uORFs and main ORF, may offer some resistance to the translation inhibition engendered by phosphorylated eIF2α. This is most evident with the African-like arrangement (Fig. 3G) and might explain why the African-like virus outgrows American WT in tissue culture (Fig. 4D).

Bioinformatics analyses do not reveal any apparent homology of ZIKV uORFs to known proteins, although AlphaFold2 analysis[29] suggests that uORF1 adopts an alpha-helical conformation (Fig. 5C), whereas uORF2 is unlikely to have a well-defined secondary structure (Supplementary Fig. 14A). However, in the African uORF, the N terminus was predicted with low confidence to have an α-helical structure similar to uORF1 (Supplementary Fig. 14A). Related uORFs can be

predicted in other neurotropic flaviviruses, such as Japanese encephalitis virus (JEV)[42] and WNV (Supplementary Fig. 17), and they are also expected to adopt alpha-helical conformations. The expression in *trans* of uORF2 resulted in a cytoplasmic localisation that was not changed upon infection. However, transfected uORF1 was predominantly cytoskeletal and led to the collapse of the vimentin network, with no change observed in microfilaments or microtubules. The African uORF-encoded protein partially co-aligned with vimentin, similarly to the uORF1 peptide, although in this case, intermediate filaments did not collapse.

Previous electron tomography experiments have revealed that ZIKV infection leads to a reorganisation of microtubules and intermediate filaments, forming a cytoskeletal cage that surrounds viral factories[30,31]. The remodelling of the cytoskeleton around these sites of

viral RNA replication and virion assembly confines these processes to a closer environment, increasing the local concentration of all necessary factors for high viral replication efficiency[30]. Our results reveal that the African-like mutant virus and, more dramatically, the uORF1-KO virus delay the formation of the cytoskeletal cage in infected cells. The consequences of this delay need further investigation and could potentially enhance innate immune recognition upon infection.

Vimentin is a type III intermediate filament and displays typical features of such proteins, with an N-terminal domain (-100 aa), a central α-helical rod (310 aa) and a C-terminus of variable length (15–1400 aa). The central rod domain allows the formation of coiled-coil dimers that can assemble into higher-order filaments. The bundling of peripheral vimentin into thicker assemblies in the perinuclear area of the cells, as observed upon uORF1 transfection, has been previously observed in cells[43]. It has been associated with post-translational modifications of vimentin, such as phosphorylation of the head domain[43,44] or deacetylation (by histone deacetylase 6) upon expression of certain oncogenes[45]. It is unknown whether vimentin is subjected to any post-translational modifications upon ZIKV infection but, based on the α-helical prediction of AlphaFold2, the uORF1 protein could potentially interact with the central rod domain of vimentin, intercalate during dimer formation, and antagonise correct polymerisation. The disruption of the helical structure of the uORF1-encoded peptide due to the insertion of a proline residue likely prevents the co-alignment of this protein with vimentin and its subsequent collapse (Fig. 5D). In addition, vimentin, in conjunction with nestin, are the first intermediate filaments expressed during early brain development[46]. Using ALI-COs, a highly accessible slice culture version of whole-cortical organoids for viral infections, with improved oxygenation, nutrition and longevity[33,34], we also observed that vimentin also collapsed in infected cells (Supplementary Fig. 18), similar to what was previously observed in Vero and U251 cells. The potential role of this phenomenon in virus replication and pathogenesis has not been studied yet during ZIKV infection, and a potential interaction of uORF1 and African uORF with the cytoskeleton in ALI-COs still needs to be assessed. Interestingly, during neural development, the cytoskeleton is involved in different processes (e.g., orientation of the mitotic spindle, assembly of primary cilia, etc.) that are fundamental for the correct generation and maturation of neurons[47]. The disruption of any of these processes impairs neurogenesis (differentiation of NPCs into mature neurons) in the developing brain, provoking neurological defects such as microcephaly, the key pathological aspect of ZIKV infection.

Furthermore, the use of ALI-COs has allowed optimal characterisation of the effects exerted by the different ZIKV 5′-UTR mutant viruses. Our finding that the uORF1 knockout virus has a defect in infecting ALI-COs suggests that the protein encoded by uORF1—only present in the Asian/American strains—is an essential neurotrophic factor required for ZIKV infection of neural cells in this developing brain model. Interestingly, a virus with the African uORF in the American virus background was also impaired, although to a lesser extent, in cerebral organoids and iPSC-derived CNs. This may explain, at least in part, how a mutation in the 5′ region in a rarely detected pathogen such as ZIKV drove a major epidemic in the Americas in 2015/2016. Until now, the only known associations with an increase in neuropathogenesis in epidemic Asian/American strains compared to pre-epidemic Asian strains (e.g., Cambodia 2010) were the S139N mutation in the membrane precursor (prM) protein[48] and the V473M mutation in the E protein[49]. The V473M substitution also enhanced viremia and intrauterine transmission in a mouse model[49]. However, it has been recently described that a lysine residue at position 21 in the prM protein in the African ZIKV lineage is associated with increased viral replication in neural cell lines and enhanced neurovirulence and neuroinvasiveness in suckling mice compared to Asian/American strains that have a glutamic acid in that position[50]. Not only are viral proteins associated with ZIKV neurotropism, but the 3′ UTR of the viral genome promotes viral replication in undifferentiated neural stem cells and NPCs by interacting with the host RNA-binding protein Musashi homologue 1 (MSI1)[51] in a bipartite fashion[52]. MSI1 is highly expressed in NPCs and, when depleted, is associated with microcephaly and other neurodevelopmental disorders[53]. This suggests that MSI1 could be a contributor to ZIKV foetal neurotropism and neuropathogenesis[51].

Our discovery of previously overlooked multiple uORFs in the 5′ UTR of different ZIKV strains with a role in virus replication uncovers the necessity to investigate further and characterise the role of these 5′ uORFs in the neurovirulence and neuropathogenesis of other flaviviruses. To date, they have only been described in neurotropic flaviviruses (ZIKV, JEV[42] and WNV, Supplementary Fig. 17) and they might account for the divergence in tropism between neurotropic and viscerotropic flaviviruses such as dengue or yellow fever viruses.

Lastly, the authors would like to indicate that the current classification of ZIKV into two different lineages, African and Asian/American, is probably inadequate and needs updating as it is not only stigmatising but also might not reflect the genetic diversity that the virus accumulated after the 2015/2016 epidemic. A better option will be to use the new terminology proposed by Seabra et al.[54], based on phylogenetic analyses, clustering techniques within- and between-group pairwise genetic distances, and evolutionary analyses to define genetic groups and subgroups. Their proposed nomenclature avoids geographical terminology using alpha-numerical labels instead. The formerly named African and Asian lineages will be now called ZA and ZB genotypes, respectively. For instance, the African isolate Dak84 will now belong to the ZA group and the American isolate PE243 will now be in the ZB.2.0 group, which includes the Polynesian sequences and the basal American lineage.

## Methods
### Cells and viruses
Vero (ATCC, CCL81) and U251 (ATCC, CRL-1620) cells were maintained in Dulbecco's modification of Eagle's medium (DMEM) supplemented with 5% and 10% (vol/vol) foetal calf serum (FCS), respectively. Culture medium also contained 100 U/mL penicillin, 100 μg/mL streptomycin and 2 mM L-glutamine. Cells were incubated at 37 °C in the presence of 5% $CO_2$. *A. albopictus* C6/36 cells (ATCC, CLR 1660) were maintained in L-15 medium supplemented with 10% FCS at 28 °C without $CO_2$. The American strain ZIKV/Brazil/PE243/2015 (Genbank accession number KX197192.1) was kindly provided by Prof. Alain Kohl (Liverpool School of Tropical Medicine, Liverpool, UK), and the African strain ZIKV/*A.taylori*-tc/SEN/1984/41662-DAK (Genbank accession number KU955592.1) was obtained from the European Virus Archive. ZIKV mutant viruses were rescued by electroporation of Vero cells with SP6-transcribed RNAs of pCC1-SP6-ZIKV derivatives[25]. Virus stocks were amplified in Vero cells and concentrated by ultracentrifugation (110,500 × *g* for 3 h at 4 °C). All mutant viruses were passaged five times at a low MOI (0.01), and viral RNA was subjected to RT-PCR analysis to confirm the presence or reversion of the introduced mutations.

Upon reaching 70–80% confluence, Vero and U251 cells were infected with parental or mutant viruses in serum-free DMEM supplemented with 100 U/mL penicillin, 100 μg/mL streptomycin and 2 mM L-glutamine at the indicated MOI. After adsorption for 1 h at 37 °C, the inoculum was removed and replaced with DMEM containing 2% FCS and 20 mM HEPES pH 7.4. In cases where the infection was preceded by transfection, the transfection medium was washed three times with PBS prior to infection. C6/36 cells were infected as above but using L-15 medium. Mosquito cells were incubated at 28 °C.

### Plasmids
A pCC1BAC vector containing the ORF of the ZIKV BeH819015 isolate flanked by the 5′ and 3′ UTRs for ZIKV PE243 (pCC1-SP6-Am ZIKV)[25] was kindly provided by Prof. Andres Merits (University of Tartu, Tartu, Estonia). Mutations in the 5′ UTR were generated by site-directed

mutagenesis with the indicated oligonucleotides (Supplementary Table 7) in a subcloned gene fragment of the first 2000 nucleotides of ZIKV PE243 maintained in pcDNA.3. Following sequencing to confirm no inadvertent changes had arisen, mutated fragments were cloned back into pCC1-SP6-Am ZIKV at the *EcoR1* and *AvrII* sites.

The previously described pSGDLuc vector[55] was used as a template to design the Renilla and the FF-Luc reporters (pREN-Luc and pFF-Luc). The pREN-Luc cassette was amplified using the following oligonucleotides 5′- TCCGCCCAGTTCCGCCCATTCTCCGC and 5′-GCGCTCTAGATTATCTCGAGGTGTAGAAATAC and cloned back into pSGDLuc previously digested with *AvrII* and *XbaI*. This created a plasmid expressing only the Ren-Luc gene under the T7 promoter. For the FF-Luc reporters, the first 194 nucleotides of the ZIKV genome were cloned in the pSGD plasmid previously digested with *XhoI* and *BglII* using a different set of oligonucleotides (Supplementary Table 7) to introduce the uORF1, uORF2 and main ORF in frame with the 2A-FF-Luc gene. In addition, a T7 promoter was introduced upstream of the ZIKV coding region. Mutations within the uORFs were introduced by site-directed mutagenesis (Supplementary Table 7).

The uORF1, uORF2 and African uORF coding sequences were amplified using specific oligonucleotides (Supplementary Table 7) and cloned in frame at the C-terminus with a FLAG-, TAP- or mCherry tag. *PacI*- and *AflII*-digested PCR fragments were ligated into pCAG previously digested with these restriction enzymes. The I14P mutation in pCAG-uORF1-TAP-1X Pro was introduced by site-directed mutagenesis (Supplementary Table 7). All sequences were confirmed by Sanger sequencing.

### RNA transcript preparation and luciferase reporter assays

Transcription reactions were performed using the mMESSAGE mMACHINE™ T7 Transcription Kit (Thermo Fisher Scientific) for 2 h at 37 °C and terminated by treatment with TURBO™ DNase (Thermo Fisher Scientific) for 30 min at 37 °C. Prior to transfection, RNA transcripts were polyadenylated for 30 min at 37 °C. Vero and U251 cells were transfected in triplicate with Lipofectamine 2000 reagent (Invitrogen) by RNA reverse-transfection, where cells in suspension were added directly to the RNA complexes in 96-well plates. For each transfection, 100 ng of purified T7 ZIKV-uORFs FF-Luc RNA and 5 ng of purified T7 Ren-Luc RNA plus 0.3 μl Lipofectamine 2000 in 20 μl Opti-MEM (Gibco-BRL) supplemented with RNaseOUT (Invitrogen; diluted 1:1000 in Opti-MEM) was added to each well containing $10^5$ cells. RNA to be transfected was purified using an RNA Clean and Concentrator kit (Zymo Research). Transfected cells in DMEM supplemented with 2% FCS and 20 mM HEPES pH 7.4 were incubated at 37 °C for the indicated times. At 6 h p.t., cells were infected with American isolate PE243 and African isolate Dak84 at MOI:3 by adding 50 μl of DMEM-2% FCS medium-containing virions. Firefly and Ren-Luc activities were determined using the Dual-Luciferase Stop & Glo Reporter Assay System (Promega). Translation efficiency was calculated as the ratio of FF-Luc units (uORFs or main ORF translation) to Ren-Luc units (transfection control), normalised by the ratio of the American main ORF WT translation.

### DNA transfection

Vero and U251 cells were transiently transfected with pCAG-uORF plasmids using a commercial liposome method (TransIT-LT1, Mirus). Transfection mixtures containing plasmid DNA, serum-free medium (Opti-MEM; Gibco-BRL) and liposomes were set up as recommended by the manufacturer and added dropwise to the tissue culture growth medium.

### Plaque assays

Viral titres were determined by plaque assay. Viral supernatant was 10-fold serially diluted in serum-free DMEM. Vero cells in 6-well plates were infected with 400 μL of diluted supernatant. After 1 h at 37 °C with regular rocking, the inoculum was removed and replaced with a 1:1 mixture of 3% low-melting agarose and MEM 2X (20% 10X MEM, 8 mM L-glutamine, 0.45% Na$_2$CO$_3$, 0.4% non-essential amino acids, 4% HEPES pH 7.4 and 4% FCS). Plates were incubated at 37 °C for 5 days prior to fixing with formal saline (4% formaldehyde and 0.9% NaCl) at room temperature. Cell monolayers were stained with 0.1% toluidine blue to visualise plaques. The number of plaque-forming units per mL (PFU/mL) was calculated as n° plaques * 2.5. The final number of PFU/mL per time point was calculated by averaging the results of three biological repeats.

### TCID$_{50}$ assays

ZIKV replication was assessed using a 50% tissue culture infective dose (TCID$_{50}$) assay in Vero cells. Supernatant derived from infected i³Neurons cells was subjected to 10-fold serial dilutions. Cells were fixed and stained at 96 h p.i. as indicated above. Wells showing any cytopathic effect were scored positive. TCID$_{50}$ values were calculated using the Reed–Muench calculator[56].

### Virus competition assays

Dual infection/competition assays were performed in duplicate on Vero and U251 cells using the American WT and different mutant viruses at an equal or 9:1 ratio at a total MOI of 0.1. Media collected from infected plates was used for six blind passages using 1:10,000 volume of obtained virus stock (corresponding to MOI 0.05–0.2). Total RNA from passages 0 (16 h p.i. to determine viral input), 2, 4 and 6 was isolated[18]. cDNA was synthesised from 1000 ng total RNA using M-MLV Reverse Transcriptase (Promega) and 1 μL random primers (1:4 dilution of random primers, Invitrogen™). The ZIKV 5′ UTR region was amplified using specific primers (Supplementary Table 7). Following PCR reactions, the resulting amplicons were subjected to electrophoresis in 3% agarose gels, and DNA bands were purified using the Zymoclean Gel DNA recovery kit (Zymo Research) and subjected to Sanger sequencing. Quantification was performed by integrating area under the curve for each nucleotide and expressing it as percentage of the total signal at that base.

### Subcellular fractionation

Vero or U251 cells seeded in 4-cm dishes were transfected with plasmids of interest. At 40 h p.t., samples were collected and immediately subjected to subcellular fractionation using the 'Subcellular Protein Fractionation Kit for Cultured Cells' (Thermo Fisher Scientific). Purified fractions were stored in Laemmli's buffer at −20 °C for downstream biochemical characterisation.

### Immunoblotting

Cells were lysed in 1X Laemmli's sample buffer. After denaturation at 98 °C for 5 min, proteins were separated by 12% SDS–PAGE and transferred to a 0.45 μm nitrocellulose membrane. For proteins below 16 kDa, 20% SDS–PAGE and 0.22 μm nitrocellulose membrane were used instead. Membranes were blocked (5% non-fat milk powder or bovine serum albumin (BSA) in PBST [137 mM NaCl, 2.7 mM KCl, 10 mM Na$_2$HPO$_4$, 1.5 mM KH$_2$PO$_4$, pH 6.7, and 0.1% Tween 20]) for 30 min at room temperature and probed with specific primary antibodies (Supplementary Table 8) at 4 °C overnight. Membranes were incubated in the dark with IRDye-conjugated secondary antibodies diluted to the recommended concentrations in PBST for 1 h at room temperature. Blots were scanned using an Odyssey Infrared Imaging System (Licor).

### Selective 2′ hydroxyl acylation analysis by primer extension (SHAPE)

SHAPE reactions were conducted as follows[57]: 2 pmol of purified in vitro transcribed RNA template (full-length WT ZIKV RNA and mutant derivatives) was denatured at 95 °C for 3 min, then incubated

on ice for 3 min before the addition of folding buffer (final concentration 100 mM HEPES pH 8, 6 mM MgCl₂, 100 mM NaCl). Refolding was conducted at 37 °C for 30 min. Samples were divided into two equal volumes and treated with either *N*-methylisatoic anhydride (NMIA) dissolved in DMSO (10 mM final concentration) or DMSO only, for 45 min at 37 °C. Samples were ethanol precipitated and resuspended in 10 μL ddH₂0 before being reverse transcribed. NMIA- or DMSO-treated RNA samples were mixed with 5′ NED labelled primer (final concentration 1 μM) (5′ CTTCCTAGCATTGATTATTCTCAGCATG 3′, Applied Biosystems). Annealing was conducted at 85 °C for 3 min, 60 °C for 10 min, and 35 °C for 10 min. Reverse transcription with Superscript III (Invitrogen) was conducted according to the manufacturer's instructions at 53 °C for 40 min. Sequencing ladders were prepared by reverse transcription of 1 pmol of non-treated RNA in the presence of ddCTP, using 5′ VIC labelled primer. All primer extensions were terminated by alkaline hydrolysis, neutralised with HCl and ethanol precipitated before being resuspended in ddH₂0. The sequencing ladders were divided equally between cDNAs from NMIA- and DMSO-treated RNA. The final samples were dried onto sequencing plates before being resuspended in formamide and fractionated by capillary electrophoresis (Applied Biosystems 3130xl analyser, University of Cambridge, Department of Biochemistry DNA Sequencing Facility). SHAPE was performed in triplicate for each genome from separate RNA preparations. Each SHAPE dataset was analysed with the QuSHAPE software[58]. The average normalised NMIA reactivity was calculated for each individual nucleotide and used to model RNA secondary structure using the RNA structure software[59] using the SHAPE reactivity profile as soft pseudo-free energy constraints. RNA secondary structures were drawn using XRNA.

### Immunofluorescence microscopy
Vero or U251 cells were seeded in a 24-well plate with coverslips to be transfected and/or infected. At the indicated times, cells were fixed with 50:50 methanol: acetone for 10 min. Coverslips were blocked for 30 min in 5% BSA in 1X PBS at room temperature. Coverslips were incubated in 1% BSA with specific primary antibodies (Supplementary Table 8). Secondary antibodies were diluted in 1X PBS and applied for 1.5 h at room temperature. Cells were washed with 1X PBS three times prior to incubation with DAPI (1:1000) for 5 min and mounted with Prolong Gold antifade (Thermo Fisher Scientific). All confocal images were captured with a Zeiss LSM 700 laser scanning microscope with the ZEN microscope software. A 63X oil immersion objective with the pinhole set to 1 airy unit was utilised, as well as the same laser power, gain, and zoom for all sets of images and Z-stacks obtained. Z-stacks were obtained with 1024 × 1024 pixels within a 16-bit range. ImageJ was used to make representative Z-stack montages and merged Z-stack images[60]. General 20X and 40X immunofluorescence imaging for quantification were performed using an EVOS FL AUTO 2 microscope and the associated-imaging system software.

### Growth and infection of iPSC-derived i³Neurons
i³Neuron stem cells (initially provided by Dr. Michael Ward, NINDS/NIH, USA) were maintained at 37 °C in a complete E8 medium (Gibco) on plates coated with Matrigel (Corning) diluted 1:50 in DMEM. Initial 3-day differentiation was induced with DMEM/F-12, HEPES (Gibco) supplemented with 1X N2 supplement (Thermo Fisher Scientific), 1X NEAA, 1X Glutamax and 2 μg/mL doxycycline. 10 μM rock inhibitor (Y-27632, Tocris) was added during the initial plating of cells to be differentiated and cells were plated onto Matrigel-coated plates. Differentiation medium was replaced daily and after 3 days of differentiation, partially differentiated neurons were re-plated into CN media in coverslips in 24-wells coated with 100 μg/mL poly-L-ornithine (PLO, Sigma). CN media consisted of neurobasal plus medium (Gibco) supplemented with 1X B27 supplement (Gibco), 10 ng/mL BDNF (Peprotech), 10 ng/mL NT-3 (Peptrotech), 1 μg/mL laminin (Gibco) and

1 μg/mL doxycycline. After initial replating, neurons were then maintained in CN media without doxycycline for a further 11 days until mature. Fully differentiated neurons were infected with ZIKV viruses at an MOI 10 in CN media. After 2 h, the virus inoculum was removed and replaced with 50% fresh−50% conditioned media. After 96 h p.i., infected cells were fixed 1:1 using 20 mM PIPES pH 6.8, 300 mM NaCl, 10 mM EGTA, 10 mM glucose, 10 mM MgCl₂, 2% (w/v) sucrose and 5% (v/v) PFA for 10 min; and permeabilised for 15 min with 0.3% Triton X-100. Cells were blocked, incubated and imaged as described above.

### Generation, sampling, preparation and infection of human cerebral organoids
ALI-COs, generated from the embryonic stem cell H9 line (WiCell), were prepared as previously described[33,34]. At 82 DIV, four ALI-CO slices derived from two independent cerebral organoids were infected with 5,000,000 PFU of the different ZIKV mutant viruses (corresponding to an MOI of 5 considering an average of 1,000,000 cells/ALI-CO slice) in brain organoid slice media, consisting of: neurobasal medium (Thermo Fisher Scientific) supplemented with 1X B27 supplement (Thermo Fisher Scientific), 0.45% (w/v) glucose (Sigma-Aldrich), 1X Glutamax (Thermo Fisher Scientific) and 1% antibiotic-antimycotic (Thermo Fisher Scientific). After 24 h, the virus inoculum was removed and replaced with fresh media. Culture medium was replaced by fresh medium every 24 h, and at 7 days post-infection, ALI-COs were fixed with 4% PFA for 3 h. Before cryo-sectioning, ALI-CO tissues were sequentially incubated overnight in 5% and 20% sucrose for cryoprotection and later embedded in OCT. ALI-COs were sectioned at 6 μm thickness. All staining steps were done in permeabilisation/blocking buffer (5% BSA, 0.3% Triton X-100 in PBS) at room temperature and their duration was as follows: permeabilisation/blocking (1 h); primary antibodies incubation (overnight); wash steps (x3) 10 min each with just PBS 1X; secondary antibodies incubation (1 h); wash steps (x3) 10 min each with just PBS 1X; and DAPI-counterstaining (1:1000) for 5 min. Sections mounted and cured overnight with Prolong Gold antifade (Thermo Fisher Scientific) and imaged as described above.

### Mosquitoes
All in vivo mosquito experiments used the 15th and 16th laboratory generations of an *Ae. aegypti* colony originally established from wild specimens caught in Barranquilla, Colombia, in 2017. Mosquitoes were maintained under controlled insectary conditions (28° ± 1 °C, 12 h:12 h light:dark cycle and 70% relative humidity)[36]. Larvae were reared in dechlorinated tap water supplemented with a standard diet of Tetra-Min fish food (Tetra). Adults were kept in insect cages (BugDorm) with permanent access to a 10% sucrose solution.

### Mosquito exposure to ZIKV
Mosquitoes were orally challenged with ZIKV by membrane feeding[11,36]. Briefly, 7-day-old female mosquitoes were starved for 24 h prior to the infectious blood meal. The artificial blood meal consisted of a 2:1 mix of washed rabbit erythrocytes (BCL) and ZIKV suspension containing 6.0 × 10⁶ PFU/mL, supplemented with 10 mM adenosine triphosphate (Merck). The infectious blood meal was offered to mosquitoes for 15 min via a membrane-feeding apparatus (Hemotek Ltd.) with porcine intestine as the membrane. Mosquitoes were allowed blood feed for 15 min, and fully engorged females were sorted on ice, transferred into 1-pint cardboard containers and maintained under controlled conditions (28° ± 1 °C, 12 h:12 h light:dark cycle and 70% relative humidity) in a climatic chamber with permanent access to 10% sucrose solution for 21 days in the first experiment and 17 days in the second and third experiments.

At 7, 14 and 21 days post-blood feeding in the first experiment and 7, 10, 14 and 17 days post-blood feeding in the second and third experiments, mosquitoes were paralysed using triethylamine (Sigma)

for 5 min to collect saliva in vitro as previously described[11]. Briefly, the legs of each mosquito were removed, and the proboscis was inserted into a 20-µL pipet tip containing 10 µL of foetal bovine serum (FBS). After 30 min of salivation, the saliva-containing FBS was collected and mixed with 40 µL of DMEM (Sigma) containing 2% PenStrep (Gibco) and then stored at −80 °C until use. After the salivation, the head and body remainder of each mosquito were separated with forceps and scalpels and stored individually in separate tubes at −80 °C.

### ZIKV detection in mosquitoes
To assess transmission potential, infectious ZIKV was detected in saliva samples by focus-forming assay in Vero E6 cells[36]. Briefly, the saliva samples were inoculated onto Vero E6 cells in 96-well plates after removing the cell culture supernatant and incubated at 37 °C for 1 h. The inoculum was removed and replaced by DMEM containing 1.6% carboxymethylcellulose, 1% FBS, 4% Anti-Anti 100X (Gibco) and 1% PenStrep (Gibco). After 5 days of incubation at 37 °C, cells were fixed with 4% formaldehyde solution (Sigma) for 30 min. After fixation, cells were permeabilised with 1× PBS (Gibco) containing 0.1% Triton X-100 (Sigma) for 10 min, blocked with 1% BSA (Sigma) in PBS, and then reacted with mouse anti-flavivirus group antigen monoclonal antibody clone D1-4G2-4-15 (Merck) in PBS for 1 h at room temperature (20–25 °C). After washing with PBS three times, cells were incubated with a 1:1000 dilution of Alexa Fluor 488-conjugated goat anti-mouse IgG (Invitrogen) in PBS for 1 h at room temperature. Immunopositive signals were confirmed under a fluorescence microscope EVOS FL (Thermo Fisher Scientific) with appropriate barrier and excitation filters. Transmission potential was determined qualitatively (i.e., positive or negative).

To determine the rates of ZIKV infection and systemic dissemination, mosquito heads and bodies were placed individually in 300 µL of squash buffer containing 10 mM of Tris pH 8.0 (Thermo Fisher Scientific), 50 mM of NaCl (Thermo Fisher Scientific) and 1.27 mM of EDTA pH 8.0 (Thermo Fisher Scientific) supplemented with proteinase K (Eurobio Scientific) at a final concentration of 0.35 mg/mL and homogenised. A 100-µL sample of the homogenate was transferred to a PCR plate and incubated for 5 min at 56 °C, followed by 10 min at 98 °C to extract total RNA. Detection of ZIKV RNA was performed by two-step RT-PCR. According to the manufacturer's protocol, the cDNA was synthesised using M-MLV reverse transcriptase (Thermo Fisher Scientific) with the extracted RNA and random hexameric primers. The synthesised cDNA was subsequently amplified by PCR using DreamTaq DNA polymerase (Thermo Fisher Scientific) with the following primer pair: ZIKV-PCR-Fw (5′-GTATGGAATGGAGATAAGGCCCA-3′) and ZIKV-PCR-Rv (5′-ACCAGCACTGCCATTGATGTGC-3′), according to the manufacturer's protocol. The cycling conditions were as follows: 2 min at 95 °C, 35 cycles of 30 s at 95 °C, 30 s at 55 °C, and 90 s at 72 °C with a final extension step of 7 min at 72 °C. Amplicons were visualised by electrophoresis on a 2% agarose gel.

The 5′-UTR mutations of the ZIKV strains were verified by Sanger sequencing in 14–15 head samples for each strain in mosquitoes collected on day 14 post-blood feeding of the second experiment. The synthesised cDNA was amplified using PrimeStar GXL (Takara Bio) with the following primer pair: ZIKV-uORF-PCR-Fw (5′-GTATGGAATGGAGATAAGGCCCA-3′) and ZIKV-uORF-PCR-Rv (5′-TATTGATGAGACCCAGTGATGGC-3′). The cycling conditions were as follows: 2 min at 98 °C, 30 cycles of 10 s at 98 °C, 15 s at 55 °C, and 60 s at 68 °C with a final extension step of 2 min at 68 °C. Amplified products were directly sequenced in both directions with the following primer pair: ZIKV-uORF-PCR-Fw (same as above) and ZIKV-uORF-SEQ-Rv (5′-GACCCAGCAGAAGTCCGGCTGGC-3′). The expected 5′-UTR sequence was confirmed in all the tested mosquitoes.

### Statistical analysis of results
Data were analysed in GraphPad Prism 10.2.0 (GraphPad Software, San Diego, CA, USA). Values represent mean ± standard deviation.

Statistical significance was evaluated using two-tailed $t$-tests on luciferase data or $\log_{10}$(virus titre) data, two-way ANOVA on the collapse of vimentin in Vero- and U251-infected cells, and one-way ANOVA to calculate the percentage of infected cells in cerebral organoids. These methods did not assume equal variances for the two populations being compared to calculate the $p$ values. Differences as compared to the control with $p$ value ≤ 0.05 were considered statistically significant.

The prevalence of ZIKV infection, dissemination and transmission in mosquitoes were analysed by logistic regression as a function of the experiment, ZIKV strain and time point. The initial statistical model included all their interactions, which were removed from the final model if their effect was non-significant ($p < 0.05$). Time was considered a continuous variable. To account for minor differences in the measured virus concentration, the $\log_{10}$-transformed blood meal titre was also included in the model as a covariate.

### Ribosomal profiling and RNA-Seq data
$10^7$ Vero, U251 or C6/36 cells were grown on 10-cm dishes and infected with parental or mutant viruses at MOI 3. At 24 h p.i., cells treated with CHX (Sigma-Aldrich, 100 µg/mL) were incubated for 3 min and then rinsed with 5 mL of ice-cold PBS before being flash-frozen. Cells harvested by 'flash-freezing' were directly rinsed with 5 mL of ice-cold PBS, flash-frozen in a dry ice/ethanol bath, and lysed with 400 µL of lysis buffer (20 mM Tris-HCl pH 7.5, 150 mM NaCl, 5 mM MgCl$_2$, 1 mM DTT, 1% Triton X-100, 100 µg/mL CHX and 25 U/mL TURBO DNase). The cells were scraped extensively to ensure lysis, collected and triturated ten times with a 26-G needle. Cell lysates were clarified by centrifugation at $13,000 \times g$ for 20 min at 4 °C. Lysates were subjected to Ribo-Seq and RNA-Seq based on previously reported protocols[18]. Vero and U251 ribosomal RNA (rRNA) was removed using Ribo-Zero Gold rRNA removal kit (Illumina); and C6/36 rRNA contamination was removed by treatment with duplex-specific nuclease as previously described[61]. Library amplicons were constructed using a small RNA cloning strategy adapted to Illumina small RNA v2 to allow multiplexing. Amplicon libraries were deep sequenced using an Illumina NextSeq500 platform. Ribo-Seq and RNA-Seq sequencing data have been deposited in the ArrayExpress database (http://www.ebi.ac.uk/arrayexpress) under the accession codes: E-MTAB-5418 (Ribo-Seq and RNA-Seq in Vero and C6/36 cells infected with PE243 (MOI:3) and pre-treated with CHX; https://www.ebi.ac.uk/biostudies/arrayexpress/studies/E-MTAB-5418?query=E-MTAB-5418), E-MTAB-12967 (RNA-Seq in Vero and U251 cells infected with PE243 (MOI:3) and Dak84 (MOI:3) and flash-frozen; https://www.ebi.ac.uk/biostudies/arrayexpress/studies/E-MTAB-12967?query=E-MTAB-12967), E-MTAB-12968 (RNA-Seq in Vero cells infected with American WT, African-like, uORF1-KO and uORF2-PTC1 (MOI:3); (https://www.ebi.ac.uk/biostudies/arrayexpress/studies/E-MTAB-12968?query=E-MTAB-12968), E-MTAB-12969 (Ribo-Seq in Vero cells infected with American WT, African-like, uORF1-KO and uORF2-PTC1 (MOI:3); https://www.ebi.ac.uk/biostudies/arrayexpress/studies/E-MTAB-12969?query=E-MTAB-12969) and E-MTAB-12970 (Ribo-Seq in Vero and U251 cells infected with PE243 (MOI:3) and Dak84 (MOI:3) and flash-frozen; https://www.ebi.ac.uk/biostudies/arrayexpress/studies/E-MTAB-12970?query=E-MTAB-12970).

### Computational analyses of sequence data
Reads were demultiplexed. Adaptors were trimmed off using the FASTX-Toolkit version 0.0.14 (http://hannonlab.cshl.edu/fastx_toolkit/), and sequences shorter than 25 nt after trimming were discarded. Sequences not linked to an adaptor and sequences constituted of adaptors ligated together were also removed. Reads remaining after this initial selection were sequentially mapped to (i) rRNA; (ii) the American or African ZIKV genomic RNA (viral genome sequences were confirmed by de novo assembly using Trinity version2.8.5); (iii) mRNA; (iv) non-coding RNA (ncRNA); (v) genomic DNA (gDNA) and (vi) a

contaminants database, using bowtie (version 1.2.3) with parameters -v n_mismatches --best (i.e., maximum of n_mismatches mismatches permitted, report best match) where n_mismatches was 1 for mapping in U251 cells and 2 for Vero and C6/36 cells. The remaining reads were classified as unmapped (Supplementary Table 9).

The vRNA database included the Genbank accession numbers for ZIKV/Brazil/PE2015 (KX197192.1) and ZIKV/*A.taylori*-tc/SEN/1984/41662-DAK (KU955592.1.). For ZIKV mutant viruses, the American ZIKV BeH819015 isolate genomic sequence was used (Genbank accession number: KU365778.1). As described in Mutso et al.[25], the missing nucleotides of the 5'-UTR and 3'- UTR (first 13 and last 67 nucleotides of the genome, respectively) were taken from ZIKV isolate PE243 (Genbank accession number: KX197192.1).

The rRNA databases included the Genbank accession numbers listed in Supplementary Table 10. The mRNA sequences for each host were downloaded from NCBI RefSeq. The ncRNA databases for *Chlorocebus sabaeus* were retrieved from Ensembl release 91 and compiled with 534 tRNA sequences of *Macaca mulatta* and Genbank accession XM_007980346; for *Homo sapiens*, the ncRNA sequences were downloaded from Ensembl release 105 and for *A. albopictus* from Ensembl release 55. The gDNA sequences for each host were retrieved from the same Ensembl releases as the ncRNA. The contaminants database comprises Genbank accessions of potential contaminants that might be found in the lab and are used as quality control to ensure that no/very few reads map to this database. The database includes viruses (e.g., HSV-1, PRRSV) and bacteria (e.g., *Escherichia coli*) that are regularly used within the Division of Virology, University of Cambridge, and other common potential contaminants (e.g., mycoplasma sequences, Supplementary Table 11). The composition of libraries (Supplementary Fig. 1) indicated some contamination of Vero Dak84 RNA-Seq (replicate 1 and 2) with Toscana virus (TOSV). These reads do not exhibit the same features as the rest of the reads in those libraries (i.e., length distribution of the reads does not have the same shape), indicating that the contamination occurred after lysates were harvested and thus, they do not affect our conclusions.

Quality control analyses of the reads (Supplementary Figs. 2–5), including length distribution, phasing, and phasing per read length, were carried out using Ribo-Seq and RNA-Seq reads mapping to host mRNA and viral RNA[18]. To assess ribonucleoprotein contamination of Ribo-Seq samples, a comparison was made between the number of reads mapping to CDSs and the 3' UTR of host mRNAs only (Supplementary Fig. 6).

For the visualisation of read densities on the viral genome, the position of the P-site was calculated by adding a fixed +12 nt offset to the 5' end of each read. Read densities were normalised by the sum of total viral RNA and total host mRNA and plotted in reads per million mapped reads (RPM). The translational efficiency (TE) was calculated by the division of reads per kilobase per million mapped reads (RPKM) values for Ribo-Seq by the corresponding RNA-Seq values. A 15-nt mean running filter was previously applied to RPKM values. The genomic regions with coordinates 15–125 or 18–190 for the parental and mutant viruses, respectively, were chosen for the zoom plots of the 5' UTR. The region from genomic coordinate 10,370 to the end of the genome was used for the zoom plots of the 3' UTR. Summarising bar charts show the percentage of reads in each phase, normalised by the total number of reads in the designated region (Supplementary Table 12).

### Reporting summary

Further information on research design is available in the Nature Portfolio Reporting Summary linked to this article.

## Data availability

All data are available in the main text, the Supplementary Information and the Figshare repository (https://doi.org/10.6084/m9.figshare. 25431730). Ribo-Seq and RNA-Seq sequencing data have been deposited in the ArrayExpress database (http://www.ebi.ac.uk/arrayexpress) under the accession codes: E-MTAB-5418, E-MTAB-12967, E-MTAB-12968, E-MTAB-12969 and E-MTAB-12970. Source data are provided with this paper.

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

## Acknowledgements

The authors are indebted to Andrew Firth (Department of Pathology, University of Cambridge) for intellectual input, technical support in bioinformatics and critical reading of the manuscript. The authors would like to thank Alain Kohl (Liverpool School of Tropical Medicine, Liverpool, UK) and Lindomar J. Pena and Rafael Oliveira de Freitas França (Fiocruz Recife, Pernambuco, Brazil) for the provision of PE243 ZIKV RNA used to generate the virus stock; and Andres Merits (University of Tartu, Tartu, Estonia) for providing the pCC1-SP6-Am ZIKV plasmid. The African ZIKV isolate (Dak84) was provided by the 'European Virus Archive goes Global (EVAg) project' funded by the European Union's Horizon 2020

research and innovation programme under Grant Agreement No. 653316. We thank Catherine Lallemand for assistance with mosquito rearing. We are grateful to Claudia Romero-Vivas, who initially provided the mosquito colony from Colombia. N.I. would like to thank David Carpentier (Confocal Microscopy Facility, Department of Pathology, University of Cambridge), Louise Howard (Histology Facility, Department of Pathology, University of Cambridge) and Hashim Ali (University College London) for their helpful assistance. C.L. was supported by the 'Wolfson-Pathology PhD studentship' (University of Cambridge). G.M.C. was supported by a Wellcome Trust Grant [203864/Z/16/Z]. A.M.D. and H.S. were supported by the Wellcome Trust Senior Research Fellowships [106207/Z/14/Z, 220814/Z/20/Z] and a European Research Council Grant [646891] awarded to Andrew Firth (Department of Pathology, University of Cambridge). J.E.D. and A.S.N. were supported by a Wellcome Trust Senior Research Fellowship [219447/Z/19/Z] awarded to J.E.D. L.W.M. and I.G. were supported by a Wellcome Trust Grant [207498/Z/17/Z] to I.G. Sir Henry Dale Fellowship [220620/Z/20/Z] from the Wellcome Trust and the Royal Society, MRC project Grant [MR/T000376/1] and an Isaac Newton Trust/Wellcome Trust ISSF/University of Cambridge Joint Research Grant to V.L. N.McG. is funded by a Wellcome Trust Sir Henry Dale and Royal Society Fellowship [204464/Z/16/Z] and ERC award funded by UKR [EP/Y016262/1]. Research in the A.L. laboratory was funded by the Medical Research Council UK (UKRI: MR/P008658/1; MR/X006867/1 awarded to A.L.). This work was also supported by the French Government's Investissement d'Avenir programme Laboratoire d'Excellence Integrative Biology of Emerging Infectious Diseases (Grant ANR-10-LABX-62-IBEID to L.L.) and MSDAVENIR (Grant INTRANZIGEANT to L.L.). S.T. was supported by an MSCA fellowship from the European Union (Grant MSCA-2021-PF-01 ZIKV-MosTransmit). I.B. was supported by Grants from the Wellcome Trust [202797/Z/16/Z] and the Biotechnology and Biological Sciences Research Council UK [BB/V000306/1]. N.I. was supported by an Isaac Newton Trust Grant [18.40r], a Royal Society Research Grant [RGS \R1\191137], an Isaac Newton Trust/Wellcome Trust ISSF/University of Cambridge Joint Research Grant, a Wellcome Trust Career Development Award [227788/Z/23/Z] and partially by the NIH Grant [5R21AI147172].

## Author contributions

Conceptualisation: N.I.; methodology: C.L., G.M.C., A.M.D., S.T., L.E.C. and V.L.; formal analysis: C.L., S.T., H.S., J.C.K., S.C.G., L.L. and N.I.; investigation: C.L., S.T., H.S. and N.I.; resources: G.G., A.S.N., L.W.M., J.C.K., I.G., J.E.D., S.C.G., A.L. and L.L.; data curation: C.L. and N.I.; writing—original draft: C.L., I.B. and N.I.; writing—review and editing: C.L., G.M.C., A.M.D., S.T., H.S., G.G., A.S.N., L.E.C., L.W.M., N.McG., V.L., J.C.K., I.G., J.E.D., S.C.G., A.L., L.L., I.B. and N.I.; visualisation: C.L. and N.I.; supervision: J.E.D., S.C.G., A.L., L.L., I.B. and N.I.; project administration: N.I.; funding acquisition: V.L., N.McG., I.G., J.E.D., A.L., L.L., I.B. and N.I.

## Competing interests

The authors declare no competing interests.
