## [Peer Review File · Nature Communications]

Zika viruses encode 5' upstream open reading frames required for the infection of human brain cellsReviewers' Comments:

Reviewer #1:

Remarks to the Author:

In the manuscript titled 'Zika viruses encode multiple 5' upstream open 1 reading frames with a role in neurotropism' by Lefèvre et al., the authors describe the discovery of previously unknown uORFs in the Zika virus 5' UTR region using ribosome profiling. The authors showed that Overall, this is an exciting new study into a previously undescribed molecular feature of ZIKV. In vivo experiments in animal models certainly would have been of interest, but I understand that these may not replicate what happens in humans. Overall it is an interesting paper that may appeal to a broad audience with some concerns that the authors are encouraged to address.

Aside from minor issues described below, my major concern is with the design of the mosquito experiments – the design seems a bit strange and the n is neither clear nor consistent between timepoints and groups. I do not think any major scientific conclusions can be drawn from this experiment as it is (see more details below) as it lacks statistical power and experimental consistency. A well-designed mosquito infection experiment would collect mosquitoes from the same experiment at the same timepoints for the same groups and if needed maybe opt for less timepoints to gather more meaningful data. Low n experiments with ~10 mosquitoes can be used as part of a larger experiment when conditions are repeated in the same fashion and samples collected at the same timepoints. With more statistical power, it might be that the African-like ORF, for example, would show delayed/decreased dissemination rates based on the data shown.

Specific comments:

In line 61, the authors say that the American ZIKV strain is generally considered more pathogenic. Without reference or explanation, this statement seems somewhat controversial – while the American/Asian lineage is the virus that caused the pandemic and showed severe disease manifestations, nearly all studies performed in animal models or cell culture indicate (to this reviewer's knowledge) that the African strains (both East and West) may in fact be more pathogenic with faster and more severe birth defects and overall pathogenesis.

In line 95, the authors mention ZIKV isolate Dak84 as a model for the original East African strain. Since I was certain that this is a strain from Dakar (West Africa), I wanted to follow up on this and noted that the provided reference did not mention this strain.

Figure 2 B-D is quite confusing. Specifically, 2B. Instead of first showing all translation levels or uORF1, uORF2 and then the main ORF in one figure for comparison, there are numerous knockout conditions shown, but only explained after already referring to the Figure and in Figure 2C/D. It might be easier to follow if these panels were separated, appropriately numbered, and explained step-by-step.

Lines 199-204: How can the raw luciferase values be lower, but translation efficiency increased? Because Renilla luciferase was also lower? It might be worth explaining that a bit better since it also implies that the term translation efficiency is not ideal here. The final conclusion indicates that

virus infection increased expression, while it sounds more like it ‘decreased expression less than it did for other cellular mRNAs’ (i.e. the Ren-Luc)

Line 224: A revertant at passage 1 using a clean infectious clone mutant stock seems unlikely. The odds of this exact mutation occurring so rapidly and taking over the population seem very low and rather seem to indicate cross contamination of stocks or plasmids in the process. I would encourage (at least in the future, if not for this manuscript) to re-sequence the uORF2-KO plasmid, generate a new clonal stock of plasmid through transformation/single colony, and a new virus stock. I do not believe that ‘the translation efficiency of uORF2 might be critical to ZIKV replication’ as in line 226-227 is a fair statement to make from these data. As pointed out, the uORF-2-KO revertant seems relatively unlikely, and the fact that no virus could be recovered from the uORF2-AUG seems obvious when the authors showed that this essentially completely inhibited translation of the correct polyprotein ORF (Figure 2B), clearly because the second AUG was then not recognized efficiently as a start codon anymore. In fact, data from the uORF2-PTC1 mutant indicate that it may NOT be highly relevant to ZIKV replication.

Line 236: the authors state ‘note that the American WT:uORF2-PTC1 comparison was instead 10:90’ but do not explain why. There is no obvious reason for this choice from the 50:50 data.

Figure 6 legend+data – please indicate the n in the legend, not just through the symbols which are hard to see when everything is overlapping. Where there 2 experiments of 10 and one of 20? Or 2 of 20 and one of 10? Please be accurate here. It also says that there were 3 experiments, but it appears that (maybe?) only 2 experiments were done for each timepoint? 3 experiments were performed, but not all 3 included all 3 timepoints? The mosquito totals in line 360 do not mean much without a breakdown of groups and timepoints. It seems like it might be a low n per group and timepoint, but is hard to decipher. With 5 timepoints and 3 groups, 461 (99+128+234) total mosquitoes would indicate an n of 30 per group and timepoint. Spread out over 3 experiments that appear not to cover all timepoints, it will be hard to make any sense of this data (with an infection range of 66-100). 21 days appears to have only been done in one experiment with differing n between the group (10 for African-like and 20 for the others). Describing any transmission rate from 10 mosquitoes is not robust.

Reviewer #2:

Remarks to the Author:

This study identified the translation of multiple unrecognised upstream open reading frames (uORFs) in the genomic 5' region, in addition to the single long polyprotein found in ZIKV. They used reverse genetics to examine the impact on ZIKV fitness of different uORFs mutant viruses. This finding is unique, however, I have several concerns.

Major:

1. The viral titer of uORF1-KO mutant virus in Fig 3C is significantly higher than that of American WT strain, while the result of Fig 5A shows the opposite results. How to explain this phenomenon?

2. Does uORF1 have different effects on ZIKV infection in different cell lines? Considering four different cell types (Vero, U251, C6/36, i3Neurons) are used for infection experiments in this article, this point is worth discussing.

3. In line 294-297, the authors classified vimentin phenotype into four categories and quantified the vimentin collapse in two independent experiments. The authors should repeat the experiment three times to make the data more rigorous. Besides, it is noticed that the proportion of complete collapse of vimentin filaments (type 4) induced by uORF1-KO virus infection is less than that of Am WT virus. Does this difference affect the infection of two strains in Vero cells? Whether uORF1-KO mutant virus induces delayed formation of vimentin cage in other cell types?

4. It is interesting to know the relationship between cytoskeletal cage, ZIKV uORF1 and neurotropism. Comparing the morphology of cytoskeleton of neurons infected by uORF1-KO and WT ZIKV may help to illustrate this question.

5. How uORF interact with vimentin, and further trigger its rearrangements. Vimentin has many post-translational modifications affecting its assembly/disassembly dynamics. Vimentin has also been reported can be cleaved. The underlying mechanism is critical to address this correlation.

6. Some efforts to address the possibilities raised would greatly strengthen the paper. Without better evidence of cause: effect, I think stating that uORF regulate cytoskeleton remodeling is not yet warranted. Deeper insight into how uORF leads to the cytoskeletal cage would strengthen the paper from a mechanistic standpoint. Without some of these additions I think this paper might be better suited for a specialized journal.

Minor:

1. Line 276-278 states that “uORF1-TAP also appeared in the nuclear fraction, but this was probably due to passive diffusion of this small protein through the nuclear membrane.” However, the data in Fig S12 F and H show that uORF1-mCherry is not present in the nuclear fraction. How to explain this difference? Is this difference related to the type of protein tag?

2. Why are the experimental conditions in Figure S12 based on transfection combined with ZIKV infection, while in Figure 4A, direct transfection of pCAG-ORF1-TAP is performed for protein detection?

3. In line 366-367, the authors conclude that three strains of ZIKV have similar dissemination and transmission prevalence in mosquitoes. But the results of Fig 6B (right panel) appear to show that the transmission prevalence of uORF1-KO mutant virus is lower than that of American WT virus at 17 to 21 days post blood meal.

Reviewer #3:

Remarks to the Author:

This work identifies multiple new open reading frames (ORFs) in the Zika virus 5' genomic region. Using reverse genetics, the authors also investigated the functional relevance of one of the ORFs on Zika virus infection. Moreover, the authors investigated the significance of this ORF on virus growth and tropism in primary cell culture models. Based on the cell culture work, the authors conclude that the ORFs play a role in Zika virus neuropathogenesis. The work presented here is novel and has merit for publication. The use of ribosome profiling, in combination with whole transcriptome sequencing, provides high-resolution maps of Zika virus translation. The ribosome profiling results and how the new ORFs were identified are clearly explained and well presented. Moreover, the functional evaluation of the role of new ORFs on virus growth, virulence, and tropism is a strength of the manuscript. However, the manuscript falls short in the primary cell culture work and the conclusions reached are not convincing based on the methodology. The key concerns are highlighted below.

1) The work starts by identifying two novel ORFs and their role in viral replication was characterised in cell lines. However, it is not clear why only uORF1 was further characterised in primary cell cultures. As the title suggests that the role of multiple ORFs with a role in neurotropism and the first part of the manuscript focuses on both ORFs, the omission of uORF2 in the cell culture experiments was unexpected. As stated in lines 268-269, the authors speculate on the ORFs having intrinsic biological activity based on their sizes. If the biological activity was expected for uORF2, can the authors explain why this has been ignored in the later parts of the manuscript?

2) The neurotropism of Zika virus between different strains is suggested to be due to uORF1 and one of the functions of this uORF1 is speculated to be vimentin reorganisation. However, the use of the African uORF1 alone does not preclude the absence of similar reorganisation in the African wild type. If this is indeed the key difference between the American and African strains then it is important to show the vimentin organisation following infection with the African WT. Moreover, the infection with African WT in all cell culture experiments is necessary to rule out the compensation of this activity by other viral proteins.

3) In lines 298-300, it is suggested the uORF1 likely helps in the formation of cytoskeletal cages during infection. However, it is not clear how this can be suggested based on the pictures shown in Figure 4. Current data is not sufficient to suggest the formation of these cages and higher-resolution images are required.

4) As the function of uORF1 is linked to vimentin reorganization, please demonstrate a similar role in the primary cell cultures.

5) In the cortical organoid cultures, it is suggested that astrocytes and neural progenitors are infected, rather than cortical neurons. This seems to indicate that the cortical neurons are not the primary target cells for Zika virus infection in the brain. Characterisation of the uORFs in the astrocytes and NPCs rather than in cortical neurons will be of added value.

7) In Figures 5 A and C, there is a clear difference between the percentage of infected cells and PFU

i.e. the PFU levels do not correspond to the number of infected cells. What is the explanation for this difference? This suggests that while more cells might be infected, a large number of the infections are abortive and the vimentin reorganisation may not provide a huge benefit in the primary cultures. Moreover, the copy numbers are not shown in the primary cell cultures. Please include this data.

8) Characterising the percentage of virus-infected cells using the confocal images presented in Figure 5 is not appropriate. This is heavily influenced by the thickness (and position) of the slicing and the position of the z-plane. For instance, in Figure 5D characterising the percentage of infected NPCs (nestin +ve cells), the American WT image shows characterisation in what clearly appears to be progenitor regions. However, regions shown in African-like and uORF1-KO do not represent progenitor regions based on morphology. Similarly, in Figure 5E, the GFAP +ve regions in the American WT and uORF1-KO are clearly different regions of the cells. Virus localisation within the cell can be vastly different leading to incorrect characterisation of the percentage of infected cells using these images. FACS would be a more appropriate approach for characterising the percentage of infected cells between the different conditions.

In addition to the above comments, a few minor ones are listed below:

9) The strain names i.e. American isolate or PE243 and African isolate or Dak84 are used interchangeably throughout the text. This is, at times, confusing.

10) In line 91, it is stated "revealing the location of translating ribosomes in a cell". This could be misconstrued as the location of ribosomes in the cell is not identified.

11) In line 126, it is stated that the ratio of negative to positive sense RNA is 1%. This is consistent with previous reports (<https://doi.org/10.3390/v10120728>). In that work, the authors suggest there are significant differences in the ratio of negative to positive-sense RNA between American and African isolates. Do the authors observe the same in their data?

12) In lines 142-143, existing sequence data has been analysed but the number of sequences analysed has not been reported.

13) Mutations are introduced in the stem-loop but portions of this loop following the start codon are, presumably, a part of the ORF. While this becomes clear later on in the section starting on line 168, it would be useful to add a sentence specifying this earlier for clarity.

14) In Figure 3, competition assays are performed. Was there full-length sequencing performed at passage 6 to see if there are any mutations that consistently arise downstream of the uORFs?

15) In lines 236-237, the ratio of American WT:uORF2-PTC1 is switched compared to the others. Why?

16) In lines 291-297, different categories of vimentin reorganisation are described. This is very

subjective and it is not clear what is the added value of this analysis. Can the authors explain how the categorisation was agreed upon and what this data signifies?

17) In line 302, section title, cerebral organoids are mentioned but elsewhere in the text and in the methods, they are called cortical organoids. Which is it?

18) Furthermore, new nomenclature for these organoids has been proposed (<https://doi.org/10.1038/s41586-022-05219-6>). Although the nomenclature is instructive and not authoritative, adaptation to this new nomenclature is advisable given the involvement of key opinion leaders in the nervous system organoid field.

19) Different MOIs are used in the cortical neuron cultures compared to the cortical organoid cultures. What is the rationale for this? Also, for the organoids, how was the MOI calculated?

20) In line 328, it is said that the data corroborate the involvement of Zika virus 5' UTR in neurovirulence. What is the author's definition of neurovirulence? See this article (<https://doi.org/10.1016/j.tins.2022.02.006>) on one definition of neurovirulence that is commonly used. At best, the authors demonstrate neurotropism but no damaging effects due to the uORFs in the primary cultures have been shown.

21) In lines 339-340, it is stated that uORF1 is essential for the infection of cortical neurons and NPCs in human tissue. However, the 2D cell culture data and 3D cell culture data point towards differential cell tropism. Moreover, the 3D culture data indicates preferential infection of NPCs and astrocytes over cortical neurons. Therefore, this statement needs to be adjusted.

22) In Figure 2C, the number of the mutation does not seem to align with the numbers mentioned in the text.

23) In Figure 4C, the virus already appears to be clustered by 18 h.p.i. but the supposed cage structures are absent until 24 h.p.i. These pictures go against the suggestion of cytoskeletal cages around viral factories and protection from innate immune recognition in lines 443-444.

24) The y-axis label on Figure 5G is not correct.

25) PFUs and copy numbers in the organoid experiments are not shown. Why?

26) No information is provided on the PSC line used to make the organoids. Please add this.

27) In the introduction, the authors discuss stigmatization (line 55). The authors are correct in mentioning far-reaching social consequences but they still label their strains by referring to place names (American and African). Although this appears to be standard practice in the Zika virus field, the COVID-19 pandemic has highlighted issues with the nomenclature of strains based on place names. Therefore, the authors are strongly encouraged to take the lead and refer to strain numbers rather than place names.

In summary, the authors present novel results on previously unidentified ORF in the Zika virus genome and characterized the role of these ORFs on Zika virus infection, including infection of primary cell cultures. The manuscript furthers the understanding of Zika virus pathogenesis and provides novel targets for countering Zika virus infection. Overall, the manuscript has merit for publication. It is understandable that it may be difficult to perform repeat experiments on brain organoids with a BSL-3 pathogen but the current conclusions are overreaching from the data presented and the manuscript must, at least, be tempered to reflect the data prior to publication.

Reviewer #4:

Remarks to the Author:

In this manuscript, Lefèvre and colleagues describe newly identified upstream ORFs translated from two different strains of Zika virus. The translation of the uORFs is demonstrated by ribosome profiling in several cell lines including human and mosquito cells, and by reporter assays. Using reporters and viral mutants, the authors show that translation of these uORFs affects downstream translation of the main ORF as well as virus replication. The authors also find that one of the uORFs encodes a protein that localizes with Vimentin fibers and induces their collapse, contributing to the formation of the viral replication compartment, and is important for viral replication specifically in human cells. The ribosome profiling and RNA-seq data shown is of good quality and thoroughly controlled. The main contributions of the paper are the discovery of a viral uORF with a defined function, which is potentially important for viral pathogenesis and epidemiology, and the thorough annotation of ZIKV genomes.

Below are several comments that can help improve clarity and robustness and confidence:

1. The description of the luciferase reporter constructs is not clear enough. From the text and illustration in fig 2A, it is unclear whether the full viral 5'UTR is present in all constructs with the luciferase reporter fused to different initiation contexts (main ORF, uORF1 and uORF2) or whether the reporter is fused at different places in the 5'UTR. If the latter is correct, the illustration in fig 2A that shows identical 5'UTRs is misleading, and it is unclear how a uORF2 PTC can have only a mild effect on the uORF2 reporter that should be terminated before luc is translated. Please expand on the components of the reporter constructs and on the interpretation of the results in this regard.
2. More emphasis should be put on distinction between the translation of uORFs (as translation events with potential regulatory effects) and uORF proteins (as translation products with potential activities) and the contribution of each to the phenotypes observed (including viral mutants phenotype, reporter assay downstream translation). This is especially missing in the context of the African-like fusion uORF American and uORF1. Why does the African-like protein, that has an identical N-terminus as the uORF1 protein, not have the same function?
3. In the ribosome profiling data, a strong peak also appears in the African 5'UTR around position 80 in the -1 frame, in both the human and the mosquito data. Could it be derived from a second in-frame initiation for the annotated fusion-uORF?
4. The authors observe uORF1-protein dependent changes in Vimentin cytoskeletal structures during infection, and quantify it by estimating the extent of Vimentin collapse in each cell. Since the nature of the estimation is subjective, the authors' interpretation of the results is accordingly

balanced. The images provided also show what seems to be a significant change in E protein distribution (potentially representing viral replication factories), scoring the cells based on this signal can be a complementary measure of infected cell morphology changes that will improve the robustness of the quantification.

5. It is possible that one or more of the non-canonical start-codons downstream of PTC1 become active and phenotypically relevant only when uORF1 translation is prevented or prematurely terminated. The authors clearly demonstrate the importance of the frameshift sequences to downstream translation of uORF1, but to confidently determine that no leaky scanning plays a part in the observed phenotype, it seems that the mutations (mut 1-4) should be done in the context of PTC1.

6. Please provide exact IDs/accession numbers for viral genome reference used for alignment, and for the contaminants reference.

Minor comments:

1. Legend of sup fig 6- the meaning of the sentence “the RNA-Seq value is less than unity due to differences in the transcript isoforms present in the sample compared to the RefSeq mRNA database” is unclear, please rephrase.

2. Sup fig 7 D-E- please indicate exactly which samples are shown zoomed in (replicate and treatment).

3. Fig 2 legend- “All p-values are from comparisons of the mutant with the respective wild-type in the same ORF.” but tests in fig 2E-F seem to be comparing between infected and mock samples.

4. Supplementary information directs to fig 3H which does not exist.

5. In the supplementary information text, the initial introduction of the frameshift model does not mention the frameshift happens to ribosomes initiating translation of the main ORF. Adding this information earlier will make the explanation of the complicated multiple-translation region more clear.

6. Fig 5G- the y-axis label should change to reflect the legend text (E+ cells that are also positive for marker). Currently it suggests the reverse ratio (the percent of marker positive cells that are also E+).

7. Line 399- states “The inhibitory effect of uORF1 start codon mutation”, seems like it should be the inhibitory effect of uORF1 expression.

Response to Reviewers

We are very grateful to the reviewers for their expert engagement with the work, their constructive criticisms and their suggestions on how to improve the manuscript. We have responded below to the questions point-by-point and describe several additional experiments that provide new insights, clarify issues and further support our findings, conclusions and mechanistic interpretations. We hope this improved manuscript is now acceptable for publication in Nature Communications.

Reviewer #1 (Remarks to the Author):

In the manuscript titled 'Zika viruses encode multiple 5' upstream open 1 reading frames with a role in neurotropism' by Lefèvre et al., the authors describe the discovery of previously unknown uORFs in the Zika virus 5' UTR region using ribosome profiling. The authors showed that Overall, this is an exciting new study into a previously undescribed molecular feature of ZIKV. In vivo experiments in animal models certainly would have been of interest, but I understand that these may not replicate what happens in humans. Overall it is an interesting paper that may appeal to a broad audience with some concerns that the authors are encouraged to address.

Comment 1.1: *Aside from minor issues described below, my major concern is with the design of the mosquito experiments – the design seems a bit strange and the n is neither clear nor consistent between timepoints and groups. I do not think any major scientific conclusions can be drawn from this experiment as it is (see more details below) as it lacks statistical power and experimental consistency. A well-designed mosquito infection experiment would collect mosquitoes from the same experiment at the same timepoints for the same groups and if needed maybe opt for less timepoints to gather more meaningful data. Low n experiments with ~10 mosquitoes can be used as part of a larger experiment when conditions are repeated in the same fashion and samples collected at the same timepoints. With more statistical power, it might be that the African-like ORF, for example, would show delayed/decreased dissemination rates based on the data shown.*

Response: We apologise for any lack of clarity, but the mosquito experiment design did in fact precisely follow this Reviewer's recommendations. In each experiment, mosquitoes were collected at exactly the same time points for all groups. In the first experiment, mosquitoes were collected on days 7, 14 and 21. In the two subsequent experiments, an additional time point was collected, and the time points were adjusted to days 7, 10, 14 and 17 to gather more meaningful data based on the outcome of the first experiment. The subsequent analysis combined the three experiments in a well-established statistical framework (logistic regression) that accounts for differences in

sample size and does not require all time points to be present in all experiments (i.e., a full-factorial design). This has now been indicated in the manuscript (Lines 414 – 422). To further clarify this point, we have now provided the number of mosquitoes for each time point in each experiment in **Supp Table 4**; and the results of the logistic regression analysis are shown in **Supp Table 5**. We did not detect any statistically significant differences between virus strains in terms of infection, dissemination or transmission prevalence. In the previous version of **Figure 7** (previously Figure 6), the misleading visual impression that the virus with African-like uORF had reduced dissemination was likely a consequence of the poor visibility of the confidence intervals of the logistic fits, for which we apologise to the referees. To address this issue, we have replaced the shaded ribbons representing the confidence intervals with vertical error bars for each data point in the new version of this figure.

Comment 1.2: *In line 61, the authors say that the American ZIKV strain is generally considered more pathogenic. Without reference or explanation, this statement seems somewhat controversial – while the American/Asian lineage is the virus that caused the pandemic and showed severe disease manifestations, nearly all studies performed in animal models or cell culture indicate (to this reviewer’s knowledge) that the African strains (both East and West) may in fact be more pathogenic with faster and more severe birth defects and overall pathogenesis.*

Response: This is a valid point and the text has been rephrased accordingly (Lines 63 and 64).

Comment 1.3: *In line 95, the authors mention ZIKV isolate Dak84 as a model for the original East African strain. Since I was certain that this is a strain from Dakar (West Africa), I wanted to follow up on this and noted that the provided reference did not mention this strain.*

Response: We apologise for this error. The text has been rephrased to address this, and the corresponding reference (Ladner *et al.*, 2016 PMID: 27174284) has been added (presently Line 99).

Comment 1.4: *Figure 2 B-D is quite confusing. Specifically, 2B. Instead of first showing all translation levels or uORF1, uORF2 and then the main ORF in one figure for comparison, there are numerous knockout conditions shown, but only explained after already referring to the Figure and in Figure 2C/D. It might be easier to follow if these panels were separated, appropriately numbered, and explained step-by-step.*

Response: We apologise for the confusion. As suggested by the Reviewer, we have integrated the translation levels of the wild-type uORF1, uORF2 and the main ORF luciferase reporters into a new **Fig 2B** with a more logical order of data presentation. In addition, the results of the different mutant reporter RNA experiments are now shown in **Fig 2E** after being explained in **Fig 2C** and **2D**.

Comment 1.5: *Lines 199-204: How can the raw luciferase values be lower, but translation efficiency increased? Because Renilla luciferase was also lower? It might be worth explaining that a bit better since it also implies that the term translation efficiency is not ideal here. The final conclusion indicates that virus infection increased*

expression, while it sounds more like it 'decreased expression less than it did for other cellular mRNAs' (i.e. the Ren-Luc)

Response: The reviewer is correct to point out that this could have been described more clearly, so we have modified the text to improve this section (Lines 209 – 212). As seen in **Supp Table 1**, the raw luciferase values fall somewhat upon virus infection, but the FF-luc/Ren-Luc ratio increases, indicating that translation of the main and uORFs ORF (FF-luc) is more efficient. We propose that the viral 5' UTR arrangement could potentially protect the expression of the uORFs and the main ORF compared to other cellular mRNAs (in this case, represented by the Ren-Luc reporter mRNA).

Comment 1.6: *Line 224: A revertant at passage 1 using a clean infectious clone mutant stock seems unlikely. The odds of this exact mutation occurring so rapidly and taking over the population seem very low and rather seem to indicate cross contamination of stocks or plasmids in the process. I would encourage (at least in the future, if not for this manuscript) to re-sequence the uORF2-KO plasmid, generate a new clonal stock of plasmid through transformation/single colony, and a new virus stock. I do not believe that 'the translation efficiency of uORF2 might be critical to ZIKV replication' as in line 226-227 is a fair statement to make from these data. As pointed out, the uORF-2-KO revertant seems relatively unlikely, and the fact that no virus could be recovered from the uORF2-AUG seems obvious when the authors showed that this essentially completely inhibited translation of the correct polyprotein ORF (Figure 2B), clearly because the second AUG was then not recognized efficiently as a start codon anymore. In fact, data from the uORF2-PTC1 mutant indicate that it may NOT be highly relevant to ZIKV replication.*

Response: On the sensible suggestion of the reviewer, we resequenced the stock of plasmid pCCI-SP6 ZIKV uORF2-KO used to generate the corresponding infectious clone, and as can be seen in the chromatogram (**Responses Figure 1**, left panel), the sequence is unambiguous, with no trace of cross-contamination with the wild-type plasmid. This supports our view that reversion to the wild-type sequence (grey peak in the underlined A in the right panel chromatogram) occurred during early stages of virus stock generation. We note that passage 1 (p1) virus is a high-titre viral stock derived from passage 0 (p0), which itself is harvested seven days after electroporation of the genomic RNA transcript into Vero cells. Thus, the adenosine (A) to guanosine (G) transition mutation must have occurred during this period. As seen in **Supp Fig 11B**, about 20% of the sequences have reverted to WT by passage 1 and this increases to 40% at passage 5. We hypothesise that the cellular adenosine deaminases acting on RNA (ADARs) might play a role since they catalyse the hydrolytic C6 deamination of A to produce inosine in RNA substrates, which results in nucleotide substitution as inosine is mis-read as G by RNA-dependent polymerases during RNA replication (George *et al.*, 2011 PMID: 21182352). Due to this reversion we did not take the analysis of this mutant virus any further.

Responses **Figure 1:** Sequencing histograms of the infectious clone plasmid (pCCI-SP6 ZIKV uORF2-KO, left panel) and RT-PCR product of the uORF2-KO virus at passage 1. Mutated nucleotides are underlined.

The referee is skeptical about whether the translation efficiency of uORF2 is critical to ZIKV replication, but we argue that the reversion to the wild-type sequence reflects a functional necessity to maintain the uORF2 initiation codon. However, we agree that the toxicity of the uORF2 AUG mutation is explicable in terms of the reduced translation of the main polyprotein, and that we have as yet no strong replication phenotype of the uORF2-PTC1 mutant. In future work, further analysis of the role of uORF2 translation in virus replication is warranted, but for the present manuscript, we agree with the reviewer's comments and have modified the text to illustrate these concerns (Lines 232, 234 – 238).

Comment 1.7: Line 236: the authors state 'note that the American WT:uORF2-PTC1 comparison was instead 10:90' but do not explain why. There is no obvious reason for this choice from the 50:50 data.

Response: For the competition assays presented in this work, cells were infected with an equal ratio (50%:50%) of each virus or 90% of the 'less virulent' virus and 10% of the 'more virulent' virus. Virulence was assigned based on the multi-step growth experiment in U251 cells (**Fig 3C**) and Vero cells (**Supp Fig 11C**). As the viral titre for uORF2-PTC1 mutant was significantly lower at 48 h p.i. in Vero cells, uORF2-PTC1 was assigned as the 'less virulent' virus in this case. We have modified the text to clarify this (Lines 247 – 249).

Comment 1.8: Figure 6 legend+data – please indicate the *n* in the legend, not just through the symbols which are hard to see when everything is overlapping. Where there 2 experiments of 10 and one of 20? Or 2 of 20 and one of 10? Please be accurate here. It also says that there were 3 experiments, but it appears that (maybe?) only 2 experiments were done for each timepoint? 3 experiments were performed, but not all 3 included all 3 timepoints? The mosquito totals in line 360 do not mean much without a breakdown of groups and timepoints. It seems like it might be a low *n* per group and timepoint, but is hard to decipher. With 5 timepoints and 3 groups, 461 (99+128+234) total mosquitoes would indicate an *n* of 30 per group and timepoint. Spread out over 3 experiments that appear not to cover all timepoints, it will be hard to make any sense of this data (with an infection range of 66-100). 21 days appears to have only been done in one experiment with differing *n* between the group (10 for African-like and 20 for the others). Describing any transmission rate from 10 mosquitoes is not robust.

Response: We have now provided a full breakdown of the number of mosquitoes for each time point and each experiment in **Supp Table 4**. To increase the readability of **Figure 7** (previous Figure 6), we have merged the three experiments so that only one symbol is shown for each group at each time point. For clarity, we have also replaced the shaded ribbons representing the confidence intervals of the logistic fits with vertical

error bars for each data point. The main analysis shown in **Supp Table 5** includes the experiment as a covariate and accounts for differences in sample size. It does not compare each time point separately but rather estimates the overall effect of time on the phenotype. Therefore, the results are robust to variation in sample size between individual time points and experiments.

Reviewer #2 (Remarks to the Author):

This study identified the translation of multiple unrecognised upstream open reading frames (uORFs) in the genomic 5' region, in addition to the single long polyprotein found in ZIKV. They used reverse genetics to examine the impact on ZIKV fitness of different uORFs mutant viruses. This finding is unique, however, I have several concerns.

Comment 2.1: *The viral titer of uORF1-KO mutant virus in Fig 3C is significantly higher than that of American WT strain, while the result of Fig 5A shows the opposite results. How to explain this phenomenon?*

Response: The experiments in **Fig 3C** were carried out in U251 cells, an immortalised cell line derived from a malignant glioblastoma tumour, whereas the infections shown in **Fig 6A** (previously Fig 5A) were carried out in differentiated hiPSC-derived glutamatergic cortical neurons that are highly similar to primary cortical neurons at the level of single cells (Handel *et al.*, 2016 PMID: 26740550). Primary cells reflect the phenotype of healthy cells *in vivo* whose innate immune responses are fully competent. However, tumour-derived immortalised cell lines have defects in the induction of interferon-stimulated genes and other antiviral defences. These defects can significantly affect virus replication, especially when cells are infected at lower, more physiologically relevant, multiplicities of infection, as shown in **Fig 3C** and **3D**. Replication of the uORF1-KO virus is clearly reduced in the primary cells, consistent with a role for this element in primary cells.

Comment 2.2: *Does uORF1 have different effects on ZIKV infection in different cell lines? Considering four different cell types (Vero, U251, C6/36, i3Neurons) are used for infection experiments in this article, this point is worth discussing.*

Response: In this study, the uORF1-KO mutant virus was tested in the immortalised cell lines, Vero and U251, and in the hiPSC-derived glutamatergic cortical neurons (i³Neurons). As described in the new **Fig 5B** and **5C**, the uORF1-KO mutant virus can cause some collapse of vimentin in Vero- and U251-infected cells (from 71.28% to 66.38% in Vero cells and from 66.80% to 61.48% in U251 cells, calculated as the average of three independent experiments where 30 cells were quantified per condition). These data indicate that the expression of the uORF1 polypeptide likely helps in the formation of the cytoskeletal cage in infected cells. However, in i³Neurons, this could not be investigated due to the limited infection (**Fig 6A**). The effect of the uORF1-KO mutant virus was not further investigated in the C6/36 mosquito cells as results presented in **Fig 7B** indicate that these uORFs might be dispensable in the mosquito vector.

Comment 2.3: *In line 294-297, the authors classified vimentin phenotype into four categories and quantified the vimentin collapse in two independent experiments. The*

authors should repeat the experiment three times to make the data more rigorous. Besides, it is noticed that the proportion of complete collapse of vimentin filaments (type 4) induced by uORF1-KO virus infection is less than that of Am WT virus. Does this difference affect the infection of two strains in Vero cells? Whether uORF1-KO mutant virus induces delayed formation of vimentin cage in other cell types?

Response: In this revised version of this manuscript, we have quantified the collapse of vimentin as previously described by Zhang and colleagues, 2022 (PMID: 35193960), who investigated how vimentin filaments undergo drastic reorganisation upon viral protein synthesis to form a perinuclear cage-like structure that embraces the ZIKV replication complexes. The collapse of vimentin in infected cells was assessed by measuring the area occupied by this intermediate filament using fluorescence microscopy in relation to the total cell area measured with brightfield microscopy (**Fig 5** and **Supp Fig 15**). These experiments were performed in triplicate, as suggested by the Reviewer, at two different time points with the American WT, the African-like and the uORF1-KO mutant viruses in Vero (**Fig 5A** and **5B**) and U251 (**Fig 5C** and **Supp Fig 15**) cells. These new data confirm that the expression of the uORF1 polypeptide likely helps in the collapse of vimentin and the formation of the cytoskeletal cage in infected cells, and suggest that the African-like 5' arrangement is less efficient than the American WT virus at promoting the formation of the cytoskeletal cage, probably due to its inability to collapse vimentin as described in **Supp Fig 14**. All this new information has been included in Lines 333 – 351 and **Figure 5** and **Supp Fig 15**.

African-like and uORF1-KO mutant viruses reached significantly higher titres (~6-fold) than the American WT in U251 cells (**Fig 3C**), which was further confirmed in **Fig 3D** where both viruses were able to outcompete American WT. The phenotypic differences amongst viruses were also evident, albeit subtler, in Vero-infected cells (**Supp Fig 11C** and **Supp Fig 11D**).

Comment 2.4: *It is interesting to know the relationship between cytoskeletal cage, ZIKV uORF1 and neurotropism. Comparing the morphology of cytoskeleton of neurons infected by uORF1-KO and WT ZIKV may help to illustrate this question.*

Response: A comparison of cytoskeleton morphology in i³Neurons infected with American WT or uORF1-KO mutant virus poses some challenges. First is the limited infection of i³Neurons with the uORF1-KO mutant virus. This makes obtaining statistically significant results from the number of infected cells extremely difficult. Second is the time frame of infection. The collapse of vimentin in Vero and U251 cells was measured within one round of replication (i.e., 24-28 h p.i.). However, significant infection of i³Neurons is delayed until 72 h p.i. (**Fig 6A**), which suggests that multiple rounds of replication have taken place. This results in a mixed population of cells at different stages of infection, and this would make the quantification of the vimentin collapse extremely unreliable. Vimentin is also an early-expressed neuronal intermediate filament that is rapidly downregulated and substituted by mature neurofilaments in mature neurons (Bott *et al.*, 2020. PMID: 31970897). We have attempted to detect expression of vimentin in our i³Neurons, through immunohistochemistry using antibodies targeting vimentin and a marker for mature neurons (MAP2). As shown in **Responses Figure 2**, none of the i³Neurons was positive for vimentin. This suggests that ZIKV might potentially interact with other

intermediate filaments during infection which although extremely interesting, is beyond the scope of the present manuscript.

Figure 2: Representative fluorescence images of i3Neurons stained with antibodies against vimentin (green) and MAP2 (magenta). Nuclei were counter-stained with DAPI (blue). Scale bars, 25 μ m.

Comment 2.5: *How uORF1 interact with vimentin, and further trigger its rearrangements. Vimentin has many post-translational modifications affecting its assembly/disassembly dynamics. Vimentin has also been reported can be cleaved. The underlying mechanism is critical to address this correlation.*

Response: As observed by Western blotting in **Fig 4A** and **Supp Fig 13**, vimentin is not cleaved upon uORF1-transfection nor upon ZIKV infection (see whole-cell lysate – WCL, and cytoskeletal fraction lanes). Thus, we have ruled out that vimentin could be cleaved during infection. In addition, in their investigation of vimentin cage formation during ZIKV infection, Zhang *and colleagues* (PMID: 35193960) were also able to rule out mechanisms that included modification of vimentin by phosphorylation, the aggresome processing machinery that includes dynein, dynastic and microtubules, and the actin retrograde flow. Indeed, these authors speculated at the time (2022) that a viral protein might interact directly or indirectly with vimentin leading to the formation of the cytoskeletal cage (PMID: 35193960).

Since the manuscript was submitted, we have examined the predicted structure of the uORF1-encoded protein using AlphaFold2 and this indicates that uORF1 can adopt an alpha-helical conformation with high confidence (**Fig 4C**). This raises the possibility that it may interact directly with vimentin and modulate its function. To investigate this, we inserted a helix-destabilising proline residue in the middle of the predicted α -helix (I14P, uORF1-TAP-1X Pro). As shown in **Fig 4D**, uORF1-TAP-1X Pro was unable to form granular structures in the cytoplasm in comparison with the unmodified uORF1-TAP (**Fig 4B**). In addition, the fluorescence intensity profiles of vimentin and this mutant uORF1 did not co-align (**Fig 4D**, bottom panel). This supports the view that the helical structure of uORF1 is essential for the collapse of vimentin in transfected cells. These new data are described in Lines 296 – 305 and in **Fig 4C** and **4D**.

Comment 2.6: *Some efforts to address the possibilities raised would greatly strengthen the paper. Without better evidence of cause: effect, I think stating that uORF regulate cytoskeleton remodeling is not yet warranted. Deeper insight into how uORF leads to the cytoskeletal cage would strengthen the paper from a mechanistic standpoint. Without some of these additions I think this paper might be better suited for a specialized journal.*

Response: These concerns are now addressed in comments 2.3 and 2.5.

Comment 2.7: Line 276-278 states that “uORF1-TAP also appeared in the nuclear fraction, but this was probably due to passive diffusion of this small protein through the nuclear membrane.” However, the data in Fig S12 F and H show that uORF1-mCherry is not present in the nuclear fraction. How to explain this difference? Is this difference related to the type of protein tag?

Response: The differences in cellular localisation observed are likely due to the size and chemical properties of the different tags used. The molecular weight of unmodified uORF1 is 3.1 kDa; it is 7.6 kDa when fused to the TAP tag and 30.1 kDa when fused to mCherry. uORF1-TAP is sufficiently small to diffuse passively through the nuclear membrane, but proteins in the 30-60 kDa size threshold are less likely to do so (Timney *et al.*, 2016 PMID: 27697925), consistent with the size of uORF1-mCherry. In addition, we cannot rule out oligomerization of uORF1-mCherry as has previously been observed with some other mCherry fusion proteins (Suzuki *et al.*, 2012 PMID: 22649538).

Comment 2.8: Why are the experimental conditions in Figure S12 based on transfection combined with ZIKV infection, while in Figure 4A, direct transfection of pCAG-ORF1-TAP is performed for protein detection?

Response: In Supp Fig 12, transfection-only of ORF2-FLAG, African ORF-FLAG and uORF1-mCherry and transfection of the aforementioned plasmids coupled with ZIKV infection are shown. These were controls to show similar subcellular localisation of ZIKV uORF-encoded peptides independent of fusion tag and virus co-infection.

Comment 2.9: In line 366-367, the authors conclude that three strains of ZIKV have similar dissemination and transmission prevalence in mosquitoes. But the results of Fig 6B (right panel) appear to show that the transmission prevalence of uORF1-KO mutant virus is lower than that of American WT virus at 17 to 21 days post blood meal.

Response: In the previous version of Figure 6 (now **Figure 7**), the visual impression that the virus with uORF1 has reduced transmission on day 21 is misleading and a formal comparison at this single time point confirms the lack of statistically significant difference (likelihood-ratio test: $\text{Chi}^2 = 1.507$, $p = 0.4707$). This time point is only represented by one experiment and the small sample sizes (2, 7, and 5 mosquitoes in each group, respectively) result in substantial uncertainty. To clarify this point, we have replaced the shaded ribbons representing the confidence intervals of the logistic fit with vertical error bars for each data point in the new version of Figure 6. The 95% confidence intervals of transmission prevalence on day 21 are clearly overlapping between the three groups.

Reviewer #3 (Remarks to the Author):

This work identifies multiple new open reading frames (ORFs) in the Zika virus 5' genomic region. Using reverse genetics, the authors also investigated the functional relevance of one of the ORFs on Zika virus infection. Moreover, the authors investigated the significance of this ORF on virus growth and tropism in primary cell culture models. Based on the cell culture work, the authors conclude that the ORFs play a role in Zika virus neuropathogenesis. The work presented here is novel and has merit for publication. The use of ribosome profiling, in combination with whole

transcriptome sequencing, provides high-resolution maps of Zika virus translation. The ribosome profiling results and how the new ORFs were identified are clearly explained and well presented. Moreover, the functional evaluation of the role of new ORFs on virus growth, virulence, and tropism is a strength of the manuscript. However, the manuscript falls short in the primary cell culture work and the conclusions reached are not convincing based on the methodology. The key concerns are highlighted below.

Comment 3.1: *The work starts by identifying two novel ORFs and their role in viral replication was characterised in cell lines. However, it is not clear why only uORF1 was further characterised in primary cell cultures. As the title suggests that the role of multiple ORFs with a role in neurotropism and the first part of the manuscript focuses on both ORFs, the omission of uORF2 in the cell culture experiments was unexpected. As stated in lines 268-269, the authors speculate on the ORFs having intrinsic biological activity based on their sizes. If the biological activity was expected for uORF2, can the authors explain why this has been ignored in the later parts of the manuscript?*

Response: Initially, we hoped to be able to study uORF2 in more detail in the present work. However, as described in Lines 270 – 275, **Supplementary Information** and **Supp Fig 12**, we could not easily generate a virus unable to express the uORF2 as this ORF is translated in both through translation from the UUG initiation codon and by –1 ribosomal frameshifting. Further mutation of uORF2 is feasible, but the requirement to maintain RNA secondary in this region adds significant complexity to mutant design. In addition, the subcellular localisation of the uORF2-encoded protein did not shed light on any particular cellular compartment (**Supp Fig 13**), and the protein is intrinsically disordered based on the AlphaFold2 prediction (**Supp Fig 14**). Thus, we focused on uORF1. The biological role of uORF2, beyond the modulation of main ORF translation, will be the subject of future work.

Comment 3.2: *The neurotropism of Zika virus between different strains is suggested to be due to uORF1 and one of the functions of this uORF1 is speculated to be vimentin reorganisation. However, the use of the African uORF1 alone does not preclude the absence of similar reorganisation in the African wild type. If this is indeed the key difference between the American and African strains then it is important to show the vimentin organisation following infection with the African WT. Moreover, the infection with African WT in all cell culture experiments is necessary to rule out the compensation of this activity by other viral proteins.*

Response: The role of African uORF in the collapse of vimentin has now been included in the updated version of this manuscript, as discussed in the response to comment 2.3. For these experiments, we used an infectious clone termed ‘African-like’, which includes the American WT polyprotein and the African 5’ UTR arrangement. We consider that the inclusion in this analysis of an authentic African WT virus, such as Dak84, would have included extra variables, such as differences in virus replication or host responses, that might bias the results and not be directly accountable for the specific role of the African uORF in the formation of the cytoskeletal cage.

Comment 3.3: *In lines 298-300, it is suggested the uORF1 likely helps in the formation of cytoskeletal cages during infection. However, it is not clear how this can be*

suggested based on the pictures shown in Figure 4. Current data is not sufficient to suggest the formation of these cages and higher-resolution images are required.

Response: In this updated version of the manuscript, we provide fluorescence profiles for uORF1-TAP and the different cytoskeletal markers that describe the co-alignment between uORF1-TAP and vimentin (**Fig 4B**); and an explanation of how the structure of the uORF1-peptide might be responsible for the collapse of vimentin (Lines 296 – 305, **Fig 4C** and **4D**, comment 2.5). In addition, through quantification of vimentin collapse (Zhang *et al.*, 2022, PMID: 35193960), we show that a mutant virus unable to express the uORF1-peptide (uORF1-KO) does not form the vimentin cytoskeletal cage (Lines 339 – 351, **Fig 5**, **Supp Fig 15** and comment 2.3).

Comment 3.4: *As the function of uORF1 is linked to vimentin reorganization, please demonstrate a similar role in the primary cell cultures.*

Response: This is addressed in the response to another referee (comment 2.4).

Comment 3.5: *In the cortical organoid cultures, it is suggested that astrocytes and neural progenitors are infected, rather than cortical neurons. This seems to indicate that the cortical neurons are not the primary target cells for Zika virus infection in the brain. Characterisation of the uORFs in the astrocytes and NPCs rather than in cortical neurons will be of added value.*

Response: This is an insightful comment and we are currently trying to establish iPSC-derived astrocytes and NPCs in the laboratory, but we are not in a position yet to do these experiments. This will be the subject of future work. However, in additional work presented in the revised manuscript, we assessed the vimentin cage phenotype in the human glioblastoma-astrocytoma cell line, U251 (**Fig 5D** and **Supp Fig 15**). U251 cells are a model for astrocytes as they continue to express the astrocytic marker glial fibrillary acidic protein (GFAP) over many cell passages, and they have been used extensively in ZIKV studies (Chavali *et al.*, 2017 PMID: 28572454; Sher *et al.*, 2019 PMID: 30984137). This indicates that astrocytes are indeed a permissible cell type for the virus.

Comment 3.6: *In Figures 5 A and C, there is a clear difference between the percentage of infected cells and PFU i.e. the PFU levels do not correspond to the number of infected cells. What is the explanation for this difference? This suggests that while more cells might be infected, a large number of the infections are abortive and the vimentin reorganisation may not provide a huge benefit in the primary cultures. Moreover, the copy numbers are not shown in the primary cell cultures. Please include this data.*

Response: Please note that in this figure, the panels in question investigate different cell models, namely, i³Neurons and air-liquid cerebral organoids (ALI-COs). **Fig 6A** (previously **Fig 5A**) shows TCID₅₀ titres of the supernatants of a time-course of i³Neurons infected with the different ZIKV mutant viruses, whereas **Fig 6C** (previously **Fig 5C**), describes the percentage of cells positive for the viral envelope protein (E⁺ cells) in relation to the total number of nuclei in ALI-COs infected with the different mutant viruses. This has been now clarified in the corresponding Figure legend.

Comment 3.7: *Characterising the percentage of virus-infected cells using the confocal images presented in Figure 5 is not appropriate. This is heavily influenced by the thickness (and position) of the slicing and the position of the z-plane. For instance, in Figure 5D characterising the percentage of infected NPCs (nestin +ve cells), the American WT image shows characterisation in what clearly appears to be progenitor regions. However, regions shown in African-like and uORF1-KO do not represent progenitor regions based on morphology. Similarly, in Figure 5E, the GFAP +ve regions in the American WT and uORF1-KO are clearly different regions of the cells. Virus localisation within the cell can be vastly different leading to incorrect characterisation of the percentage of infected cells using these images. FACS would be a more appropriate approach for characterising the percentage of infected cells between the different conditions.*

Response: To improve our estimations of virus infection in ALI-COs, we have now scaled up this experiment to include for each sample four ALI-CO cell slices (derived from two independent cortical organoids) and taken 6- μ m tissue sections from six different Z-axis sections (to minimise Z-axis position bias) and stained for the viral E protein. Positive cells were quantified in relation to the total number of nuclei as previously described for cerebral organoids infected with SARS-CoV-2 (Mesci *et al.*, 2022, PMID: 36327326). These data, obtained from the quantification of 33 images per virus type with approximately 400 – 500 nuclei per image, are presented in **Fig 6C** (previous Fig 5C). In addition, an updated quantification of cells positive for nestin, GFAP or MAP2 that are also positive for the viral E protein (eleven images per cellular marker) has now been included in **Fig 6G**. New confocal images for nestin- and GFAP-positive cells infected with African-like and uORF1-KO mutant viruses that resemble regions of progenitor cells have also been included in **Fig 6D** and **Fig 6E**. Together, these data increase our accuracy in the determination of the percentages of infected cells and corroborate the involvement of the ZIKV 5' UTR uORFs in neurotropism.

With regard to the Reviewer's comment that flow cytometric techniques can be more accurate than microscopy in determining the percentage of infected cells, we have also attempted to perform flow cytometric analyses of ALI-COs infected with American WT, African-like, and uORF1-KO viruses (**Responses Figure 3**). Infected ALI-COs were manually dissociated by pipetting after papain digestion for 20 min at 37°C before being fixed in 4% formaldehyde for 20 min. Cells were later stained, as detailed in **Responses Figure 3D**. Unfortunately, due to their fragile nature, many cells were lost upon dissociation of the ALI-COs, resulting in sub-optimal flow cytometric data. However, we were able to identify neural progenitor cells (NPCs; positive for vimentin and SOX2 [vimentin⁺SOX2⁺, **Responses Figure 3A**]) and to determine the proportion of ZIKV-infected cells (E⁺ cells) by gating on the NPCs.

C.

Median Fluorescence			
	ZIKV E prot	Negative Control	Fold change over negative control
American WT_1	Cellular debris only		
American WT_2	15051	1128	13.3430851
African-like_1	17280	1327	13.0218538
African-like_2	13837	2426	5.70362737
uORF 1 KO_1	3244	956	3.39330544
uORF 1 KO_2	6585	1272	5.17688679

D.

	Species	Catalogue number	Dilution
Primary antibodies			
anti-vimentin	mouse	Abcam, ab8069	1:500
anti-SOX2	goat	Novusbio, AF2018	1:500
anti-ZIKV E prot	rabbit	GeneTex, GTX133314	1:200
Secondary antibodies			
anti-mouse	donkey	Invitrogen Alexa Fluor 488, A-21202	1:1000
anti-rabbit	donkey	Invitrogen Alexa Fluor 647, A-31573	1:1000
anti-goat	donkey	Invitrogen Alexa Fluor 594, A-11058	1:1000

Responses Figure 3: Flow cytometric analyses of ZIKV-infected ALI-COs. (A) Flow cytometric gating strategy used to identify NPCs (vimentin⁺SOX2⁺ cells). (B) ZIKV E⁺ cells were identified by analysing dot plots and histograms. (C) Mean fluorescence intensity of ZIKV E⁺ A647. (D) Antibodies used for flow cytometric analyses. Data was acquired using an Attune NxT (ThermoFisher, configuration can be found here <https://ppms.eu/path-cam/?FLOW>) and analysed using FlowJo v10.7 (Treestar). Fluorescence minus one (FMO) was used as a negative control.

It was difficult to determine the actual percentage of infected cells with dot plots and histograms due to the low cell number and spread of the fluorescence (**Responses Figure 3B**). However, the Mean Fluorescence Intensity data (**Responses Figure 3C**) revealed that the uORF1-KO mutant virus, and to a lesser extent, the African-like mutant virus, displayed reduced neurotropism in comparison with the American WT. Thus, the flow cytometric data aligns with our fluorescence microscopy findings (**Fig 6C-G** of the manuscript). In future experiments, we hope to be able to optimise this methodology. While the flow cytometric data we have aligns with our microscopy findings, we have not added these data to the revised manuscript due to their limited quality.

Comment 3.8: *The strain names i.e. American isolate or PE243 and African isolate or Dak84 are used interchangeably throughout the text. This is, at times, confusing.*

Response: We agree with the Reviewer that this can be confusing. To prevent it, we have renamed PE243 and Dak84 as ZIKV American isolate PE243 and ZIKV African isolate Dak84, respectively, throughout the text.

Comment 3.9: *In line 91, it is stated "revealing the location of translating ribosomes in a cell". This could be misconstrued as the location of ribosomes in the cell is not identified.*

Response: To avoid this confusion, this has been rephrased as "revealing the position of translating ribosomes on the mRNA at the time of harvesting with single-nucleotide precision" (Line 95).

Comment 3.10: *In line 126, it is stated that the ratio of negative to positive sense RNA is 1%. This is consistent with previous reports (<https://doi.org/10.3390/v10120728>). In that work, the authors suggest there are significant differences in the ratio of negative to positive-sense RNA between American and African isolates. Do the authors observe the same in their data?*

Response: Barnard *et al.*, 2018 (PMID: 30572570) quantified by RT-qPCR positive- and negative-strand viral RNA and found that the Asian/American lineage isolates (PRVABC59 and HS-2015-BA-01) displayed substantially more negative-strand replicative intermediates than the African lineage isolate (MR-766) in human astrocytoma cells (U251 cells). The negative-to-positive RNA ratio was 3-4% for the American isolates and 1% for the African isolate. We have quantified this negative-to-positive RNA ratio in our available RNA-Seq samples (**Responses Table 1**), and we do not see any significant differences in either Vero- or U251-infected cells with PE243 or Dak84. Certain data points might suggest significance in isolation, but none were statistically significant. We hypothesise that more replicates are needed to draw definite conclusions.

Responses Table 1. Negative-to-positive RNA ratio in available RNA-Seq samples. Vero_REP1, Vero_REP2 and U251_REP1 are data included in the manuscript. U251_REP2 and U251_REP3 are unpublished data from the Irigoyen lab.

Sample	vRNA		(-) RNA : (+) RNA Ratio (%)
	Pos.	Neg.	
PE243_RNA_Vero_REP1	538022	5380	0.999%
PE243_RNA_Vero_REP2	548739	4422	0.805%
Dak84_RNA_Vero_REP1	25191	716	2.842%
Dak84_RNA_Vero_REP2	81987	750	0.914%
PE243_RNA_U251_REP1	783443	32425	4.138%
PE243_RNA_U251_REP2	261762	8552	3.267%
PE243_RNA_U251_REP3	233097	7271	3.119%
Dak84_RNA_U251_REP1	1267963	29684	2.341%
Dak84_RNA_U251_REP2	159753	108	0.067%
Dak84_RNA_U251_REP3	309561	6710	2.167%

Comment 3.11: *In lines 142-143, existing sequence data has been analysed but the number of sequences analysed has not been reported.*

Response: The alignment of representative 5' UTR sequences of African and Asian/American isolates has now been included in **Supp Figure 8**. This alignment includes seven African isolates from 1968 to 2015 and 21 Asian/American isolates from 1966 to 2016. As observed in **Supp Figure 8**, all the African isolates contain the following 5' UTR arrangement ⁷⁷UAUUGGA⁸⁴, whereas, in the Asian/American lineage, this has changed to ⁷⁷UAUUUGGA⁸⁵ due to the insertion of a uracil residue at position 81 in the 1966 Malaysian lineage.

Comment 3.12: *Mutations are introduced in the stem-loop but portions of this loop following the start codon are, presumably, a part of the ORF. While this becomes clear later on in the section starting on line 168, it would be useful to add a sentence specifying this earlier for clarity.*

Response: We apologise if this was unclear. We have modified the diagrams in **Fig. 2A** to clarify this.

Comment 3.13: *In Figure 3, competition assays are performed. Was there full-length sequencing performed at passage 6 to see if there are any mutations that consistently arise downstream of the uORFs?*

Response: All ZIKV mutant viruses were subjected to RNA-Seq at passage 2. No additional mutations (apart from the expected mutation in the 5' UTR) were found in the American WT, uORF1-KO or uORF2-PTC1 mutant viruses. Two point mutations arose in the African-like virus; C862A, which results in a phenylalanine to leucine change in the membrane protein, and the second A3221C, changing a lysine into glutamine in NS1. These mutations are a long way downstream of the 5' UTR and probably cannot affect local folding in the 5' UTR. Further, they do not remarkably

change amino acid polarity or charge to suspect they could strongly influence virus replication. This is something that might warrant investigation in future studies.

Comment 3.14: *In lines 236-237, the ratio of American WT:uORF2-PTC1 is switched compared to the others. Why?*

Response: This is addressed in the response to another referee (comment 1.7).

Comment 3.15: *In lines 291-297, different categories of vimentin reorganisation are described. This is very subjective and it is not clear what is the added value of this analysis. Can the authors explain how the categorisation was agreed upon and what this data signifies?*

Response: This is addressed in the response to another referee (comment 2.3).

Comment 3.16: *In line 302, section title, cerebral organoids are mentioned but elsewhere in the text and in the methods, they are called cortical organoids. Which is it?*

Response: Brain organoids used in this piece of research are cerebral cortical organoids that resemble part of the dorsal forebrain. The nomenclature in the manuscript has been changed accordingly.

Comment 3.17: *Furthermore, new nomenclature for these organoids has been proposed (<https://doi.org/10.1038/s41586-022-05219-6>). Although the nomenclature is instructive and not authoritative, adaptation to this new nomenclature is advisable given the involvement of key opinion leaders in the nervous system organoid field.*

Response: See comment 3.16 immediately above.

Comment 3.18: *Different MOIs are used in the cortical neuron cultures compared to the cortical organoid cultures. What is the rationale for this? Also, for the organoids, how was the MOI calculated?*

Response: i³Neurons are less permissive to infection than regular cell lines (e.g. Vero and U251) thus, a high MOI (10 PFU/cell based on titre in Vero cells) was used to get sufficient neuron infection. In addition, based on previous optimisation, an MOI of 5 (based on titre in Vero cells) was selected as it ensures appropriate infection of cerebral cortical organoids. MOI was calculated according to the average number of cells per ALI-CO slice at 82 days *in vitro* culture, which is approx. 1,000,000 cells/slice.

Comment 3.19: *In line 328, it is said that the data corroborate the involvement of Zika virus 5' UTR in neurovirulence. What is the author's definition of neurovirulence? See this article (<https://doi.org/10.1016/j.tins.2022.02.006>) on one definition of neurovirulence that is commonly used. At best, the authors demonstrate neurotropism but no damaging effects due to the uORFs in the primary cultures have been shown.*

Response: The reviewer makes a salient point. Our assays in primary cultures reveal the ability of ZIKV to infect and replicate in various cells of the nervous system (e.g., neural progenitor cells, cells of the astroglial lineage and cortical neurons), but they do

not demonstrate the ability of ZIKV to cause pathology. We have corrected the sentence in question to take this into account (Line 380).

Comment 3.20: *In lines 339-340, it is stated that uORF1 is essential for the infection of cortical neurons and NPCs in human tissue. However, the 2D cell culture data and 3D cell culture data point towards differential cell tropism. Moreover, the 3D culture data indicates preferential infection of NPCs and astrocytes over cortical neurons. Therefore, this statement needs to be adjusted.*

Response: This sentence has been altered to take the reviewer's comment into account (see Lines 392 – 394 and also the response to another referee [comment 3.5]).

Comment 3.21: *In Figure 2C, the number of the mutation does not seem to align with the numbers mentioned in the text.*

Response: We apologise for a typo in Fig 2C where the uORF2-PTC1 in pink should have been read as uORF2-KO. This has been corrected.

Comment 3.22: *In Figure 4C, the virus already appears to be clustered by 18 h.p.i. but the supposed cage structures are absent until 24 h.p.i. These pictures go against the suggestion of cytoskeletal cages around viral factories and protection from innate immune recognition in lines 443-444.*

Response: Although cytoskeletal remodelling by ZIKV has previously been suggested to interfere with antiviral responses (Cortese *et al.*, 2017, PMID: 28249158), the clustering of the virus at 18 h p.i. and prior to the formation of the cytoskeletal cage contradicts the idea of protection from innate immune recognition. We have thus modified this statement in the manuscript (Lines 323, 509 – 511).

Comment 3.23: *The y-axis label on Figure 5G is not correct.*

Response: This has been corrected.

Comment 3.24: *PFUs and copy numbers in the organoid experiments are not shown. Why?*

Response: At 82 days *in vitro*, two ALI-CO slices derived from two independent cortical organoids per mutant virus were infected with 5,000,000 PFU of different ZIKV mutant viruses (corresponding to an MOI of 5 considering an average of 1,000,000 cells per ALI-CO slice) in brain organoid slice media. This information has now been included in the text (Lines 991 – 993).

Comment 3.25: *No information is provided on the PSC line used to make the organoids. Please add this.*

Response: ALI-COs were generated from the embryonic stem cell H9 line (WiCell). This information has now been included in the text (Line 990).

Comment 3.26: *In the introduction, the authors discuss stigmatization (line 55). The*

authors are correct in mentioning far-reaching social consequences but they still label their strains by referring to place names (American and African). Although this appears to be standard practice in the Zika virus field, the COVID-19 pandemic has highlighted issues with the nomenclature of strains based on place names. Therefore, the authors are strongly encouraged to take the lead and refer to strain numbers rather than place names.

Response: We completely agree with this Reviewer that the current classification of ZIKV into the two different lineages, African and Asian/American, is probably inadequate and needs updating. Not only is it stigmatising, but also it might not reflect the genetic diversity that the virus accumulated after the 2015/2016 epidemic.

To lead this endeavour, we have included a paragraph in the Discussion (Lines 573 – 583) that advocates the use of the new terminology proposed by Seabra *et al.*, 2022 (PMID: 35478717). This classification is based on phylogenetic analyses, clustering techniques, within- and between-group pairwise genetic distances, and evolutionary analyses to define genetic groups and subgroups. Their proposed nomenclature avoids geographical terminology, using alpha-numerical labels instead. For instance, the formerly named African and Asian lineages are called ZA and ZB genotypes here. Whereas Dak84 belongs to the ZA group, the PE243 isolate will now be in the ZB.2.0 group, which includes the Polynesian sequences and the basal American lineage.

Comment 3.27: *In summary, the authors present novel results on previously unidentified ORF in the Zika virus genome and characterized the role of these ORFs on Zika virus infection, including infection of primary cell cultures. The manuscript furthers the understanding of Zika virus pathogenesis and provides novel targets for countering Zika virus infection. Overall, the manuscript has merit for publication. It is understandable that it may be difficult to perform repeat experiments on brain organoids with a BSL-3 pathogen but the current conclusions are overreaching from the data presented and the manuscript must, at least, be tempered to reflect the data prior to publication.*

Response: We thank the Reviewer for this comment and hope we have addressed their concerns in this rebuttal letter and in the updated version of this manuscript.

Reviewer #4 (Remarks to the Author):

In this manuscript, Lefèvre and colleagues describe newly identified upstream ORFs translated from two different strains of Zika virus. The translation of the uORFs is demonstrated by ribosome profiling in several cell lines including human and mosquito cells, and by reporter assays. Using reporters and viral mutants, the authors show that translation of these uORFs affects downstream translation of the main ORF as well as virus replication. The authors also find that one of the uORFs encodes a protein that localizes with Vimentin fibers and induces their collapse, contributing to the formation of the viral replication compartment, and is important for viral replication specifically in human cells. The ribosome profiling and RNA-seq data shown is of good quality and thoroughly controlled. The main contributions of the paper are the discovery of a viral uORF with a defined function, which is potentially important for viral pathogenesis and epidemiology, and the thorough annotation of ZIKV genomes.

Below are several comments that can help improve clarity and robustness and confidence:

Comment 4.1: *The description of the luciferase reporter constructs is not clear enough. From the text and illustration in fig 2A, it is unclear whether the full viral 5'UTR is present in all constructs with the luciferase reporter fused to different initiation contexts (main ORF, uORF1 and uORF2) or whether the reporter is fused at different places in the 5'UTR. If the latter is correct, the illustration in fig 2A that shows identical 5'UTRs is misleading, and it is unclear how a uORF2 PTC can have only a mild effect on the uORF2 reporter that should be terminated before luc is translated. Please expand on the components of the reporter constructs and on the interpretation of the results in this regard.*

Response: We apologise for the lack of clarity. T7 RNA polymerase-capped-derived synthetic reporter mRNAs were prepared in which PE243 uORF1, uORF2 or a 5' portion of the main ORF was placed upstream of, and in frame with, the firefly luciferase (FF-Luc) reporter gene. In addition, ZIKV and FF-Luc sequences are separated by the short, foot and mouth disease virus 2A autoprotease-encoding sequence that liberates the FF-Luc enzyme following expression in cells.

uORF1-2A-FFLuc includes the complete 5'-UTR (107 nucleotides) plus 22 nucleotides of the polyprotein. Note that in this case, the uORF1 stop codon has been substituted by a tryptophan residue to allow the expression of the luciferase reporter. uORF2-2A-FFLuc includes the complete 5'-UTR (107 nucleotides) plus 89 nucleotides of the polyprotein, and the main ORF-2A-FFLuc includes the complete 5'-UTR (107 nucleotides) plus 87 nucleotides of the polyprotein. Differences in the polyprotein length are due to frame correction. The corresponding oligonucleotides for this cloning strategy are included in **Supp Table 7**. **Fig 2A** has been changed accordingly, and the description of the luciferase reporters has now been added to the corresponding figure legend for clarity.

Comment 4.2: *More emphasis should be put on distinction between the translation of uORFs (as translation events with potential regulatory effects) and uORF proteins (as translation products with potential activities) and the contribution of each to the phenotypes observed (including viral mutants phenotype, reporter assay downstream translation). This is especially missing in the context of the African-like fusion uORF American and uORF1. Why does the African-like protein, that has an identical N-terminus as the uORF1 protein, not have the same function?*

Response: We have now made minor text modifications to improve clarity when discussing uORF function in terms of protein synthesis regulation, encoded product function and how these might contribute to mutant phenotypes. We have also provided new data on the African-like fused uORF and its function. We transfected Vero cells with plasmids encoding the African uORF and the American uORF2 (**Supp Fig 14**) and performed confocal analysis with different proteins marking cytoskeletal proteins. Whereas the uORF2 peptide localised in the cytoplasm with no preference for any cytoskeletal marker (**Supp Fig 14B**), the African uORF-encoded protein had a granular perinuclear pattern and partially co-aligned with vimentin (**Supp Fig 14C**) similarly to uORF1 peptide but was unable to collapse intermediate filaments. This has been further discussed in Lines 307 – 316. We have also tested how the African uORF

protein helps in the formation of the cytoskeletal cage during infection and our results indicate that the African-like 5' arrangement is less efficient than the American WT virus at promoting the formation of the cytoskeletal cage, probably due to its inability to collapse vimentin. This has now been addressed in Lines 339 – 351, **Fig 5, Supp Fig 15**, and previous comment 2.3.

Comment 4.3: *In the ribosome profiling data, a strong peak also appears in the African 5'UTR around position 80 in the -1 frame, in both the human and the mosquito data. Could it be derived from a second in-frame initiation for the annotated fusion-uORF?*

Response: The strong peak at position 79 in the African uORF coincides with the position of the initiation of translation of the uORF2 in the Asian/American lineage. Thus, the accumulation of RPFs at this site could denote the utilisation of this UUG codon as an alternative initiation site on the African uORF downstream of its 'canonical' initiation codon at position 25. To test this, the putative alternative initiation codon in the African uORF was knocked out in the context of the African uORF-2A-

FFLuc reporter. Here, any luciferase expression driven from the utilisation of this “internal” initiation codon will be prevented, and the luciferase output of the reporter RNA (African uORF-uORF2-KO-2A-FFLuc) might be expected to be lower than that corresponding to the wild-type African uORF-2A-FFLuc. However, as observed in **Responses Figure 4**, the translation efficiency of the two RNAs was very similar. This indicates that the UUG initiation codon is not utilised, at least in this context of luciferase reporters.

Responses Figure 4: Relative FF-Luc activity for the African uORF-2A-FFLuc and the African uORF-uORF2 KO-2A-FFLuc compared to uORF2-2A-FFLuc. 100% translation accounted for the main ORF WT (data not shown). All *t*-tests were two-tailed and did not assume equal variance for the two populations being compared.

The origin of the RiboSeq spike at position 79 is uncertain. Ostensibly, it would seem to represent, as the Reviewer suggests, a second in-frame initiation event from the annotated African uORF. However, we note that this peak is not present in infected U251 cells. Further, the read length of this peak is centered around 26 nt rather than the predominant 29-30 nt distribution, suggesting its origin is unusual. We have hypothesised that ribosomes initiating at the main ORF AUG might stall 40S complexes over the UUG and promote more efficient initiation at this non-AUG codon, as it is been known that in some circumstances, such stalled ribosomes might have an altered mRNA footprint (e.g. Wu *et al.*, PMID: 30686592). However, as we have shown that mutation of the UUG does not affect translational output, at least in transfection assays, we cannot easily explain the origin of this peak. To alert readers to this, we have added a sentence in the corresponding legend (Lines 656 – 658) pointing out that this peak has an unusual read length and was seen only in Vero cells. Note that all other peaks in the manuscript datasets are derived from ribosome-protected fragments of conventional length (predominantly 29-30 nt distribution).

Comment 4.4: The authors observe uORF1-protein dependent changes in Vimentin cytoskeletal structures during infection, and quantify it by estimating the extent of Vimentin collapse in each cell. Since the nature of the estimation is subjective, the authors' interpretation of the results is accordingly balanced. The images provided also show what seems to be a significant change in E protein distribution (potentially representing viral replication factories), scoring the cells based on this signal can be a complementary measure of infected cell morphology changes that will improve the robustness of the quantification.

Response: This is addressed in the response to another referee (comment 2.3).

Comment 4.5: It is possible that one or more of the non-canonical start-codons downstream of PTC1 become active and phenotypically relevant only when uORF2 translation is prevented or prematurely terminated. The authors clearly demonstrate the importance of the frameshift sequences to downstream translation of uORF2, but to confidently determine that no leaky scanning plays a part in the observed phenotype, it seems that the mutations (mut 1-4) should be done in the context of PTC1.

Response: In order to determine that leaky scanning did not play a role in uORF2 translation, the experiment suggested by the Reviewer was performed. As described in the Supplementary Information section, the potential initiation sites between codons 6 and 34 were sequentially mutated within the context of the uORF2-PTC1-2A-FFLuc reporter mRNA and tested in transfected cells. In all cases, luciferase levels were unchanged compared to the non-mutated uORF2-PTC1-2A-FFLuc (see attached **Responses Figure 5** and **Responses Table 2**). Although translation efficiency was slightly lower than the corresponding uORF2-2A-FFLuc reporter mRNA, these differences were not statistically significant when calculated as an ordinary one-way ANOVA. These results clearly show that leaky scanning does not play an obvious role in the observed uORF2 translation phenotype.

Responses Figure 5: Relative FF-Luc activity for the different 'alternative' non-canonical initiation codon mutants of uORF2-PTC1-2A-FFLuc in Vero-transfected cells compared to uORF2-2A-FFLuc. 100% translation accounted for the main ORF WT (red square). Statistical analysis was repeated measures one-way ANOVA. All p-values are from comparisons of the 'mutant' luciferase reporters with the uORF2- WT-FFLuc.

Responses Table 2: Relative FF-Luc activity (in %) for the tested luciferase mutants in three replicates.

	REP1	REP2	REP3	Average	SD
ORF2-WT	5.32177874	5.51964889	7.18403616	6.00848793	1.02285061
ORF2-PTC1	3.78483056	4.11865135	5.98294083	4.62880758	1.18453228
ORF2-PTC1 Mut1	3.74464271	4.37789284	5.37089908	4.49781154	0.81973339
ORF2-PTC1 Mut2	3.44259787	4.44200032	5.5100042	4.46486746	1.03389284
ORF2-PTC1 Mut3	3.75493867	4.3470634	5.81266987	4.63822398	1.05931359
ORF2-PTC1 Mut4	3.62749282	4.52040949	5.66912101	4.60567444	1.02348131

Comment 4.6: Please provide exact IDs/accession numbers for viral genome reference used for alignment, and for the contaminants reference.

Response: Genbank accession numbers for viral genomes used for alignment have now been included in the 'Computational analyses of sequence data' section (Lines 1136 – 1141).

The accession numbers of all mycoplasma sequences in the contaminants database have been included in the new **Supp Table 11**. The accession numbers of the genomes of the viruses investigated in the lab are not provided as they are redacted for confidentiality by Prof. Andrew Firth (Department of Pathology, University of Cambridge).

Comment 4.7: Legend of sup fig 6- the meaning of the sentence "the RNA-Seq value is less than unity due to differences in the transcript isoforms present in the sample compared to the RefSeq mRNA database" is unclear, please rephrase.

Response: This has been rephrased for clarity.

Comment 4.8: Sup fig 7 D-E- please indicate exactly which samples are shown zoomed in (replicate and treatment).

Response: The zoom plots of **Supp Fig 7D** and **7E** were generated from the RNA-Seq densities of replicate 1 of flash-frozen Vero cells infected for 24 h with PE243 and Dak84 ZIKV isolates. This information has been introduced into the corresponding figure legend.

Comment 4.9: Fig 2 legend- "All p-values are from comparisons of the mutant with the respective wild-type in the same ORF." but tests in fig 2E-F seem to be comparing between infected and mock samples.

Response: In **Fig 2**, all p-values are from comparisons of the mutant with the respective non-mutated luciferase reporter (i.e., derived from the American wild-type) in the same ORF. This has been clarified in the corresponding figure legend.

Comment 4.10: Supplementary information directs to fig 3H which does not exist.

Response: This has been corrected.

Comment 4.11: *In the supplementary information text, the initial introduction of the frameshift model does not mention the frameshift happens to ribosomes initiating translation of the main ORF. Adding this information earlier will make the explanation of the complicated multiple-translation region more clear.*

Response: Thank you for pointing out this omission; a clarification sentence has been introduced into the Supplementary Text.

Comment 4.12: *Fig 5G- the y-axis label should change to reflect the legend text (E+ cells that are also positive for marker). Currently it suggests the reverse ratio (the percent of marker positive cells that are also E+).*

Response: Apologies, this has been corrected.

Comment 4.13: *Line 399- states “The inhibitory effect of uORF1 start codon mutation”, seems like it should be the inhibitory effect of uORF1 expression.*

Response: Apologies, this has been corrected in the text.

Reviewers' Comments:

Reviewer #2:

Remarks to the Author:

I am fine with the revision.

Reviewer #3:

Remarks to the Author:

The authors have performed additional experiments and have addressed most of my comments. Overall, this work uses novel techniques to advance knowledge on potential determinants of Zika virus neurotropism. I have a few additional comments on the latest version of the manuscript below:

- 1) The title still refers to multiple ORFs with a role in neurotropism. However, since uORF2 could not be characterized, the authors only show the role of uORF1 in neurotropism. Thus, the title needs to be adjusted to reflect this.
- 2) Line 83, of the introduction states that "... uORF1 can modulate virus growth, virulence and tropism in the human brain". What is shown are results from human brain organoids and not the human brain so please adjust this.
- 3) In line 84, it is stated that the uORF1 is essential for neurotropism in the adult and developing brain. However, the work presented here does not show that. The brain organoids, to an extent, mimic the developing embryonic brain but not the adult brain.
- 4) It is not clear how the data in Figure 3 supports the statement in lines 236-238. Would it not simply be the case that the reversion is necessary because it results in increased translation of the main ORF rather than the an added role for uORF2 in translation?
- 5) Please add, in figure 6, the E+ cells/total nuclei data for the i3Nuerons infected with ZIKV.
- 6) The identity of the organoids is still not clear. In lines 353-354, they are called cerebral cortical organoids. Cerebral organoids was the term used for whole-brain organoids (the proposed nomenclature refers to them as unguided neural organoids). Do these slice cultures have whole-brain identity or they only represent the cerebral cortex (making them regionalized neural organoids).
- 7) In figure 6, the regional organization of the organoid can only be seen in the American WT conditions in 6D and 6E. Although the other two panels show NPCs and "astrocyte-precursors" (which would also be neural stem cells), there is a lack of organization. Can the authors show a similarly organized panel for the African-like and uORF1-KO? If not, please add a discussion on the reproducibility of the organoid protocol.

8) There is an extensive discussion on vimentin. However, there is a lack of vimentin expression in neurons (figure 2 of the response to the reviewers) and based on the organoid work, NPCs are the primary targets in which the vimentin has not been characterized. Therefore, the extended discussion on vimentin reorganization and its role in neuropathogenesis is highly speculative based on cell line results. There is not sufficient data to support the speculation on vimentin reorganization and neuropathogenesis. Please adjust or remove this section.

Reviewer #4:

Remarks to the Author:

Thank you for addressing my concerns and questions. I believe the revised text and figures are clear and the added experiments and image analysis make the study more robust. I have no further comments.

Reviewer #5:

Remarks to the Author:

The manuscript by Lefevre and colleagues investigates potential unrecognized upstream open reading frames (uORFs) in the 5' region of the Zika virus (ZIKV), presenting an interesting topic. However, the experiments conducted to explore the function of uORFs are not solid enough to support the conclusions. The proposed mechanism, where the uORF1-encoded protein assists in the formation of a cytoskeletal cage, is intriguing. But this effect is suggested to be general across all cell types. The manuscript does not adequately explain why this mechanism specifically contributes to neurotropism, leaving a significant gap in the understanding of the uORF1's specific role in Zika virus pathology.

The animal studies presented primarily focus on mosquitoes and do not align well with the central theme of neurotropism, as they lack experiments related to the mosquito nervous system.

Conversely, the absence of mammalian animal experiments to complement the findings observed in U251 cells is a notable omission. For the comments from Referee #1, the authors have addressed several queries, primarily concerning the technical and statistic issues. However, regarding logical concerns, the revisions and experimental results presented in the manuscript substantiate the referee's apprehensions that uORF1 and other uORFs may not significantly impact viral fitness or neurotropism changes.

The detailed comments on the authors' responses to Reviewer #1 are listed below:

Comment 1.1, the authors addressed the referee's questions by increasing the mosquito sample size and reorganizing the presentation of the data, making the conclusions more meaningful. The revised conclusion overturns the previous one regarding the influence of mutant viruses on mosquito transmission.

Comment 1.2, although the authors revised the description, it remains inaccurate because the

African ZIKV strains exhibit higher neurovirulence compared to the American strains [Nat Commun 12:916, 2021, serves as one example]. One possible explanation for the scarcity of clinical reports of microcephaly in Africa might be the inadequate medical facilities and surveillance systems.

Comment 1.3, addressed, but it should be noted that Dak84 is not the most commonly used African strain; MR766 is more widely recognized and utilized.

Comment 1.4, I can't find the previous Figure 2, but the authors have reorganized the figures as suggested by Reviewer #1, resulting in a clearer presentation.

Comment 1.5, the responses substantiate the reviewer's concerns that the observed increase in translation efficiency is due to the reduction in the internal control renilla expression, rather than an increase in firefly luciferase. This calls into question the reliability of the conclusions drawn.

Comment 1.6, It is a critical concern for the manuscript. The authors provided explanations, but these confirm the referee's concerns. The mutations utilized in this study may inhibit the translation of the correct polyprotein ORF, contributing to the observed changes in viral replication. Thus, the uORF2-KO/PCT1 mutant may not be significantly relevant to ZIKV replication.

Comment 1.7, the authors justified the use of 10:90 virus ratios for the competition assay, which is reasonable. However, the results of the competition assay raise concerns. In competition assays, both virus types have a high possibility of entering one specific cell, which means that the expression of uORF1 and uORF2 and any potential influence on viral replication will affect both viruses equally. Therefore, the observed higher fitness of the African-like/uORF1-KO ZIKV is likely due to structural changes from the introduced nucleotide mutations or other factors rather than the production of the uORF1 peptide.

Comment 1.8, similar to comment 1.2, the authors have revised the experiment to address statistical and technical concerns. However, the entire mosquito experiments are not highly relevant to the core focus of this manuscript.

Reviewer #6:

Remarks to the Author:

Response to Reviewers 2

We are very grateful to the reviewers for their second expert engagement with the work, their constructive criticisms and their suggestions on how to improve the manuscript. We have responded below to the questions point-by-point and describe some additional experiments that provide new insights, clarify issues and further support our findings, conclusions and mechanistic interpretations. We hope this further improved manuscript is now acceptable for publication in *Nature Communications*.

Reviewer #2 (Remarks to the Author):

I am fine with the revision.

Response: We thank the Reviewer for their satisfaction with the updated version.

Reviewer #3 (Remarks to the Author):

The authors have performed additional experiments and have addressed most of my comments. Overall, this work uses novel techniques to advance knowledge on potential determinants of Zika virus neurotropism. I have a few additional comments on the latest version of the manuscript below:

Comment 3.1: *The title still refers to multiple ORFs with a role in neurotropism. However, since uORF2 could not be characterized, the authors only show the role of uORF1 in neurotropism. Thus, the title needs to be adjusted to reflect this.*

Response: In this manuscript, we have identified the presence of two upstream open reading frames (uORFs) for the Asian/American ZIKV lineage termed uORF1 and uORF2 and a unique uORF for the African lineage termed African uORF. The Reviewer is correct in indicating that uORF2 could not be further characterised, but experiments presented in this piece of research indicate that the Asian/American uORF1 and the African uORF have a potential role in neurotropism; thus, we consider that the use of the word multiple is still valid in the title.

Comment 3.2: *Line 83, of the introduction states that "... uORF1 can modulate virus growth, virulence and tropism in the human brain". What is shown are results from human brain organoids and not the human brain so please adjust this.*

Response: This has now been adjusted and corrected in the text (Lines 85 – 86).

Comment 3.3: *In line 84, it is stated that the uORF1 is essential for neurotropism in the adult and developing brain. However, the work presented here does not show that. The brain organoids, to an extent, mimic the developing embryonic brain but not the adult brain.*

Response: This has now been adjusted and corrected in the text (Lines 85 – 86).

Comment 3.4: *It is not clear how the data in Figure 3 supports the statement in lines 236-238. Would it not simply be the case that the reversion is necessary because it results in increased translation of the main ORF rather than the added role for uORF2 in translation?*

Response: This is indeed a better interpretation of the data. Mutating the original initiation codon of uORF2 (UUG) into UUA (uORF2-KO mutant) is detrimental to polyprotein expression, as demonstrated with the luciferase reporter assay (**Fig 2E**, right panel). Thus, the rapid reversion of the mutated UUA into the original UUG suggests the necessity of maintaining a functional initiation codon and, therefore, uORF2 translation for the correct expression of the viral polyprotein. This has now been clarified in Lines 234 – 241.

Comment 3.5: *Please add, in figure 6, the E+ cells/total nuclei data for the i3Neurons infected with ZIKV.*

Response: To prevent potential redundancy with **Fig 6A**, the percentage of E⁺ cells in relation to the total number of nuclei in i³Neurons infected with the different ZIKV mutant viruses has now been included as **Supp Fig 16B**. These data confirm that infection with African-like and uORF1-KO mutant viruses is limited to single cells, suggesting little viral spread. This has now been included in Lines 366 – 368.

Comment 3.6: *The identity of the organoids is still not clear. In lines 353-354, they are called cerebral cortical organoids. Cerebral organoids was the term used for whole-brain organoids (the proposed nomenclature refers to them as unguided neural organoids). Do these slice cultures have whole-brain identity or they only represent the cerebral cortex (making them regionalized neural organoids).*

Response: The air-liquid interface cerebral organoids (ALI-COs) were generated using an unguided differentiation and slicing at 50 days *in vitro* to overcome size-related restrictions to nutrient diffusion, allowing improved core tissue survival and cell differentiation processes (Giandomenico *et al.*, 2019 PMID: 30886407; Szebenyi *et al.*, 2021 PMID: 34675437). This typically generates organoids with a dorsal forebrain identity with high consistency in cell types and tissue architecture, including the cortical plate structures that form the basis of our observations. Apart from minor variations, we demonstrated very similar cell type composition by scRNA-seq and immunolabelling approaches between organoids derived from different cell lines and batches (Szebenyi *et al.*, 2021 PMID: 34675437).

Based on this, to keep the terminology consistent, we refer to ALI-COs as cerebral organoids throughout the text now.

Comment 3.7: *In figure 6, the regional organization of the organoid can only be seen in the American WT conditions in 6D and 6E. Although the other two panels show NPCs and "astrocyte-precursors" (which would also be neural stem cells), there is a lack of organization. Can the authors show a similarly organized panel for the African-like and uORF1-KO? If not, please add a discussion on the reproducibility of the organoid protocol.*

Response: We agree with the Reviewer that the regional organisation of the organoid is mostly observed in **Figures 6D** and **6E**, representing cell infections with the American WT virus but also with the African-like virus (**Fig 6D**). At 82 days *in vitro*, some remnants of progenitor zones should be observed; however, the extent could be variable within each organoid, based on the spatial location and cell differentiation state. To better highlight the structure of these zones and assess the neurotropism more consistently, infected ALI-COs were stained with SOX2, a progenitor cell marker helping define germinal zones. As described in **Responses Figure 1**, SOX2 distribution (in magenta) was similar in the different conditions, indicating that the cortical plates are consistently found in organoids with only minor variability, which justifies the comparisons in the infection efficacy between the different virus types. We

included immunofluorescence images below to demonstrate this point, and several examples of progenitor zones are indicated by dotted white squares.

Additionally, it drew our attention to a reduction of SOX2⁺ cells in infected cells (E⁺ cells, in green), which is extremely noticeable in American WT and African-like infected ALI-COs. This phenomenon has been previously described in infected macaque fetuses (Adams Waldorf *et al.*, 2018 PMID: 29400709; Tisoncik-Go *et al.*, 2024 PMID: 38890352), and it will be subjected to future research.

Responses Figure 1: Representative images (40X resolution) of ALI-COs infected with the American WT, the African-like and the uORF1-KO viruses (MOI:5) for 7 days showing immunoreactivity for the viral E protein (green) and SOX2 (magenta), a progenitor marker (AF2018-NOVUS, 1:200 dilution). Nuclei were counter-stained with DAPI (blue). Scale bars, 25 μ m.

Comment 3.8: *There is an extensive discussion on vimentin. However, there is a lack of vimentin expression in neurons (figure 2 of the response to the reviewers) and based on the organoid work, NPCs are the primary targets in which the vimentin has not been characterized. Therefore, the extended discussion on vimentin reorganization and its role in neuropathogenesis is highly speculative based on cell line results. There is not sufficient data to support the speculation on vimentin reorganization and neuropathogenesis. Please adjust or remove this section.*

Response: This is a reasonable point. To address this comment, we stained infected ALI-COs with vimentin. As observed in the new **Supplementary Figure 18**, vimentin collapses in infected cells, forming a cytoskeletal cage similar to the one described for Vero and U251-infected cells (**Fig 5** and **Supp Fig 15**). This pattern is completely different to the filamentous

one in mock-infected cells. Note that these ALI-COs were harvested at 7 days post-infection, and therefore, the dynamics of the collapse of vimentin with the different mutant viruses could not be quantified throughout a time course of infection. This will be investigated further in the future.

Additionally, the potential role in neuropathogenesis and microcephaly of the collapse of intermediate filaments during ZIKV infection has not been assessed yet and will be the subject of future research. The discussion section has now been modified accordingly (Lines 530 – 541).

Reviewer #4 (Remarks to the Author):

Thank you for addressing my concerns and questions. I believe the revised text and figures are clear and the added experiments and image analysis make the study more robust. I have no further comments.

Response: We thank the Reviewer for their satisfaction with the updated version.

Reviewer #5 (Remarks to the Author):

The manuscript by Lefevre and colleagues investigates potential unrecognized upstream open reading frames (uORFs) in the 5' region of the Zika virus (ZIKV), presenting an interesting topic. However, the experiments conducted to explore the function of uORFs are not solid enough to support the conclusions.

Comment 5.1: *The proposed mechanism, where the uORF1-encoded protein assists in the formation of a cytoskeletal cage, is intriguing. But this effect is suggested to be general across all cell types. The manuscript does not adequately explain why this mechanism specifically contributes to neurotropism, leaving a significant gap in the understanding of the uORF1's specific role in Zika virus pathology.*

Response: Indeed, the precise molecular details of how uORF1 specifically functions in neurotropism are not yet fully understood, but the work detailed in this manuscript has identified the previously unknown molecular players, tested their role and activity and has defined a completely new area of ZIKV research. Specifically relevant to the question, we have determined that a virus unable to express the Asian/American uORF1 severely delays the formation of the cytoskeletal cage in infected tissue culture cells and that the African-like 5' UTR arrangement is less efficient in promoting the collapse of vimentin. We have also found that the expression of the African uORF and the uORF1 modulates the ability of ZIKV to infect and replicate in our primary cultures of the nervous system (e.g., neural progenitor cells, cells of the astroglial lineage and cortical neurons), thus neurotropism.

In future research, we will hope to determine the precise molecular mechanisms of how the uORF1 and the African uORF help in the collapse of vimentin and potentially other intermediate filaments in iPSC-derived astrocytes and neural progenitor cells (NPCs). In addition, using previously established high-throughput techniques in the lab (e.g., RNA-Seq and Ribo-Seq), we will assess the ability of the wild-type and uORF mutant viruses to cause pathogenesis in tractable models of human neurotropism such as cerebral organoids. Overall, we believe that defining and characterising the key players is a massive first step in understanding ZIKV pathology.

Comment 5.2: *The animal studies presented primarily focus on mosquitoes and do not align well with the central theme of neurotropism, as they lack experiments related to the mosquito*

nervous system. Conversely, the absence of mammalian animal experiments to complement the findings observed in U251 cells is a notable omission.

Response: Experiments carried out on female *Aedes aegypti* sought only to determine the transmission dynamics (i.e., infection, dissemination, and transmission prevalence) of the different mutant ZIKV viruses in the mosquito vector. Although studying ZIKV neurotropism and neuropathogenicity in mosquitoes would be of added value, the lack of major behavioural alterations observed in infected mosquitoes (Maire *et al.*, 2024 PMID: 38423938) indicates that their nervous system remains functional upon infection probably due to neuron-specific antiviral mechanisms such as the *Aedes aegypti* homologue of the neural factor *Hiraku genki* (AaHig) that efficiently restricts flavivirus infection of the central nervous system (Xiao *et al.*, 2015 PMID: 25915054).

Apart from non-human primates (NHP), which are excellent animal models for congenital Zika syndrome (CZS) due to their genetic closeness to humans (e.g., placental organisation and long gestation periods) (Narasimhan *et al.*, 2020 PMID: 33091001), there is a lack of a good *in vivo* model to study ZIKV pathogenesis. However, studies using NHP models are arduous, lengthy, very expensive and raise ethical challenges for host Institutions.

Alternatively, many studies have utilised the transplacental mouse model (Miner *et al.*, 2016 PMID: 27180225; Jagger *et al.*, 2017 PMID: 28910635; Jaeger *et al.*, 2019 PMID: 30995223), but it also has some limitations. For example, it is not possible to experimentally infect these mice early in gestation, before embryonic day 9 (E9), as this results in pregnancy loss. Furthermore, although the microcephaly phenotype tends to mirror that of human infection with enlarged cerebral ventricles (ventriculomegaly) (Jagger *et al.*, 2017 PMID: 28910635; Jaeger *et al.*, 2019 PMID: 30995223), in some cases, ZIKV infection produces embryonic death before E18.5 (Aubry *et al.*, 2021 PMID: 33568638) or infected mice pups show shrunken ventricles instead (Wu *et al.*, 2016 PMID: 27174054). In addition, it is also important to note that mice are not generally permissive to ZIKV infection, so it is necessary to hinder the type I IFN signalling system to allow ZIKV replication, often by genetic modification or chemical manipulation (e.g., type I IFN receptor specific blocking antibody) (Miner *et al.*, 2016 PMID: 27180225).

Therefore, to study ZIKV neurotropism, we decided to use a reliable and tractable model for human neurotropism, such as cerebral organoid slices grown at the air-liquid interface (ALI-COs). These slices recapitulate cortical cell type diversity, layering, and neurodevelopmental milestones (Giandomenico *et al.*, 2019 PMID: 30886407; Szebenyi *et al.*, 2021 PMID: 34675437). Today, even with their limitations, ALI-COs are one of the best models to recapitulate what happens in the brain during gestation.

For the comments from Referee #1, the authors have addressed several queries, primarily concerning technical and statistical issues. However, regarding logical concerns, the revisions and experimental results presented in the manuscript substantiate the referee's apprehensions that uORF1 and other uORFs may not significantly impact viral fitness or neurotropism changes.

The detailed comments on the authors' responses to Reviewer #1 are listed below:

Comment 1.1: *The authors addressed the referee's questions by increasing the mosquito sample size and reorganizing the presentation of the data, making the conclusions more meaningful. The revised conclusion overturns the previous one regarding the influence of mutant viruses on mosquito transmission.*

Response: We apologise for this confusion, but we never detected any statistically significant differences between virus strains in terms of infection, dissemination or transmission prevalence in female mosquitoes as indicated in the previous version. For the current version, we have only substituted the misleading visual impression that the African-like virus had reduced dissemination due to the poor visibility of the confidence intervals of the logistic fits by shaded ribbons representing the confidence intervals with vertical error bars for each data point (**Figure 7**).

Comment 1.2: *Although the authors revised the description, it remains inaccurate because the African ZIKV strains exhibit higher neurovirulence compared to the American strains [Nat Commun 12:916, 2021, serves as one example]. One possible explanation for the scarcity of clinical reports of microcephaly in Africa might be the inadequate medical facilities and surveillance systems.*

Response: This is a valid point. In the updated version of this manuscript, we have included this potential explanation for why microcephaly has not been detected in Africa and the suggested reference (Lines 67 – 69).

Comment 1.3: *Addressed, but it should be noted that Dak84 is not the most commonly used African strain; MR766 is more widely recognized and utilized.*

Response: We agree with the Reviewer that the MR766 strain, isolated in 1947 in Uganda, is more widely recognised and utilised. However, in order to grow the virus for continued experimental studies, the strain was passaged approximately 150 times through mice brains, raising adaptative mutations to optimise growth in neural tissue (Wetsman, 2017 PMID: 28777794). As a result, experimental data based on the use of the MR766 strain should not be used to reach definitive conclusions about the virulence of African ZIKV (Wetsman, 2017 PMID: 28777794). This is why we decided to use the Dak84 strain passaged only three times.

Comment 1.4: *I can't find the previous Figure 2, but the authors have reorganized the figures as suggested by Reviewer #1, resulting in a clearer presentation.*

Response: We thank the Reviewer for this comment.

Comment 1.5: *the responses substantiate the reviewer's concerns that the observed increase in translation efficiency is due to the reduction in the internal control renilla expression, rather than an increase in firefly luciferase. This calls into question the reliability of the conclusions drawn.*

Response: The reviewer is correct. The raw luciferase values (**Supp Table 1**) were slightly lower in the presence of the virus, reflecting some impairment of translation initiation as a result of the phosphorylation of the alpha subunit of the initiation factor 2 (p-eIF2 α) during infection (**Supp Fig 10B**). However, the expression of ZIKV FF-Luc reporters (uORFs and main ORF) was decreased less upon infection than other cellular mRNAs (i.e., Ren-Luc mRNA), thus increasing the FF-Luc/Ren-Luc ratio. Therefore, we hypothesise that the viral 5' UTR arrangement could potentially protect the expression of the uORFs and the main ORF compared to other cellular mRNAs as a response to cellular stress (i.e., eIF2 α phosphorylation), similarly to the regulation of the activation transcription factor 4 (ATF4) (Somers *et al.*, 2013 PMID: 23624144). We have further clarified this in the manuscript (Lines: 211 – 213).

Comment 1.6: *It is a critical concern for the manuscript. The authors provided explanations, but these confirm the referee's concerns. The mutations utilized in this study may inhibit the translation of the correct polyprotein ORF, contributing to the observed changes in viral*

replication. Thus, the uORF2-KO/PCT1 mutant may not be significantly relevant to ZIKV replication.

Response: We disagree with the Reviewer's comment. The mutations utilised in this study correspond to the uORFs, and none was introduced in the main ORF coding region. Introduced mutations were tested in luciferase reporters (**Fig 2E**), and they could increase the translation of the main ORF (i.e., uORF1-KO and African-like), maintain it (i.e., uORF2-PTC1) or reduce it (i.e., uORF2-KO and uORF2-AUG). Mutant viruses were generated by reverse genetics introducing the aforementioned mutations (**Fig 3**), and results obtained by growth kinetics and competition assay experiments mirrored the luciferase results (**Fig 3C** and **Fig 3D**). uORF1-KO and African-like mutant viruses, with an increased translation of the main ORF, reached significantly higher titres and were fitter in competition assays than the American WT. No difference was observed with the uORF2-PTC1 mutant virus compared to the American WT, as it was not observed in the luciferase assays. However, this mutation led us to understand that uORF2 could be translated by an alternative mechanism, a -1 ribosomal frameshifting (**Supp Fig 12**). Any virus was recovered following electroporation of uORF2-AUG mutant due to reduced translation initiation at the main polyprotein AUG as a consequence of increased recognition of the uORF2 AUG start codon as observed with the luciferase reporter assay (**Fig 2E**). Lastly, the reduction in main ORF expression by luciferase assays and the rapid reversion to the wild-type sequence in the uORF2-KO mutant virus have indicated the necessity to maintain the uORF2 initiation codon for proper virus replication. However, the role of uORF2 translation in virus replication will need to be further investigated in future work.

Comment 1.7: *The authors justified the use of 10:90 virus ratios for the competition assay, which is reasonable. However, the results of the competition assay raise concerns. In competition assays, both virus types have a high possibility of entering one specific cell, which means that the expression of uORF1 and uORF2 and any potential influence on viral replication will affect both viruses equally. Therefore, the observed higher fitness of the African-like/uORF1-KO ZIKV is likely due to structural changes from the introduced nucleotide mutations or other factors rather than the production of the uORF1 peptide.*

Response: We have provided abundant experimental evidence of the biological activity of the uORF1 peptide, and this is highly consistent with our interpretation that uORF1 levels in mutant viruses are likely a key determinant of fitness. Further, in competition assays, as indicated in Line 248, cells were infected with both viruses at a final multiplicity of infection (MOI) of 0.01 (PFU/cell). When cells are infected at a 50:50 ratio, the MOI of each individual virus is 0.005. Under these circumstances, the Poisson distribution reveals that 0.498% of cells would be a target for either virus and the probability of dual infection is only 0.00017%. Even when the ratio of 10:90 is used, the frequency of dual infection remains extremely low. At 72 h, the supernatant virus was diluted 1:10,000 before infection, corresponding to an MOI in the range ~0.1 as previously estimated (Lulla and Firth, 2019 PMID: 31192271), again with a very low chance of dual infection (0.067% at 50:50). Thus, any influence on virus replication is not related to dual infection. (Please note that in the submission, there was an error where it was stated [Line 254] that the dilution was 1:200, whereas it was, in fact, 1:10,000 as indicated in the Materials and Methods section (Line 898). This has been corrected in the updated version).

The authors are extremely aware that structured RNA elements and long-range interactions in the 5' and 3' terminal regions of the ZIKV genome are essential for virus translation and replication. Therefore, we analysed the 5' end structures in full-length RNA transcripts of the mutant viruses by using selective 2'-hydroxyl acylation analysed by primer extension (SHAPE). We found that the structure of the 5' UTR and the start of the main ORF of the mutant viruses very closely matched that of the American WT infectious clone (**Fig 3B** and **Supp Fig 11A**). A small difference was found with the modelling for the African-like mutant

virus that had a slightly shorter 3rd stem-loop (cHP), with loss of two base pairs at the bottom of the helix (**Fig 3B**), however, this 5' end arrangement naturally occurs in the ZIKV African lineage. This has been already described in the current version of the manuscript (Lines 228 – 231). Thus, we do not consider that RNA structural changes are responsible for the viral fitness of these mutant viruses.

Comment 1.8: *Similar to comment 1.2, the authors have revised the experiment to address statistical and technical concerns. However, the entire mosquito experiments are not highly relevant to the core focus of this manuscript.*

Response: As ZIKV typically cycles between humans and *Aedes* mosquitoes, we considered it very important to test whether ZIKV uORFs also had a detectable effect on the mosquito vector. Our experiments concluded that although uORFs were expressed in mosquito cells, we did not see a measurable effect on transmission by the mosquito vector *in vivo*, suggesting that uORFs only play a role in the mammalian host.

Reviewer #6 (Remarks to the Author):

Response: This reviewer did not raise any specific comments.

Reviewers' Comments:

Reviewer #3:

Remarks to the Author:

The authors have sufficiently addressed my comments from the previous review round. I would like to congratulate the authors on an impressive piece of work.

Reviewer #5:

Remarks to the Author:

The core concerns raised by the reviewer have not been clearly or logically addressed in the responses. While the study demonstrates the existence of uORFs in both vertebrate and invertebrate cells, their biological functions are still not adequately supported.

Below is the detailed response and discussion:

For comment 5.1: The authors still did not explain why targeting the cytoskeletal cage with uORFs results in neurotropism without affecting other cell types.

For comment 5.2: The authors investigate transmission dynamics but do not compare neurotropism in mosquitoes, making this experiment less meaningful for supporting the core hypothesis. Transgenic mice, readily available for ZIKV infection and neuro-related symptom studies, would provide critical information to support the conclusion and are easier to obtain than organoids. The authors discuss several animal models in their response but do not explain why they didn't confirm the function of uORFs in mammalian models.

For comment 1.3: Despite being passaged hundreds of times, MR766 retains high infectivity in neuron cells and animals compared to the Asian lineage virus. High passage numbers do not hinder neurotropism studies with this strain.

For comment 1.5: The authors did not clearly address the technical question raised by reviewer 1. The reduced expression of the Renilla control contributed to the increased FF-Luc/Ren-Luc ratio, making some results unreliable.

For comment 1.6: The new response reiterates the manuscript's observations without directly or logically addressing Reviewer 1's critical concerns.

For comment 1.7: The response does not address the concern. Regardless of the initial MOI, both mutant and wild-type viruses will compete with progeny viruses due to culture and multiple passages. This competition suggests that the uORF1 peptide may not be the dominant factor for higher fitness.

For comment 1.8: The dynamics of transmission are less relevant to the core focus. Detecting the

growth kinetics of different mutant viruses in the central nervous system of mosquitoes or imaging infected mosquito brain slices would be more meaningful for investigating the function of uORFs in invertebrate.

Reviewer #6:

Remarks to the Author:

Response to Reviewers 3

We are very grateful to the reviewers for their third expert engagement with the work. We have responded below to the questions point-by-point.

Reviewer #3 (Remarks to the Author):

The authors have sufficiently addressed my comments from the previous review round. I would like to congratulate the authors on an impressive piece of work.

Response: We thank the Reviewer for this kind comment.

Reviewer #5 (Remarks to the Author):

The core concerns raised by the reviewer have not been clearly or logically addressed in the responses. While the study demonstrates the existence of uORFs in both vertebrate and invertebrate cells, their biological functions are still not adequately supported.

Below is the detailed response and discussion:

For comment 5.1: *The authors still did not explain why targeting the cytoskeletal cage with uORFs results in neurotropism without affecting other cell types.*

Response: We have shown that mutant viruses lacking American uORF1 (uORF1-KO and African-like viruses) less efficiently infect human brain cells and are slower in promoting the collapse of intermediate filaments and, thus, the formation of the cytoskeletal cage. We agree with the reviewer that the precise molecular details that link these effects are not yet fully understood, but investigating this is beyond the scope of the present manuscript. Testing the phenomenon in other primary cell types and in tractable models for human neurotropism will form the basis of a separate future manuscript.

For comment 5.2: *The authors investigate transmission dynamics but do not compare neurotropism in mosquitoes, making this experiment less meaningful for supporting the core hypothesis. Transgenic mice, readily available for ZIKV infection and neuro-related symptom studies, would provide critical information to support the conclusion and are easier to obtain than organoids. The authors discuss several animal models in their response but do not explain why they didn't confirm the function of uORFs in mammalian models.*

Response: We did not aim to investigate neurotropism in mosquitoes (mosquitoes do not obviously display neurological symptoms [Maire *et al.*, 2024 PMID: 38423938; Xiao *et al.*, 2015 PMID: 25915054]), rather our goal was to determine whether Zika virus uORFs confer any fitness defect or advantage in the vector.

We agree with the Reviewer that some work in mouse models, such as the transplacental one, could confirm some of the uORF functions in mammals. However, we do not have access to this model, and it will take several months, if not years, to establish it. We consider this to be beyond the scope of the present manuscript.

For comment 1.3: *Despite being passaged hundreds of times, MR766 retains high infectivity in neuron cells and animals compared to the Asian lineage virus. High passage numbers do not hinder neurotropism studies with this strain.*

Response: As previously indicated, the MR766 strain was passaged approximately 150 times through mice brains, raising adaptative mutations to optimise growth in neural tissue (Wetsman, 2017 PMID: 28777794). Although studying some of these adaptative mutations might be extremely interesting in determining how Zika virus increases neurotropism, this is beyond the scope of the present manuscript, and we decided to test two low-passaged clinical isolates (i.e., PE243 and Dak84).

For comment 1.5: *The authors did not clearly address the technical question raised by reviewer 1. The reduced expression of the Renilla control contributed to the increased FF-Luc/Ren-Luc ratio, making some results unreliable.*

Response: In the updated version of this manuscript (Lines 209 – 212), we have toned down our interpretation of these data and acknowledged that further work would be needed to confirm the protective effect of the ZIKV 5' UTR against global translation shutdown.

For comment 1.6: *The new response reiterates the manuscript's observations without directly or logically addressing Reviewer 1's critical concerns.*

Response: We disagree with the Reviewer on this point. The mutations utilised in this study correspond to the uORFs, and none were introduced in the main ORF coding region. Thus, the translation of the main ORF corresponding to the polyprotein is correct.

However, we acknowledge that the possible reversion of the different uORF2 mutants to wild-type is somewhat speculative, and further investigation is required to confirm our conclusions. This will include the design and testing of alternative uORF2 knockout strategies. This limitation has now been acknowledged in the updated version of this manuscript (Lines 233 – 240).

For comment 1.7: *The response does not address the concern. Regardless of the initial MOI, both mutant and wild-type viruses will compete with progeny viruses due to culture and multiple passages. This competition suggests that the uORF1 peptide may not be the dominant factor for higher fitness.*

Response: As we explained in our previous response, competition between 'parental viruses' added at the beginning of the experiment and progeny viruses due to culture and multiple passages was prevented by diluting the harvested medium 1:10,000 before infecting fresh cells in the new passage. The resultant MOI is in the range of ~0.1, as previously estimated (Lulla and Firth, 2019 PMID: 31192271), again with a very low chance of dual infection (0.067% at 50:50). Thus, any influence on virus replication is not related to dual infection.

A potential explanation of why the uORF1-KO and African-like mutant viruses are fitter in tissue culture cells compared to the American WT is indicated in the Discussion section of this manuscript (Lines 458 – 468). This is potentially due to the role of uORFs in modulating the downstream main ORF translation. In addition, the higher viral fitness of the mutant viruses can also be related to the use of immortalised cell lines in these competition assays (e.g., U251 cells derived from a malignant glioblastoma tumour). Tumour-derived immortalised cell lines have defects in the induction of interferon-stimulated genes and other antiviral defences. These defects can significantly affect virus replication, especially when cells are infected at lower, more physiologically relevant, multiplicities of infection.

For comment 1.8: *The dynamics of transmission are less relevant to the core focus. Detecting the growth kinetics of different mutant viruses in the central nervous system of mosquitoes or imaging infected mosquito brain slices would be more meaningful for investigating the function of uORFs in invertebrate.*

Response: See previous comment 5.2.

Reviewer #6 (Remarks to the Author):

Response: This Reviewer did not raise any specific comments.